# GSBA$^{\mathrm{K}}$: *top-K* GEOMETRIC SCORE-BASED BLACK-BOX ATTACK

**Md Farhamdur Reza**[1,*], **Richeng Jin**[2,*], **Tianfu Wu**[1] & **Huaiyu Dai**[1]
[1]NC State University    [2]Zhejiang University
{mreza2, tianfu_wu, hdai}@ncsu.edu   richengjin@zju.edu.cn

## ABSTRACT

Existing score-based adversarial attacks mainly focus on crafting *top-1* adversarial examples against classifiers with single-label classification. Their attack success rate and query efficiency are often less than satisfactory, particularly under small perturbation requirements; moreover, the vulnerability of classifiers with multi-label learning is yet to be studied. In this paper, we propose a comprehensive surrogate free score-based attack, named geometric s̲core-based b̲lack-box a̲ttack (GSBA$^{\mathrm{K}}$), to craft adversarial examples in an aggressive *top-K* setting for both untargeted and targeted attacks, where the goal is to change the *top-K* predictions of the target classifier. We introduce novel gradient-based methods to find a good initial boundary point to attack. Our iterative method employs novel gradient estimation techniques, particularly effective in *top-K* setting, on the decision boundary to effectively exploit the geometry of the decision boundary. Additionally, GSBA$^{\mathrm{K}}$ can be used to attack against classifiers with *top-K* multi-label learning. Extensive experimental results on ImageNet and PASCAL VOC datasets validate the effectiveness of GSBA$^{\mathrm{K}}$ in crafting *top-K* adversarial examples.

## 1 INTRODUCTION

Deep neural networks (DNNs) are vulnerable to adversarial examples (Goodfellow et al., 2014; Moosavi-Dezfooli et al., 2016; Jia et al., 2022). In white-box attacks (Szegedy et al., 2013; Carlini & Wagner, 2017; Moosavi-Dezfooli et al., 2016; Madry et al., 2017), an adversary possesses complete access to the internal structure and parameters of the target DNN, whereas in black-box attacks, this information is not available, making them more practical in real-world scenarios. Black-box attacks can be of two types: transfer-based (Dong et al., 2018; Wang & He, 2021; Wang et al., 2024; Li et al., 2020c; Wei et al., 2023) and query-based, where the former crafts adversarial examples exploiting a surrogate model, and the latter makes queries for the outputs from the target classifier to craft adversarial examples intended to deceive it. Within query-based black-box attacks, two subcategories exist: decision-based (Chen et al., 2020; Ma et al., 2021; Reza et al., 2023) and score-based (Guo et al., 2019a; Ilyas et al., 2018b;a; Andriushchenko et al., 2020) attacks. In the former, the adversary has access to the *top-1* predicted label from the target model, while in the latter, the adversary can retrieve the full set of prediction probabilities for all classes.

Numerous endeavors have been undertaken towards effective score-based black-box attacks (Chen et al., 2017; Bhagoji et al., 2018; Ilyas et al., 2018a; Guo et al., 2019a) against classifiers with single-label multi-class classification, where the classifiers' goal is to predict the *top-1* classification label corresponding to an input. Score-based adversarial attacks can be either gradient-based (Ilyas et al., 2018a;b; Chen et al., 2017; Bhagoji et al., 2018), or gradient-free (Andriushchenko et al., 2020; Guo et al., 2019a; Li et al., 2020b). Gradient-based methods rely on small perturbations in the gradient direction to steer the input towards the adversarial region, while gradient-free methods use some predefined random directions for the same purpose. While Square Attack (Andriushchenko et al., 2020), in score-based setting, offers state-of-the-art performance, it suffers from low success rates and query inefficiency, particularly when constrained by small perturbation thresholds.

In recent years, leveraging the geometry of the decision boundary has proven to enhance the efficiency and effectiveness of decision-based black-box attacks (Liu et al., 2019; Rahmati et al., 2020; Reza et al., 2023). Geometric decision-based attacks, starting from a random point on the decision boundary, iteratively refine the adversarial example exploring this boundary. However, such efforts

---

*Corresponding authors

are largely lacking for score-based attacks. A simple comparison between the usual approach of existing score-based attacks and the state-of-the-art decision-based attack CGBA (Reza et al., 2023) in finding adversarial examples considering a linear boundary in a 2D space is shown in Fig. 1. Considering imperfect gradient estimation, the geometric-based attack CGBA finds a better adversarial example along a semicircular path, starting from a random boundary point $x_{b_0}$, by exploring the decision boundary. This raises an important open question: *can the geometric properties of high-dimensional image space boundaries be harnessed to advance the field of score-based attacks?*

Traditionally, adversarial attacks have predominantly focused on generating *top-1* adversarial examples against single-label multi-class classifiers for untargeted and/or targeted attacks, wherein a well-crafted adversarial example replaces the single true label of the input image with an arbitrary label for untargeted attacks and a specific target label for targeted attacks. However, in numerous real-world applications such as web search engines, image annotation, recommendation systems, and computer vision APIs like Google Cloud Vision, Microsoft Azure Computer Vision, Amazon Rekognition, and IBM Watson Visual Recognition, the *top-K* predictions provide valuable information about the input. Thus, recently, a couple of white-box attacks (Zhang & Wu, 2020; Hu et al., 2021; Tursynbek et al., 2022; Paniagua et al., 2023) have been proposed in an aggressive *top-K* setting where the *top-K* prediction labels of an input are replaced by an arbitrary set of mutually exclusive wrong labels for untargeted attacks (Tursynbek et al., 2022), and by a given set of target labels for targeted attacks (Zhang & Wu, 2020; Paniagua et al., 2023). Among

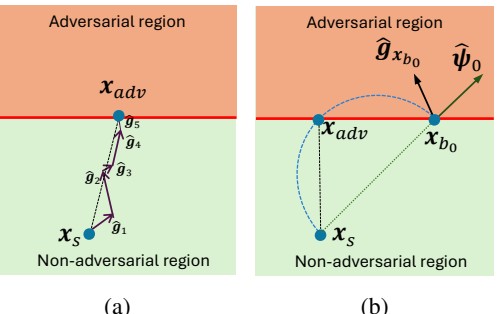

(a)  (b)

Figure 1: (a) Traditional score-based attack approach: adversarial example $x_{adv}$ is generated by iteratively adding perturbations in the direction of the estimated gradient or random directions with source $x_s$. Perturbations are weighted based on the increase in confidence toward the adversarial region. (b) Geometric-based approach CGBA (Reza et al., 2023) in finding a better adversarial example exploring the decision boundary.

these attacks, $T_kML$-AP (Hu et al., 2021) targets classifiers with multi-label learning, where the classifiers' goal is to learn multiple meaningful true labels from an image. These white-box attacks, having full access to the target classifier, can calculate the true gradient to navigate towards the adversarial region and craft *top-K* adversarial examples with high attack success rate (ASR), demonstrating the underlying vulnerability of DNNs. However, this task becomes much more challenging in the more practical black-box setting with only predicted probabilities of all classes are available, as we lack accurate intermediate gradients for navigation in the high-dimensional continuous space. In this case, identifying an initial boundary point satisfying the *top-K* target-label constraints may be akin to finding a needle in an ocean. On top of that, we need to further refine the obtained adversarial example to meet the perturbation threshold constraint across the highly irregular adversarial region.

To address the aforementioned challenges, we introduce a comprehensive query-efficient geometric score-based *top-K* black-box attack, GSBA$^K$, that employs distinct approaches to approximate gradients by querying the target classifier, aiming to efficiently identify good initial boundary points for both untargeted and targeted attacks rather than starting from a random boundary point as done by geometric decision-based attacks. However, the obtained initial boundary points, utilizing the estimated gradient direction, often significantly deviate from the optimal. Thus, we further

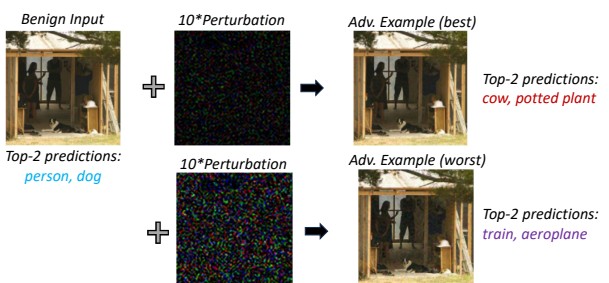

Figure 2: Crafted *top-2* adversarial examples against Inception-V3 (Szegedy et al., 2016) with *top-2* multi-label learning on the PASCAL VOC 2012 dataset (Everingham et al., 2015) by considering the best and worst target-label sets (Sec. 5.2). The prediction order based on confidence scores of the benign input is: [*person, dog, cow, potted plant, horse, chair, car, dining table, bottle, sofa, cat, tv/monitor, bicycle, sheep, boat, motorbike, bird, bus, train, aeroplane*].

introduce more accurate gradient estimation techniques *at boundary points* by leveraging the prediction probabilities. Guided by the estimated gradient on the decision boundary, GSBA$^K$ conducts a boundary point search along a semicircular trajectory, motivated by the state-of-the-art (SOTA) decision-based attack CGBA (Reza et al., 2023), to explore the decision boundary and further optimize the perturbation. GSBA$^K$ is designed to perform attacks not only on traditional single-label multi-class classification problems but also on classifiers with *top-K* multi-label learning capabilities; see Fig. 2 for crafted *top-2* adversarial examples. The contributions of this paper are summarized as follows:

- We propose GSBA$^K$, a comprehensive and query-efficient geometric score-based attack in an aggressive *top-K* setting. GSBA$^K$ incorporates novel gradient estimation techniques to locate a better initial boundary point and leverages the geometric properties of decision boundaries to enhance both query efficiency and attack versatility.
- In the difficult *top-K* targeted attack, our gradient estimators assess the impact of each query on the individual target classes and assign adaptive weights based on their significance.
- We adapt the SOTA score-based Square Attack (SA) (Andriushchenko et al., 2020) to the *top-K* setting to serve as a baseline. Comprehensive experiments on ImageNet (Deng et al., 2009) and PASCAL VOC 2012 (Everingham et al., 2015) datasets against popular classifiers underscore the efficacy of GSBA$^K$ in handling both *top-1* and complicated *top-K*, including multi-label learning scenario, across untargeted and targeted settings.

## 2 RELATED WORK

**Black-box adversarial attacks.** The *top-1* classification label is the sole piece of information available to an adversary in **decision-based attacks**. These attacks can be either gradient-free (Brendel et al., 2018; Brunner et al., 2019; Li et al., 2021; Dong et al., 2019; Maho et al., 2021) or they can involve estimating the gradient on the decision boundary (Chen et al., 2020; Li et al., 2020a; Rahmati et al., 2020; Reza et al., 2023). Based on the use of the geometric properties of the decision boundary, decision-based adversarial attacks can also be categorized as geometric decision-based adversarial attacks (Rahmati et al., 2020; Maho et al., 2021; Ma et al., 2021; Wang et al., 2022; Reza et al., 2023). While Tangent Attack (Ma et al., 2021) considers the decision boundary as a virtual hemisphere to refine the boundary point, Triangle Attack (Wang et al., 2022) used the triangle inequality to refine it. GeoDA (Rahmati et al., 2020) and SurFree (Maho et al., 2021) focus on the hyperplane boundary, with GeoDA using estimated gradient information to execute the attack, while SurFree is gradient-free. In contrast, CGBA (Reza et al., 2023) demonstrates the difference in curvature of the decision boundaries for untargeted and targeted attacks, based on which algorithms are proposed that go beyond the simplified hyperplane boundary model and exploit the distinct curvatures of the decision boundary for improved attack performance.

In **score-based attacks**, an adversary avails itself of the information of prediction probabilities of all the classes when querying the target classifier. Although most score-based attacks operate without surrogate models, some approaches (Guo et al., 2019b; Cheng et al., 2019; Yang et al., 2020) incorporate surrogate models to improve efficiency. Among the surrogate free attacks, ZOO (Chen et al., 2017) employs the finite difference method with dimension-wise estimation to approximate the gradient, requiring $2d$ queries per iteration, where $d$ is the dimension of the image. To improve the query efficiency of gradient estimation, (Bhagoji et al., 2018) reduces the search space using PCA of the input data, while AutoZOOM (Tu et al., 2019) samples noise from the low-dimensional latent space of a trained auto-encoder. NES (Ilyas et al., 2018a) uses finite differences through natural evolution strategies for gradient estimation. In the quest for further improved query efficiency, Bandits (Ilyas et al., 2018b) incorporates two priors: a time-dependent prior and a data-dependent prior. All the aforementioned gradient-based attacks iteratively add noise toward the estimated gradient direction to craft adversarial examples. SimBA (Guo et al., 2019a), however, queries along a set of orthonormal directions to obtain adversarial perturbations. Despite its simplicity, SimBA outperforms the gradient-based methods. PPBA (Li et al., 2020b) introduces a projection and probability-driven untargeted attack, focusing on reducing the solution space by employing a low-frequency constrained sensing matrix to enhance query efficiency. Conversely, SA (Andriushchenko et al., 2020) samples a small block of noise at some random locations of the image and add the noise with the image if it increases the confidence towards the adversarial region. There is another line of research that crafts sparse adversarial examples (Croce & Hein, 2019; Croce et al., 2022), focusing on attacks that limit the number of perturbed pixels to minimize detection. Nevertheless, SA offers SOTA performance

in crafting untargeted and targeted *top-1* adversarial examples satisfying $\ell_2$-norm constraint among the existing surrogate-free score-based attacks (Li et al., 2024).

***Top-K* white-box attacks.** Up to date, the vast majority of adversarial attacks in literature have been focused on the *top-1* setting, except for some pioneering white-box attacks. An ordered *top-K* white-box attack is proposed in (Zhang & Wu, 2020), which uses an adversarial distillation framework in crafting adversarial examples by minimizing the Kullback-Leibler divergence between the prediction probability distribution and the adversarial distribution. $T_kML$ (Hu et al., 2021) proposes a white-box method to create *top-K* untargeted and targeted adversarial perturbation for the multi-label learning problem. The DeepFool attack (Moosavi-Dezfooli et al., 2016) that was proposed for *top-1* adversarial examples is extended in (Tursynbek et al., 2022) to compute *top-K* untargeted adversarial examples. Moreover, (Paniagua et al., 2023) introduces a quadratic programming method to learn ordered *top-K* adversarial examples that addresses attack constraints within the feature embedding space. (Mahmood & Elhamifar, 2024) proposes a framework for crafting semantically consistent adversarial attacks on multi-label models using a consistent target set.

## 3 THREAT MODEL

Consider a classifier $P(\boldsymbol{x}) : [0,1]^{C_h \times W \times H} \rightarrow [0,1]^C$, where $C_h, W, H$ are the channel, width, height of an arbitrary input $\boldsymbol{x}$, and $C$ denotes the number of output classes. The classifier outputs prediction probabilities for all classes in response to a query. To be more precise, $P_c(\boldsymbol{x})$ represents the probability that $\boldsymbol{x}$ belongs to class $c$, with the constraint that $\sum_{c=1}^{C} P_c(\boldsymbol{x}) = 1$. For a given input $\boldsymbol{x}$, the set of *top-K* predicted labels by the classifier can be expressed as follows:

$$\hat{\mathcal{Y}}_K(\boldsymbol{x}) = \{[\arg\operatorname{sort}_{c\in[C]} P_c(\boldsymbol{x})]_i\}_{i=1}^K, \tag{1}$$

where $\arg\operatorname{sort}_{c\in[C]}$ returns indices of sorted elements in decreasing order of probability, and $[C] = \{1, 2, ..., C\}$ denotes the label set. For instance, $[\arg\operatorname{sort}_{c\in[C]} P_c(\boldsymbol{x})]_i$ contains the label index of the class with the $i^{th}$ highest prediction probability.

An adversary's objective is to generate an imperceptible adversarial example, without using surrogate models, from a benign input image $\boldsymbol{x}_s$ which is correctly classified by the classifier. While the true label set of $\boldsymbol{x}_s$ for the classic single-label multi-class classification problem is expressed as $\mathcal{C}_s = \hat{\mathcal{Y}}_1(\boldsymbol{x}_s)$, for *top-K* multi-label learning it is expressed as $\mathcal{C}_s = \hat{\mathcal{Y}}_K(\boldsymbol{x}_s)$. In a score-based attack, having the information of the prediction probabilities by querying the target classifier, the adversary aims to identify a unit direction $\hat{\boldsymbol{\Theta}}$ in which $\boldsymbol{x}_s$ is moved into the adversarial region with minimal perturbation. A query $\boldsymbol{x}_q = \boldsymbol{x}_s + \boldsymbol{r}(\hat{\boldsymbol{\Theta}})$ in the direction $\hat{\boldsymbol{\Theta}}$ is considered in the adversarial region if $\mathbb{1}(\boldsymbol{x}_q) = 1$, where $\boldsymbol{r}(\hat{\boldsymbol{\Theta}}) = \|\boldsymbol{x}_q - \boldsymbol{x}_s\|_2 \hat{\boldsymbol{\Theta}}$ represents the perturbation added in the direction $\hat{\boldsymbol{\Theta}}$. The indicator function $\mathbb{1}(\boldsymbol{x}_q)$ informs whether the query $\boldsymbol{x}_q$ falls within the adversarial region or not. For an *untargeted attack*, aiming to move the true label set outside the *top-K* predicted classes, the query success indicator function takes the following form:

$$\mathbb{1}(\boldsymbol{x}_q) = \begin{cases} 1, & \text{if } \mathcal{C}_s \not\subset \hat{\mathcal{Y}}_K(\boldsymbol{x}_q) \\ -1, & \text{otherwise.} \end{cases} \tag{2}$$

In contrast, a *targeted attack* seeks to replace the *top-K* predictions of the input $\boldsymbol{x}_s$ with a predefined set of $K$ target classes, $\mathcal{Y}_K^{(t)} \subset [C] \setminus \mathcal{C}_s$. Thus, the query success indicator function for a targeted attack is:

$$\mathbb{1}(\boldsymbol{x}_q) = \begin{cases} 1, & \text{if } \hat{\mathcal{Y}}_K(\boldsymbol{x}_q) = \mathcal{Y}_K^{(t)} \\ -1, & \text{otherwise.} \end{cases} \tag{3}$$

If $\hat{\boldsymbol{\Theta}}^*$ represents the optimal direction to obtain the desired adversarial image $\boldsymbol{x}_{adv} = \boldsymbol{x}_s + \boldsymbol{r}(\hat{\boldsymbol{\Theta}}^*)$, the optimization problem can be formulated as:

$$\hat{\boldsymbol{\Theta}}^* = \arg\min_{\hat{\boldsymbol{\Theta}}} \|\boldsymbol{r}(\hat{\boldsymbol{\Theta}})\|_2, \quad \text{s.t. } \mathbb{1}(\boldsymbol{x}_s + \boldsymbol{r}(\hat{\boldsymbol{\Theta}})) = 1. \tag{4}$$

## 4 OUR PROPOSED GSBA$^K$

The proposed GSBA$^K$, guided by the approximated gradient direction in crafting adversarial examples, involves three key steps, as depicted in Fig. 3: (a) Estimating the gradient in the non-adversarial region to approach the decision boundary iteratively to find a better initial boundary point; (b) Esti-

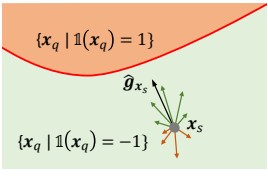 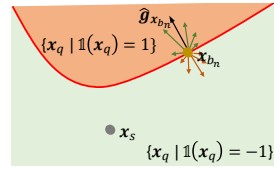 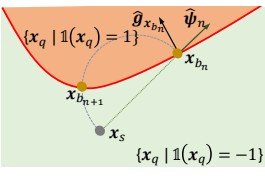

|         (a) Step 1         |         (b) Step 2         |         (c) Step 3         |

Figure 3: (a) Estimated gradient $\hat{g}_{x_s}$ on source image $x_s$ using Eq. 9 for targeted and Eq. 10 for untargeted attacks in the non-adversarial region; (b) Approximated gradient $\hat{g}_{x_{b_n}}$ at a decision boundary point $x_{b_n}$ using Eq. 12 for targeted and Eq. 13 for untargeted attacks; and (c) Subsequent boundary point $x_{b_{n+1}}$ search in the 2D plane spanned by $(\hat{g}_{x_{b_n}}, \hat{\psi}_n)$, where $\hat{\psi}_n$ denotes direction of $x_{b_n}$ from $x_s$. While the light green and the light orange regions indicate non-adversarial and adversarial regions, respectively, the dark green (dark orange) arrows indicate directions to increase (decrease) confidence toward the adversarial region.

mating the gradient at the boundary point by leveraging the prediction scores; (c) Finding the next boundary point with reduced perturbation along a semicircular trajectory under the guidance by the estimated gradient on decision boundary. **The key novelty of GSBA$^K$ is its ability to more accurately estimate gradients both within the adversarial region and at the decision boundary in the aggressive *top-K* setting**, enabling efficient boundary exploration to enhance ASR.

To estimate the gradient around a point $x$ in image space, we generate $I_n$ number of noise samples $\{z_i\}_{i=1}^{I_n}$ from the low-frequency Discrete Cosine Transformation (DCT) subspace and exploit the prediction probabilities of $x + z_i$; $\forall z_i \in \{z_i\}_{i=1}^{I_n}$ from the target classifier. The DCT subspace encapsulates critical information of an image, including the gradient information (Guo et al., 2018; Li et al., 2020a) (for more, please refer to Appendix F).

For a *top-K targeted attack* with the constraint of a target-label set $\mathcal{Y}_K^{(t)}$, to indicate the impact of an added $z_i$ to $x$ on a specific target class $c \in \mathcal{Y}_K^{(t)}$, we exploit the following information:
$$w_{x,c,z_i} := P_c(x + z_i) - P_c(x), \tag{5}$$
where a positive (or negative) value of $w_{x,c,z_i}$ indicates the increase (or decrease) in confidence towards the target class $c$ due to the added $z_i$ to $x$. For any query $x_q$, let $v_K = [\arg\operatorname{sort}_{j \in [C] \setminus \mathcal{C}_s} P_j(x_q)]_K$ denote the index of the class other than $\mathcal{C}_s$ with the $K$-th largest prediction probability. Additionally, let $c_s = \arg\max_{j \in \mathcal{C}_s} P_j(x_q)$ denote the source class with the highest prediction probability. For a *top-K untargeted attack*, where adversarial queries can take any set of *top-K* classes from $[C] \setminus \mathcal{C}_s$, the adversarial region becomes significantly broader. Thus, we exploit the following information for an untargeted attack:
$$F_{x_q} := P_{v_K}(x_q) - P_{c_s}(x_q), \tag{6}$$
where $\{x \mid F_x > 0\}$ indicates the adversarial region. With this a higher positive value of $F_{x_q}$ implies a more confident $x_q$ in the adversarial region than the non-adversarial counterpart.

## 4.1 GRADIENT ESTIMATION IN NON-ADVERSARIAL REGION

Finding a good initial boundary point $x_{b_0}$ is crucial for the success of a geometric attack. Unlike geometric decision-based attacks (Chen et al., 2020; Reza et al., 2023), which locate $x_{b_0}$ employing a random direction for the untargeted attack, and a binary search between the source image and a random target image for the targeted attack, our approach utilizes estimated gradient directions to find a better $x_{b_0}$. We estimate the gradient $g_{x_s}$ at $x_s$ by querying around it and iteratively shift the query sample in the gradient direction $\hat{g}_{x_s} := g_{x_s}/\|g_{x_s}\|_2$ using a fixed step size $\epsilon$ to locate $x_{b_0}$.

In **targeted attacks**, estimating the gradient inside the non-adversarial region at a point $x'_{b_0}$ is challenging as it involves finding the direction towards the narrow adversarial region constraint by the target-label set $\mathcal{Y}_K^{(t)}$. Querying within the non-adversarial region around $x'_{b_0}$ will result in all queries being non-adversarial, as in Fig. 3a. Since the adversarial objective is to continuously increase the confidence of the target classes, to estimate gradient at $x'_{b_0}$, the adversary may target the region $\{x'_{b_0} + z_i \mid (\min_{c \in \mathcal{Y}_K^{(t)}} P_c(x'_{b_0} + z_i) - \min_{c \in \mathcal{Y}_K^{(t)}} P_c(x'_{b_0})) > 0\}$ that ensures a gradient direction to enhance the minimum confidence among the target classes. Thus, we define the following indicator function to check whether a query around $x'_{b_0}$ satisfies the adversary's goal:

$$\phi_{\boldsymbol{x}'_{b_0}}(\boldsymbol{z}_i) := \begin{cases} 1, & \text{if } \min_{c \in \mathcal{Y}_K^{(t)}} P_c(\boldsymbol{x}'_{b_0} + \boldsymbol{z}_i) - \min_{c \in \mathcal{Y}_K^{(t)}} P_c(\boldsymbol{x}'_{b_0}) > 0 \\ -1, & \text{otherwise.} \end{cases} \tag{7}$$

However, all the queries that satisfy $\phi_{\boldsymbol{x}'_{b_0}}(\boldsymbol{z}_i) = 1$ are not equally effective. For instance, for a *top-3* targeted attack, consider the prediction probabilities with the target classes from the target classifier for $\boldsymbol{x}'_{b_0}$ as $\{0.12, 0.13, 0.1\}$. Assume, $\boldsymbol{z}_1$ changes these probabilities to $\{0.13, 0.14, 0.11\}$, while $\boldsymbol{z}_2$ modifies these to $\{0.11, 0.12, 0.11\}$. While both $\boldsymbol{z}_1$ and $\boldsymbol{z}_2$ enhance the minimum confidence among the target classes (i.e., $\phi_{\boldsymbol{x}'_{b_0}}(\boldsymbol{z}_i) = 1$ for both), $\boldsymbol{z}_1$ has a greater impact as it improves confidence across all target classes. To indicate the impact of the added $\boldsymbol{z}_i$ to $\boldsymbol{x}'_{b_0}$ on a particular target class $c \in \mathcal{Y}_K^{(t)}$, we introduce the following indicator function

$$\chi_{\boldsymbol{x}'_{b_0}, c, \boldsymbol{z}_i} = \begin{cases} 1, & \text{if } \phi_{\boldsymbol{x}'_{b_0}}(\boldsymbol{z}_i) w_{\boldsymbol{x}'_{b_0}, c, \boldsymbol{z}_i} > 0 \\ 0, & \text{otherwise,} \end{cases} \tag{8}$$

where $w_{\boldsymbol{x}'_{b_0}, c, \boldsymbol{z}_i}$ is defined in Eq. 5. Note that $\chi_{\boldsymbol{x}'_{b_0}, c, \boldsymbol{z}_i} = 1$ includes the query that satisfies $\phi_{\boldsymbol{x}'_{b_0}} = -1$ with a decreased confidence on $c$, *i.e.*, $w_{\boldsymbol{x}'_{b_0}, c, \boldsymbol{z}_i} < 0$, which will be included in gradient estimate, as experimentally it is observed that, with a very high probability, $\chi_{\boldsymbol{x}'_{b_0}, c, \boldsymbol{z}_i} = \chi_{\boldsymbol{x}'_{b_0}, c, -\boldsymbol{z}_i}$ satisfies. Thus, the gradient in the non-adversarial region for the *top-K* targeted attack can be estimated as:

$$\boldsymbol{g}_{\boldsymbol{x}'_{b_0}} = \sum_{i=1}^{I_0} \Big( \sum_{c \in \mathcal{Y}_K^{(t)}} \chi_{\boldsymbol{x}'_{b_0}, c, \boldsymbol{z}_i} \Big) \cdot \Big( \sum_{c \in \mathcal{Y}_K^{(t)}} w_{\boldsymbol{x}'_{b_0}, c, \boldsymbol{z}_i} \chi_{\boldsymbol{x}'_{b_0}, c, \boldsymbol{z}_i} \Big) \cdot \boldsymbol{z}_i, \tag{9}$$

where $\sum_{c \in \mathcal{Y}_K^{(t)}} \chi_{\boldsymbol{x}'_{b_0}, c, \boldsymbol{z}_i}$ counts the number of target classes with increased (or decreased) confidence when $\phi_{\boldsymbol{x}'_{b_0}}(\boldsymbol{z}_i) = 1$ (or $\phi_{\boldsymbol{x}'_{b_0}}(\boldsymbol{z}_i) = -1$), and $\sum_{c \in \mathcal{Y}_K^{(t)}} w_{\boldsymbol{x}'_{b_0}, c, \boldsymbol{z}_i} \chi_{\boldsymbol{x}'_{b_0}, c, \boldsymbol{z}_i}$ captures the change in confidence of these classes. Eq. 9 provides a gradient direction that effectively increases the confidence of the minimum prediction probability among $\mathcal{Y}_K^{(t)}$ and iteratively finds a boundary point $\boldsymbol{x}_{b_0}$. It is effective in finding $\boldsymbol{x}_{b_0}$ as it considers the impact of $\boldsymbol{z}_i$ on each of the target classes in $\mathcal{Y}_K^{(t)}$ and assigns weight to $\boldsymbol{z}_i$ accordingly. Additional analysis, showing the impact of each component in Eq. 9 and the rationale behind it, is provided in Appendix A.1.

Turning to **untargeted attacks**, querying around $\boldsymbol{x}'_{b_0}$ result in $F_{\boldsymbol{x}'_{b_0} + \boldsymbol{z}_i} < 0; \ \forall \boldsymbol{z}_i \in \{\boldsymbol{z}_i\}_{i=1}^{I_0}$ due to the reason discussed above. Thus, having a wider adversarial region, to estimate the gradient at $\boldsymbol{x}'_{b_0}$, we consider the impact of $\boldsymbol{z}_i$ on the enhancement of confidence towards the adversarial region relative to the non-adversarial counterpart. This gradient can be approximated as:

$$\boldsymbol{g}_{\boldsymbol{x}'_{b_0}} = \sum_{i=1}^{I_0} \Big( F_{\boldsymbol{x}'_{b_0} + \boldsymbol{z}_i} - F_{\boldsymbol{x}'_{b_0}} \Big) \cdot \boldsymbol{z}_i, \tag{10}$$

where $F_{\boldsymbol{x}'_{b_0} + \boldsymbol{z}_i} - F_{\boldsymbol{x}'_{b_0}} > 0$ signifies increased confidence towards the adversarial region in relative to the non-adversarial region, due to perturbed $\boldsymbol{x}'_{b_0} + \boldsymbol{z}_i$ compared to $\boldsymbol{x}'_{b_0}$, and vice versa. This strategy leverages the available prediction probabilities, enhancing the possibility of finding a good $\boldsymbol{x}_{b_0}$ for the untargeted attack such that $\hat{\mathcal{Y}}_K(\boldsymbol{x}_{b_0}) \in [C] \setminus \mathcal{C}_s$.

### 4.2 GRADIENT ESTIMATION ON DECISION BOUNDARY

Gradient estimation on the decision boundary plays a pivotal role in our proposed approach. For a **targeted attack**, the gradient is estimated at a boundary point $\boldsymbol{x}_{b_n}$ by querying around it in each iteration to find the next boundary point $\boldsymbol{x}_{b_{n+1}}$ with reduced perturbation. However, estimating the gradient for *top-K* targeted attacks with a narrow adversarial region is still complicated. Not all adversarial queries contribute equally to this goal. Some queries may behave anomalously, leading to a reduction in confidence across all target classes, while others may improve confidence for only a subset of the target classes. Ideally, the most effective queries are those that increase confidence across all the target classes. For instance, consider that the prediction probabilities of the target classes $\mathcal{Y}_3^{(t)}$, for a *top-3* attack, at $\boldsymbol{x}_{b_n}$ are $\{P_c(\boldsymbol{x}_{b_n})\}_{c \in \mathcal{Y}_3^{(t)}} = \{0.10, 0.08, 0.12\}$. After applying three random perturbations, the resulting prediction probabilities are $[\{P_c(\boldsymbol{x}_{b_n} + \boldsymbol{z}_i)\}_{c \in \mathcal{Y}_3^{(t)}}]_{i=1}^3 = [\{0.09, 0.07, 0.11\}, \{0.09, 0.07, 0.15\}, \{0.11, 0.09, 0.13\}]; \ \mathbb{1}(\boldsymbol{x}_{b_n} + \boldsymbol{z}_i) = 1, \forall i \in [3]$. In this example, the first perturbation is anomalous, as it reduces the overall confidence in the target classes. The second perturbation biases the result towards a particular class, while the third perturbation increases confidence in all target classes, making it the most effective for achieving the adversarial goal. Thus,

---

**Algorithm 1:** GSBA$^K$

---

1. **Input:** benign image $\boldsymbol{x}_s$, query budget $Q$, base query number $I_0$, step size $\epsilon$, tolerance $\tau$.
2. **Output:** adversarial image $\boldsymbol{x}_{adv}$.
3. $\boldsymbol{x}'_{b_0} = \boldsymbol{x}_s, Q' = 1$.
4. **while** $\mathbb{1}(\boldsymbol{x}'_{b_0}) = -1$ **do**
5. $\quad$ $\hat{\boldsymbol{g}}_{\boldsymbol{x}'_{b_0}} = \boldsymbol{g}_{\boldsymbol{x}'_{b_0}}/\|\boldsymbol{g}_{\boldsymbol{x}'_{b_0}}\|_2$ based on Eq. 9 and Eq. 10 for targeted and untargeted attacks, respectively, using $I_0$ queries.
6. $\quad$ $\boldsymbol{x}'_{b_0} = \boldsymbol{x}'_{b_0} + \epsilon * \hat{\boldsymbol{g}}_{\boldsymbol{x}'_{b_0}}, Q' = Q' + I_0 + 1$.
7. $\boldsymbol{x}_{b_0}, Q_{bin} \leftarrow \texttt{BinarySearch}(\boldsymbol{x}_s, \boldsymbol{x}'_{b_0}, \mathbb{1}(.), \tau)$.
8. $Q' = Q' + Q_{bin}, \quad n = 0$.
9. **while** $Q' \leq Q$ **do**
10. $\quad$ estimate $\hat{\boldsymbol{g}}_{\boldsymbol{x}_{b_n}} = \boldsymbol{g}_{\boldsymbol{x}_{b_n}}/\|\boldsymbol{g}_{\boldsymbol{x}_{b_n}}\|_2$ based on Eq. 12 for targeted and Eq. 13 for untargeted attacks using $\lfloor I_0\sqrt{n+1} \rfloor$ queries.
11. $\quad$ direction of the boundary point from the source image: $\hat{\boldsymbol{\psi}}_n = \frac{\boldsymbol{x}_{b_n} - \boldsymbol{x}_s}{\|\boldsymbol{x}_{b_n} - \boldsymbol{x}_s\|_2}$, as shown in Fig. 3c.
12. $\quad$ $\boldsymbol{x}_{b_{n+1}}, Q_{bs} \leftarrow$ next boundary point along a semi-circular path guided by $\hat{\boldsymbol{g}}_{\boldsymbol{x}_{b_n}}$ and $\hat{\boldsymbol{\psi}}_n$, and corresponding query cost, as discussed in CGBA (Reza et al., 2023).
13. $\quad$ $Q' = Q' + \lfloor I_0\sqrt{n+1} \rfloor + Q_{bs}, \quad n = n + 1$.
14. **return:** $\boldsymbol{x}_{adv} = \boldsymbol{x}_{b_n}$

---

we also propose an effective gradient estimation technique on a decision boundary point by leveraging the available information for the challenging *top-K* setting that filters out anomalous queries and assigns greater weight to perturbations that have a higher impact on achieving the adversarial goal. Now, we introduce an indicator function that assesses whether the query $\boldsymbol{x}_{b_n} + \boldsymbol{z}_i$ is adversarial and leads to increased confidence to the class $c \in \mathcal{Y}_K^{(t)}$:

$$\zeta_{\boldsymbol{x}_{b_n},c,\boldsymbol{z}_i} = \begin{cases} 1, & \text{if } \mathbb{1}(\boldsymbol{x}_{b_n} + \boldsymbol{z}_i)w_{\boldsymbol{x}_{b_n},c,\boldsymbol{z}_i} > 0 \\ 0, & \text{otherwise.} \end{cases} \tag{11}$$

Similar to Eq. 8, it also includes the non-adversarial queries ($\mathbb{1}(\boldsymbol{x}_{b_n} + \boldsymbol{z}_i) = -1$) while $w_{\boldsymbol{x}_{b_n},c,\boldsymbol{z}_i} < 0$, which will be incorporated in the gradient estimate by flipping the sign of $\boldsymbol{z}_i$. The gradient for the intricate *top-K* targeted attack is thus estimated as:

$$\boldsymbol{g}_{\boldsymbol{x}_{b_n}} = \sum_{i=1}^{I_n} \Big( \sum_{c \in \mathcal{Y}_K^{(t)}} \zeta_{\boldsymbol{x}_{b_n},c,\boldsymbol{z}_i} \Big) \cdot \Big( \sum_{c \in \mathcal{Y}_K^{(t)}} w_{\boldsymbol{x}_{b_n},c,\boldsymbol{z}_i}\zeta_{\boldsymbol{x}_{b_n},c,\boldsymbol{z}_i} \Big) \cdot \boldsymbol{z}_i. \tag{12}$$

Here, $\sum_{c \in \mathcal{Y}_K^{(t)}} \zeta_{\boldsymbol{x}_{b_n},c,\boldsymbol{z}_i}$ and $\sum_{c \in \mathcal{Y}_K^{(t)}} w_{\boldsymbol{x}_{b_n},c,\boldsymbol{z}_i}\zeta_{\boldsymbol{x}_{b_n},c,\boldsymbol{z}_i}$ are weights assigned to $\boldsymbol{z}_i$ to estimate the gradient. The former counts the number of target classes for which $\mathbb{1}(\boldsymbol{x}_{b_n} + \boldsymbol{z}_i)w_{\boldsymbol{x}_{b_n},c,\boldsymbol{z}_i} > 0$, while the latter captures the strength of increased (or decreased) confidence towards the adversarial region if the query is adversarial (or non-adversarial). Additionally, $\sum_{c \in \mathcal{Y}_K^{(t)}} \zeta_{\boldsymbol{x}_{b_n},c,\boldsymbol{z}_i} = 0$ indicates the query is anomalous. The proposed method offers improved gradient estimation by emphasizing the impact of the added noise $\boldsymbol{z}_i$ on each of the target classes comparing the prediction probabilities of the boundary point $\boldsymbol{x}_{b_n}$ and the perturbed $\boldsymbol{x}_{b_n} + \boldsymbol{z}_i$. It assigns more weight to $\boldsymbol{z}_i$ in estimating the gradient, if $\boldsymbol{z}_i$ impacts more target classes in $\mathcal{Y}_K^{(t)}$, while filtering out anomalous queries that do not correctly contribute to the adversarial goal. For a more detailed discussion of the rationale behind Eq. 12, including the impact of each component in it on gradient estimation and comparisons with other possible choices of gradient estimation, please refer to Appendix A.2.

In an **untargeted attack**, since the attack transitions from a small set of source classes to any *top-K* classes in $[C] \backslash \mathcal{C}_s$, the likelihood of anomalous queries is greatly reduced. In this scenario, it suffices to evaluate the impact of the added noise $\boldsymbol{z}_i$ based on how effectively it enhances confidence toward the adversarial region compared to its non-adversarial counterpart. Under this setting, the gradient can be can be straightforwardly approximated as:

$$\boldsymbol{g}_{\boldsymbol{x}_{b_n}} = \sum_{i=1}^{I_n} F_{\boldsymbol{x}_{b_n}+\boldsymbol{z}_i} \cdot \boldsymbol{z}_i. \tag{13}$$

The larger the positive value of $F_{\boldsymbol{x}_{b_n}+\boldsymbol{z}_i}$, the greater the confidence shift towards the adversarial region relative to the non-adversarial one, and more weight is assigned to $\boldsymbol{z}_i$ if the change is higher. Note, Eq. 13 also incorporates $\boldsymbol{z}_i$, which results in non-adversarial queries ($F_{\boldsymbol{x}_{b_n}+\boldsymbol{z}_i} < 0$) and

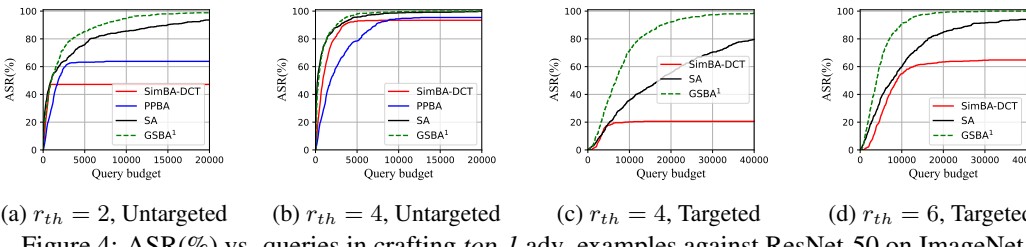

| (a) $r_{th} = 2$, Untargeted | (b) $r_{th} = 4$, Untargeted | (c) $r_{th} = 4$, Targeted | (d) $r_{th} = 6$, Targeted |

Figure 4: ASR(%) vs. queries in crafting *top-1* adv. examples against ResNet-50 on ImageNet.

weights them accordingly, as it is highly probable that $F_{\boldsymbol{x}_{b_n}+\boldsymbol{z}_i} < 0 \implies F_{\boldsymbol{x}_{b_n}-\boldsymbol{z}_i} > 0$. The impact of non-adversarial queries on attack performance is discussed in Appendix A.3.

The steps of GSBA$^K$ are outlined in Algorithm 1. Line 4-Line 6 are used to iteratively shift $\boldsymbol{x}_s$ into the adversarial region with a step size $\epsilon$. Then, a binary search between the obtained point inside the adversarial region and $\boldsymbol{x}_s$ is conducted to locate the initial boundary point $\boldsymbol{x}_{b_0}$ within a certain tolerance $\tau$. From Line 9-Line 13, GSBA$^K$ iteratively finds boundary point with reduced perturbation using the similar semicircular boundary as CGBA (Reza et al., 2023) guided by the proposed estimated gradient $\hat{\boldsymbol{g}}_{\boldsymbol{x}_{b_n}}$ on the decision boundary for both untargeted and targeted attacks. The code of our attack is available at https://github.com/Farhamdur/GSBA-K.

## 5 EXPERIMENTS

In this section, we first outline the baselines, evaluation metrics, and hyperparameters employed for both the baselines and GSBA$^K$. Subsequently, we present the experimental results on ImageNet, which contains 1,000 classes, and the PASCAL VOC 2012 dataset, which includes 20 classes, illustrating the efficacy of GSBA$^K$ in executing *top-K* attacks. The limitations and potential negative impacts of GSBA$^K$ are addressed in the supplementary material.

**Baselines, evaluation metrics and hyperparameters.** To evaluate the performance of GSBA$^K$, we choose score-based attacks SimBA-DCT (Guo et al., 2019a), PPBA (Li et al., 2020b) and SA (Andriushchenko et al., 2020) as baselines. To the best of our knowledge, SA offers SOTA performance in score-based setting. The baselines are only designed to craft *top-1* adversarial examples against classifiers with single-label multi-class classification. While SA and SimBA-DCT can generate both untargeted and targeted adversarial examples, PPBA is restricted to untargeted attacks. Nonetheless, the proposed GSBA$^K$ is versatile in crafting *top-K* adversarial examples for both untargeted and targeted attacks, and both single-label and multi-label learning. To have a strong baseline in the *top-K* setting, we adapt the loss function of SA (Andriushchenko et al., 2020) for both untargeted and targeted attacks, enabling its application in crafting *top-K* adversarial examples, coined as SA$^K$. For untargeted attacks, we modify the loss function in (Andriushchenko et al., 2020) to $L(f(\boldsymbol{x}_q), c_s) = f_{c_s}(\boldsymbol{x}_q) - f_{v_K}(\boldsymbol{x}_q)$, where $c_s$ and $v_K$ are defined in Sec. 4. Likewise, for targeted attacks, the loss function is modified to $L(f(\boldsymbol{x}_q), \mathcal{Y}_K^{(t)}) = -\min_{j \in \mathcal{Y}_K^{(t)}} f_j(\boldsymbol{x}_q) + \max_{j \in [C] \setminus \mathcal{Y}_K^{(t)}} f_j(\boldsymbol{x}_q)$ for $K > 1$. This adaptation enables the effective crafting of *top-K* adversarial examples using SA$^K$.

We evaluate our method primarily using the metric Attack Success Rate (ASR). An attack is deemed successful if it crafts an adversarial example below a specified $\ell_2$-norm perturbation threshold, $r_{th}$, using queries within the allocated budget, $Q$. The lower the $r_{th}$, the more queries are required to make an attack successful, and vice versa. We also assess the effectiveness of an attack using the median $\ell_2$-norm of the perturbation. A lower median perturbation value across all crafted adversarial examples within the allocated query budget indicates greater attack effectiveness.

Baseline implementations leverage the codes provided by the respective authors, with some suitable parameter adjustments. While SA accounts for constraints both in $Q$ and $r_{th}$ in defining attack success, SimBA-DCT and PPBA only consider the constraint in $Q$, potentially giving a false sense of success. Empirically, it is observed that the crafted perturbations in the default setting of SimBA-DCT and PPBA are often large and suffer from low ASR for a small $r_{th}$. To address this, we set $\epsilon = 0.1$ for SimBA-DCT and $\rho = 0.001$ for PPBA, where $\epsilon$ and $\rho$ control the magnitude of noises added at each query step for the respective methods, while retaining other parameters at their default settings. For SOTA SA, we use the default parameters. In the case of GSBA$^K$, we use reduced-dimensional frequency subspace with a dimension reduction factor $f = 4$ to sample low-frequency noise $\{\boldsymbol{z}_i\}$. We set the base query number $I_0 = 30$, step size $\epsilon = 6$, and tolerance $\tau = 0.0001$.

Table 1: ASR(%) for different perturbation thresholds ($r_{th}$) and query budgets ($Q$) against ResNet-50 on ImageNet.

| Attack Type | | | Untargeted Attack | | | | | | Targeted Attack | | | | |
|---|---|---|---|---|---|---|---|---|---|---|---|---|---|
| | $Q$ | | 1000 | 5000 | 10000 | 20000 | 30000 | | 5000 | 10000 | 20000 | 30000 | 40000 |
| top-1 | $SA^1$ | $r_{th}=1$ | 26.5 | 50.2 | 58.1 | 71.2 | 76.1 | $r_{th}=2$ | 3.2 | 8.1 | 16.2 | 22.0 | 29.6 |
| | $GSBA^1$ | | 29.2 | 61.4 | 76.8 | 84.5 | 87.9 | | 8.4 | 23.4 | 47.5 | 62.4 | 69.9 |
| | $SA^1$ | $r_{th}=2$ | 50.7 | 75.8 | 85.6 | 93.6 | 96.3 | $r_{th}=4$ | 19.3 | 35.5 | 55.5 | 70.4 | 79.6 |
| | $GSBA^1$ | | 50.4 | 85.3 | 94.6 | 98.7 | 99.3 | | 36.3 | 71.6 | 92.2 | 97.3 | 98.6 |
| | $SA^1$ | $r_{th}=4$ | 75.2 | 95.6 | 98.8 | 99.7 | 99.8 | $r_{th}=6$ | 37.4 | 60.1 | 84.5 | 91.3 | 94.0 |
| | $GSBA^1$ | | 74.9 | 98.0 | 99.6 | 100.0 | 100.0 | | 63.2 | 90.5 | 99.2 | 99.9 | 100.0 |
| top-2 | $SA^2$ | $r_{th}=1$ | 10.3 | 28.9 | 41.2 | 48.7 | 55.4 | $r_{th}=2$ | 0.9 | 3.1 | 6.6 | 12.9 | 16.0 |
| | $GSBA^2$ | | 12.8 | 40.5 | 55.5 | 69.7 | 74.8 | | 2.5 | 9.8 | 26.8 | 39.1 | 46.8 |
| | $SA^2$ | $r_{th}=2$ | 30.7 | 56.1 | 69.1 | 78.8 | 85.0 | $r_{th}=4$ | 7.6 | 19.1 | 32.2 | 44.8 | 52.4 |
| | $GSBA^2$ | | 25.6 | 68.7 | 83.7 | 94.9 | 97.2 | | 16.6 | 45.1 | 75.6 | 87.6 | 92.0 |
| | $SA^2$ | $r_{th}=4$ | 55.4 | 84.1 | 93.8 | 98.0 | 98.9 | $r_{th}=6$ | 19.9 | 36.6 | 57.5 | 67.4 | 73.0 |
| | $GSBA^2$ | | 50.5 | 89.2 | 97.2 | 99.5 | 100.0 | | 36.7 | 71.6 | 92.4 | 97.1 | 98.2 |
| top-3 | $SA^3$ | $r_{th}=1$ | 5.2 | 17.4 | 27.9 | 37.0 | 44.5 | $r_{th}=2$ | 0.8 | 1.1 | 2.4 | 4.0 | 5.1 |
| | $GSBA^3$ | | 5.7 | 30.5 | 47.1 | 63.4 | 67.8 | | 0.9 | 4.8 | 14.8 | 26.5 | 33.8 |
| | $SA^3$ | $r_{th}=2$ | 19.8 | 45.9 | 56.8 | 68.3 | 72.3 | $r_{th}=4$ | 3.7 | 9.3 | 19.3 | 27.0 | 31.8 |
| | $GSBA^3$ | | 17.3 | 62.6 | 77.3 | 90.2 | 93.9 | | 8.2 | 27.3 | 55.5 | 71.2 | 78.8 |
| | $SA^3$ | $r_{th}=4$ | 46.8 | 77.7 | 88.5 | 94.8 | 97.4 | $r_{th}=6$ | 10.1 | 22.3 | 33.2 | 44.6 | 51.1 |
| | $GSBA^3$ | | 43.1 | 84.8 | 95.0 | 99.1 | 99.6 | | 21.9 | 49.2 | 77.8 | 88.7 | 92.8 |
| top-4 | $SA^4$ | $r_{th}=1$ | 2.5 | 12.6 | 18.6 | 26.9 | 30.0 | $r_{th}=2$ | 0.5 | 0.8 | 1.2 | 1.6 | 2.9 |
| | $GSBA^4$ | | 3.6 | 24.7 | 40.5 | 56.8 | 61.3 | | 0.4 | 1.5 | 9.9 | 15.7 | 22.0 |
| | $SA^4$ | $r_{th}=2$ | 14.3 | 37.4 | 50.5 | 62.9 | 66.8 | $r_{th}=4$ | 1.4 | 3.1 | 11.5 | 16.6 | 19.7 |
| | $GSBA^4$ | | 12.4 | 55.6 | 72.3 | 85.3 | 92.6 | | 4.3 | 15.0 | 39.2 | 55.7 | 63.4 |
| | $SA^4$ | $r_{th}=4$ | 41.3 | 72.6 | 82.1 | 92.2 | 95.1 | $r_{th}=6$ | 5.0 | 10.9 | 22.2 | 29.0 | 33.9 |
| | $GSBA^4$ | | 34.2 | 78.6 | 93.7 | 98.9 | 99.4 | | 10.1 | 35.0 | 61.5 | 74.4 | 81.4 |

## 5.1 RESULTS AGAINST SINGLE-LABEL MULTI-CLASS CLASSIFICATION

We assess $GSBA^K$ by conducting attacks against three widely used pre-trained classifiers—ResNet-50 (He et al., 2016), ResNet-101 (He et al., 2016), and VGG-16 (Simonyan & Zisserman, 2014) trained for multi-class classification on the ImageNet (Deng et al., 2009) dataset. The pre-trained ResNet-50, ResNet-101 and VGG-16 models are sourced from PyTorch. In the case of untargeted attacks against a classifier on ImageNet, we randomly select 1000 images that are correctly classified by the respective classifier. For targeted attacks, we create 1000 sets of images, each comprising a benign image $x_s$ and a target image $x_t$. The *top-K* target labels are extracted from the target image $\hat{\mathcal{Y}}_K(x_t)$. The input image size for all classifiers on the ImageNet dataset is set as $3 \times 224 \times 224$.

**Comparison with *top*-1 baselines.** We compare the performance of the proposed $GSBA^1$ with SimBA-DCT (Guo et al., 2019a), PPBA (Li et al., 2020b) and SA (Andriushchenko et al., 2020) in crafting *top-1* adversarial examples against classifiers with multi-class classification. Fig. 4 depicts the variation of ASR against ResNet-50 on ImageNet with differing query budgets across various $r_{th}$ values. From Fig. 4, for untargeted attacks, the ASR of SA is comparable with $GSBA^1$ for $r_{th} = 4$ and reaches around 100% ASR. However, $GSBA^1$ offers better ASR than SA with reduced $r_{th}$. Experimentally, it is observed that with a higher value of $r_{th}$ than 4, both SA and $GSBA^1$ converge faster towards the 100% ASR. Conversely, for targeted attacks, $GSBA^1$ outperforms the baseline by a considerable margin in ASR. Additional results for other classifiers are provided in Appendix B.

**Results on crafting *top-K* adversarial examples.** The ASR in crafting up to *top-4* adversarial examples for both untargeted and targeted attacks against the ResNet-50 classifier on the ImageNet dataset is presented in Table 1, considering different query budgets and perturbation threshold $r_{th}$. The corresponding curves for ResNet-50, with $r_{th} = 2$ for untargeted attacks and $r_{th} = 4$ for targeted attacks, are depicted in Fig. 5. The obtained ASR for different $r_{th}$ and query budgets against other regular classifiers are given in Appendix C. Additional

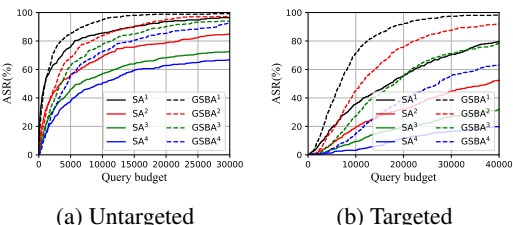

(a) Untargeted  (b) Targeted

Figure 5: ASR(%) versus queries for the attack against ResNet-50 on ImageNet.

evaluations against a number of robust classifiers are provided in Appendix D. We choose $SA^K$ for the performance comparison, as it offers SOTA performance in the *top-1* setting.

In the context of untargeted attacks, it is noteworthy that while the ASR is comparable to $SA^K$ when the query budget is relatively low, the proposed $GSBA^K$ method notably outperforms $SA^K$ as the query budget increases. Furthermore, for a given query budget and fixed $K$, the relative ASR of $GSBA^K$ compared to $SA^K$ increases as $r_{th}$ decreases. Additionally, with a fixed query budget and $r_{th}$, the relative ASR of $GSBA^K$ compared to $SA^K$ improves as $K$ increases. For the targeted attack, by contrast, $GSBA^K$ demonstrates significantly higher ASR compared to $SA^K$ across the board,

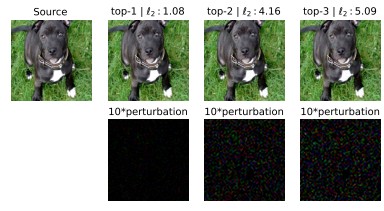
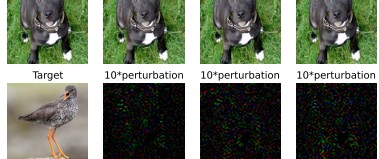

(a) Untargeted; Query budget = 2000.  (b) Targeted; Query budget = 20000.

Figure 6: Different *top-K* adversarial examples against ResNet50 for a benign input.

Table 2: ASR(%) against Inception-V3 with *top-2* multi-label learning on the PASCAL VOC 2012 dataset.

| Target type | | | Best | | | | Random | | | | Worst | | | |
|---|---|---|---|---|---|---|---|---|---|---|---|---|---|---|
| | $r_{th}$ | Q | 10000 | 20000 | 30000 | 40000 | 10000 | 20000 | 30000 | 40000 | 10000 | 20000 | 30000 | 40000 |
| *top-1* | 4 | SA[1] | 79.0 | 89.0 | 92.0 | 94.0 | 32.0 | 42.0 | 49.0 | 58.0 | 10.0 | 17.0 | 22.0 | 28.0 |
| | | GSBA[1] | 81.0 | 92.0 | 94.0 | 96.0 | 41.0 | 57.0 | 70.0 | 75.0 | 15.0 | 42.0 | 54.0 | 63.0 |
| | 6 | SA[1] | 91.0 | 94.0 | 96.0 | 96.0 | 47.0 | 60.0 | 68.0 | 73.0 | 20.0 | 31.0 | 39.0 | 46.0 |
| | | GSBA[1] | 89.0 | 97.0 | 99.0 | 99.0 | 56.0 | 73.0 | 77.0 | 84.0 | 34.0 | 59.0 | 72.0 | 78.0 |
| | 8 | SA[1] | 94.0 | 98.0 | 100.0 | 100.0 | 52.0 | 67.0 | 76.0 | 79.0 | 27.0 | 39.0 | 51.0 | 59.0 |
| | | GSBA[1] | 94.0 | 98.0 | 99.0 | 99.0 | 62.0 | 76.0 | 84.0 | 85.0 | 45.0 | 68.0 | 78.0 | 80.0 |
| *top-2* | 4 | SA[2] | 32.0 | 51.0 | 58.0 | 66.0 | 6.0 | 9.0 | 12.0 | 15.0 | 0.0 | 0.0 | 1.0 | 3.0 |
| | | GSBA[2] | 40.0 | 65.0 | 78.0 | 83.0 | 9.0 | 28.0 | 48.0 | 53.0 | 5.0 | 26.0 | 46.0 | 53.0 |
| | 6 | SA[2] | 51.0 | 73.0 | 77.0 | 82.0 | 11.0 | 23.0 | 27.0 | 32.0 | 2.0 | 8.0 | 10.0 | 11.0 |
| | | GSBA[2] | 53.0 | 80.0 | 83.0 | 89.0 | 24.0 | 44.0 | 53.0 | 62.0 | 17.0 | 50.0 | 57.0 | 67.0 |
| | 8 | SA[2] | 67.0 | 80.0 | 84.0 | 87.0 | 15.0 | 25.0 | 41.0 | 46.0 | 7.0 | 14.0 | 19.0 | 21.0 |
| | | GSBA[2] | 64.0 | 82.0 | 89.0 | 93.0 | 31.0 | 55.0 | 62.0 | 69.0 | 29.0 | 54.0 | 65.0 | 75.0 |

specifically in crafting adversarial examples with reduced perturbation. Moreover, GSBA[K] achieves substantially higher ASR than SA[K] in crafting *top-K* targeted adversarial examples with higher $K$. The crafted adversarial examples and their corresponding perturbation with different $K$ for a benign input are depicted in Fig. 6. For additional *top-K* adversarial examples and detailed insights, please refer to Appendix H.

## 5.2 ATTACK AGAINST MULTI-LABEL LEARNING

We employ the proposed GSBA[K] to attack against *top-K* multi-label learning, a task aimed at identifying the top $K$ prediction labels for a given input. To execute an attack against a target model with *top-K* multi-label learning, we use Inception-V3 (Szegedy et al., 2016), obtained from the GitHub repository of (Hu et al., 2021). This model is pre-trained on ImageNet (Deng et al., 2009) and fine-tuned on the PASCAL VOC 2012 dataset (Everingham et al., 2015). We focus on targeted *top-K* attacks, and categorize the possible target sets into three types: best, random, and worst. For a benign input, the *best target labels* refer to a set of $K$ labels, excluding true labels, with the highest prediction scores; conversely, the *worst target labels* denote those with the lowest prediction scores, and *Random target labels* represent a set of $K$ randomly selected mutually exclusive labels, excluding true labels. To perform the *top-K* attacks, we use 100 benign samples with $K$ true labels from the PASCAL VOC 2012 validation set and perform attacks considering the aforementioned three categories of target sets. The input images are resized to $3 \times 300 \times 300$ dimensions before being fed into the target model.

Table 2 demonstrates the ASR in crafting *top-1* and *top-2* adversarial examples for different query budgets and perturbation thresholds, considering the best, random and the worst target label sets. As seen from these results, GSBA[K] outperforms SA[K] across the board. Specifically, with the consideration of random and worst target labels, GSBA[K] notably outperforms SA[K]. Importantly, while SA[2] fails to converge to the desired perturbation for the worst target labels within the given query budgets, GSBA[2] maintains a significantly higher ASR. From Table 1 and Table 2, ASR decreases as $K$ increases. The impact of higher $K$ on ASR is detailed in Appendix E, while the effect of varying reduced-dimensional frequency subspaces for sampling $z_i$ explored in Appendix F.

## 6 CONCLUSION

In this work, we propose a geometric score-based attack, GSBA[K], to effectively generate strong *top-K* untargeted and targeted adversarial examples against classifiers with both single-label multi-class classification and multi-label learning tasks. It introduces novel gradient estimation techniques for the challenging *top-K* setting, efficiently finding the initial boundary point and effectively exploiting the decision boundary to iteratively refine the adversarial example by leveraging the estimated gradient direction. Experiments on large-scale benchmark datasets demonstrate that GSBA[K] offers state-of-the-art performance and would be a strong baseline in crafting *top-K* adversarial examples.

## ACKNOWLEDGEMENTS

This work was supported in part by the US National Science Foundation under grants CNS-1824518 and ECCS-2203214. T. Wu is partly supported by ARO Grant W911NF1810295, NSF IIS-1909644, ARO Grant W911NF2210010, NSF IIS-1822477, NSF CMMI-2024688 and NSF IUSE-2013451. R. Jin is supported by the National Key R&D Program of China under Grant No. 2024YFE0200804. The views and conclusions expressed in this work are those of the authors and do not necessarily reflect the official policies or endorsements, either expressed or implied, of ARO, NSF, or the U.S. Government. The U.S. Government is authorized to reproduce and distribute reprints for Governmental purposes not withstanding any copyright annotation thereon. The authors sincerely appreciate the constructive feedback provided by the anonymous reviewers and area chairs.

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

# Appendix

In this supplementary material, we provide the rationale for the design of proposed gradient estimation techniques for complicated *top-K* scenarios in Appendix A. Additionally, we conduct an ablation study to assess the impact of non-adversarial queries on gradient estimation, and the influence of gradient-based initialization on performance, which are also discussed in Appendix A. The additional results showing the comparison of the proposed GSBA[1] with *top-1* baselines are given in Appendix B, while the performance comparison between GSBA[K] and SA[K] against ResNet-101 (He et al., 2016) and VGG-16 (Simonyan & Zisserman, 2014), and several robust single-label multi-class classifiers can be found in Appendix C and Appendix D, respectively. The impact of larger $K$ on the performance of GSBA[K] is reported in Appendix E. A brief discussion of the noise sampling process from a low-dimensional frequency space is presented in Appendix F, while potential negative impacts and limitations of GSBA[K] are covered in Appendix G. Finally, a number of crafted adversarial examples against single-label multi-class classification and multi-label learning are depicted in Appendix H.

## A    ABLATION STUDY

In this section, we present an ablation study to analyze the design of the proposed gradient estimation techniques presented in Eq. 9 and Eq. 12, which estimate gradients within the adversarial regions and on the decision boundary, respectively, for the *top-K* targeted attack. We also examine the influence of non-adversarial queries on gradient estimation at the decision boundary and how this influences the performance of the proposed attack. Furthermore, we demonstrate that our gradient-based initial boundary-finding method significantly improves attack performance when compared to random boundary point initialization.

### A.1    DETAILED BREAKDOWN OF EQ. 9

Finding a good initial boundary point $\boldsymbol{x}_{b_0}$ for the targeted attack is a difficult task as it requires satisfying the constraint $\hat{\mathcal{Y}}_K(\boldsymbol{x}_{b_0}) = \mathcal{Y}_K^{(t)}$. In this section, we analyze how each component in Eq. 9 contributes to gradient estimation at a point $\boldsymbol{x}_{b_0}'$ inside the non-adversarial region to locate $\boldsymbol{x}_{b_0}$ for the targeted attack. To evaluate the effectiveness of the proposed gradient estimation, we also compare it with alternative possible methods for estimating the gradient at $\boldsymbol{x}_{b_0}'$, providing justification for the selection of our proposed approach.

**Method-1:**    The gradient at $\boldsymbol{x}_{b_0}'$ can be estimated by comparing the prediction probabilities for $\boldsymbol{x}_{b_0}'$ with those of the perturbed queries around it. The gradient estimation is give by:

$$\boldsymbol{g}_{\boldsymbol{x}_{b_0}'} = \sum_{i=1}^{I_0} \Big( \min_{j \in \mathcal{Y}_K^{(t)}} P_j(\boldsymbol{x}_{b_0}' + \boldsymbol{z}_i) - \min_{j \in \mathcal{Y}_K^{(t)}} P_j(\boldsymbol{x}_{b_0}') \Big) \cdot \boldsymbol{z}_i. \tag{14}$$

In this method, we estimate $\hat{\boldsymbol{g}}_{\boldsymbol{x}_{b_0}'}$ by comparing the minimum prediction probabilities of the target classes at $\boldsymbol{x}_{b_0}'$ with the minimum of those due to the added noise. The positive difference of $\min_{j \in \mathcal{Y}_K^{(t)}} P_j(\boldsymbol{x}_{b_0}' + \boldsymbol{z}_i) - \min_{j \in \mathcal{Y}_K^{(t)}} P_j(\boldsymbol{x}_{b_0}')$ ensures the increased confidence of the target class with minimum confidence.

**Method-2:**    Inspired by the CW (Carlini & Wagner, 2017) adversarial objective, another possible approach could be as follows:

$$\boldsymbol{g}_{\boldsymbol{x}_{b_0}'} = \sum_{i=1}^{I_0} \Big( L(\boldsymbol{x}_{b_0}' + \boldsymbol{z}_i) - L(\boldsymbol{x}_{b_0}') \Big) \cdot \boldsymbol{z}_i, \tag{15}$$

where $L(\boldsymbol{x}_{b_0}') = \min_{c \in \mathcal{Y}_K^{(t)}} P_c(\boldsymbol{x}_{b_0}') - \max_{c \notin \mathcal{Y}_K^{(t)}} P_c(\boldsymbol{x}_{b_0}')$. This formulation leverages the difference between the minimum probability among the target classes and the maximum probability of non-target classes to calculate the gradient.

**Method-3:**    This method is solely based on only considering the impact of the first weight factor, $\sum_{c \in \mathcal{Y}_K^{(t)}} \chi_{\boldsymbol{x}_{b_0}', c, \boldsymbol{z}_i}$, in Eq. 9, which counts the number of target classes with increased confidence when $\phi_{\boldsymbol{x}_{b_0}'}(\boldsymbol{z}_i) = 1$, or the number of target classes with decreased confidence when $\phi_{\boldsymbol{x}_{b_0}'}(\boldsymbol{z}_i) =$

Table 3: Caparison of different gradient estimations with the proposed gradient in Eq. 9 to locate the initial boundary for *top-K* targeted attacks.

| | Grad Estimator | $\ell_2$ (median) | $Q$ (median) | $\ell_2$ (mean) | $Q$ (mean) |
|---|---|---|---|---|---|
| *top-1* | Method-1 | **9.82** | **1802.50** | **10.01** | **2131.12** |
| | Method-3 | 11.30 | 2439.00 | 11.46 | 2811.12 |
| | Method-4 | **9.82** | **1802.50** | **10.01** | **2131.12** |
| | Proposed | **9.82** | **1802.50** | **10.01** | **2131.12** |
| *top-2* | Method-1 | 12.17 | 2950.50 | 12.64 | 3439.31 |
| | Method-3 | 12.74 | 3074.50 | 13.14 | 3693.56 |
| | Method-4 | **11.98** | **2733.50** | 12.59 | 3429.07 |
| | Proposed | 12.12 | 2750.50 | **12.57** | **3352.49** |
| *top-3* | Method-1 | 14.06 | 4258.00 | 14.83 | 4920.10 |
| | Method-3 | 14.12 | 4159.50 | 14.76 | 4899.33 |
| | Method-4 | 14.01 | 4175.00 | 14.71 | 4753.59 |
| | Proposed | **13.70** | **3896.00** | **14.30** | **4452.22** |
| *top-4* | Method-1 | 17.20 | 5818.00 | 26.99 | 6862.56 |
| | Method-3 | 16.10 | 5151.50 | 16.77 | 5995.36 |
| | Method-4 | 16.60 | 5322.00 | 17.28 | 6505.02 |
| | Proposed | **15.24** | **4640.00** | **16.17** | **5666.69** |
| *top-5* | Method-1 | 19.68 | 7368.00 | 20.27 | 9153.95 |
| | Method-3 | 17.39 | 6097.00 | 18.77 | 7305.30 |
| | Method-4 | 18.06 | 6779.00 | 19.12 | 8364.68 |
| | Proposed | **17.14** | **5709.50** | **18.03** | **7042.50** |

$-1$. Using only this factor, the gradient is estimated as:

$$g_{\boldsymbol{x}'_{b_0}} = \sum_{i=1}^{I_0} \Big( \sum_{c \in \mathcal{Y}_K^{(t)}} \chi_{\boldsymbol{x}'_{b_0}, c, \boldsymbol{z}_i} \Big) \cdot \phi_{\boldsymbol{x}'_{b_0}}(\boldsymbol{z}_i) \cdot \boldsymbol{z}_i, \tag{16}$$

where the indicator function $\phi_{\boldsymbol{x}'_{b_0}}(\boldsymbol{z}_i)$ is defined in Eq. 7.

**Method-4:** This method focuses on the second weight factor $\sum_{c \in \mathcal{Y}_K^{(t)}} w_{\boldsymbol{x}'_{b_0}, c, \boldsymbol{z}_i} \chi_{\boldsymbol{x}'_{b_0}, c, \boldsymbol{z}_i}$, in Eq. 9 to estimate the gradient at $\boldsymbol{x}'_{b_0}$. It computes the weight associated with $\boldsymbol{z}_i$ by summing the changes in prediction probabilities of the classes $c \in \mathcal{Y}_K^{(t)}$ either with increased confidence when $\phi_{\boldsymbol{x}'_{b_0}}(\boldsymbol{z}_i) = 1$ or with decreased in confidence when $\phi_{\boldsymbol{x}'_{b_0}}(\boldsymbol{z}_i) = -1$. The gradient estimation using Method-4 is given as follows:

$$g_{\boldsymbol{x}'_{b_0}} = \sum_{i=1}^{I_0} \Big( \sum_{c \in \mathcal{Y}_K^{(t)}} w_{\boldsymbol{x}'_{b_0}, c, \boldsymbol{z}_i} \cdot \chi_{\boldsymbol{x}'_{b_0}, c, \boldsymbol{z}_i} \Big) \cdot \boldsymbol{z}_i. \tag{17}$$

**Comparative analysis.** We compare the performance of the aforementioned gradient estimation methods with our proposed gradient estimation to show its effectiveness in finding the initial boundary $\boldsymbol{x}_{b_0}$. To conduct the comparison, we randomly selected 100 test samples from the ImageNet (Deng et al., 2009) dataset and measured the median and mean query counts required to locate $\boldsymbol{x}_{b_0}$ as well as the median and mean $\ell_2$-distance of obtained boundary points from the corresponding source images by performing attacks against ResNet50 (He et al., 2016). The results are demonstrated in Table 3. We exclude Method-2 which is based on CW adversarial objective, as experimentally it is observed that it often fails to find $\boldsymbol{x}_{b_0}$ in aggressive *top-K* setting. This is because $L(\boldsymbol{x}'_{b_0} + \boldsymbol{z}_i) - L(\boldsymbol{x}'_{b_0}) > 0$ doesn't mean that it increases the confidence towards the adversarial reason, and vice versa. One possible reason among numerous reasons of failure of Method-2 could be, because of the added $\boldsymbol{z}_i$ with $\boldsymbol{x}'_{b_0}$, while $\min_{c \in \mathcal{Y}_K^{(t)}} P_c(\boldsymbol{x}'_{b_0} + \boldsymbol{z}_i) \approx \min_{c \in \mathcal{Y}_K^{(t)}} P_c(\boldsymbol{x}'_{b_0})$, $\max_{c \notin \mathcal{Y}_K^{(t)}} P_c(\boldsymbol{x}'_{b_0} + \boldsymbol{z}_i) < \max_{c \notin \mathcal{Y}_K^{(t)}} P_c(\boldsymbol{x}'_{b_0})$ along with an enhanced confidence among classes $c \in [C] \setminus \mathcal{Y}_K^{(t)}$ that doesn't meet the goal to find the direction to enhance the confidence towards the adversarial region.

In the *top-1* setting, Method-1, Method-4 and the proposed gradient estimation offer the same strong performance in locating the initial boundary, as these methods converge to the same expression in this setting. However, from Table 3, the efficiency of Method-1 diminishes as $K$ increases. The performance deterioration is because Method-1 compares $\min_{c \in \mathcal{Y}_K^{(t)}} P_c(\boldsymbol{x}'_{b_0} + \boldsymbol{z}_i)$

and $\min_{c \in \mathcal{Y}_K^{(t)}} P_c(\boldsymbol{x}_{b_0}')$, but fails to account for the influence of $\boldsymbol{z}_i$ on all target classes $c \in \mathcal{Y}_K^{(t)}$. In contrast, Method-3 and Method-4 incorporate the impact of $\boldsymbol{z}_i$ on each of the target classes. While Method-3 counts the number of target classes with increased (or decreased) confidence when $\phi_{\boldsymbol{x}_{b_0}'}(\boldsymbol{z}_i) = 1$ (or $\phi_{\boldsymbol{x}_{b_0}'}(\boldsymbol{z}_i) = -1$), Method-4 considers the strength of these classes. As demonstrate in Table 3, the query efficiency of Method-3 increases with higher $K$, and the Method-4 is query efficient than Method-1 in locating $\boldsymbol{x}_{b_0}$ for $K > 1$. Nevertheless, the proposed gradient estimation method, which takes account of both Method-3 and Method-4, demonstrates superior performance in locating the initial boundary $\boldsymbol{x}_{b_0}$.

## A.2 Detailed Breakdown of Eq. 12

The proposed gradient estimation for the *top-K* targeted attack at a decision boundary point, $\boldsymbol{x}_{b_n}$, involves two key components: the count of target classes with increased (or decreased) confidence $\sum_{c \in \mathcal{Y}_K^{(t)}} \zeta_{\boldsymbol{x}_{b_n},c,\boldsymbol{z}_i}$ and the sum of the increased (or decreased) prediction probability of these classes presented as $\sum_{c \in \mathcal{Y}_K^{(t)}} w_{\boldsymbol{x}_{b_n},c,\boldsymbol{z}_i} \zeta_{\boldsymbol{x}_{b_n},c,\boldsymbol{z}_i}$ if the query is adversarial (or non-adversarial). Before delving into the impact of these components on attack performance, we explore alternative possible gradient estimation approaches.

**Approach 1:** In this scenario, we consider a more challenging setting for the gradient estimation, where an adversary only knows whether a query is within the adversarial region without obtaining prediction probabilities from the classifier as employed by CGBA (Reza et al., 2023). The estimated gradient on $\boldsymbol{x}_{b_n}$ in this challenging setting is expressed as:

$$\boldsymbol{g}_{\boldsymbol{x}_{b_n}} = \sum_{i=1}^{I_n} \mathbb{1}(\boldsymbol{x}_{b_n} + \boldsymbol{z}_i) \cdot \boldsymbol{z}_i. \tag{18}$$

**Approach 2:** In this approach, the adversary has access to prediction probabilities of all the classes, and the gradient estimation is done by summing the changes in predictions of all the target classes in $\mathcal{Y}_K^{(t)}$. The gradient estimation using this approach is given as:

$$\boldsymbol{g}_{\boldsymbol{x}_{b_n}} = \sum_{i=1}^{I_n} \Big( \sum_{c \in \mathcal{Y}_K^{(t)}} w_{\boldsymbol{x}_{b_n},c,\boldsymbol{z}_i} \Big) \cdot \boldsymbol{z}_i, \tag{19}$$

where $w_{\boldsymbol{x}_{b_n},c,\boldsymbol{z}_i} = P_c(\boldsymbol{x}_{b_n} + \boldsymbol{z}_i) - P_c(\boldsymbol{x}_{b_n})$ is defined in Eq. 5. More weight is assigned to $\boldsymbol{z}_i$, if the aggregated change across $\mathcal{Y}_K^{(t)}$ is higher.

**Approach 3:** To estimate the gradient at the boundary point $\boldsymbol{x}_{b_n}$, we may compare the minimum confidence among the target classes in $\mathcal{Y}_K^{(t)}$ at $\boldsymbol{x}_{b_n}$ with that of the queries around it by adding noise $\boldsymbol{z}_i$ to $\boldsymbol{x}_{b_n}$, and it can be represented as:

$$\boldsymbol{g}_{\boldsymbol{x}_{b_n}} = \sum_{i=1}^{I_n} \Big( \min_{c \in \mathcal{Y}_K^{(t)}} P_c(\boldsymbol{x}_{b_n} + \boldsymbol{z}_i) - \min_{c \in \mathcal{Y}_K^{(t)}} P_c(\boldsymbol{x}_{b_n}) \Big) \cdot \boldsymbol{z}_i. \tag{20}$$

This gradient estimator estimates the direction to enhance the minimum confidence among $\mathcal{Y}_K^{(t)}$.

**Approach 4:** Having the access to the prediction probabilities of all the classes, the misclassification objective of the popular Carlini-Wagner (CW) (Carlini & Wagner, 2017) can be adapted to estimate the gradient at $\boldsymbol{x}_{b_n}$ for the *top-K* attack, and it is given as follows:

$$\boldsymbol{g}_{\boldsymbol{x}_{b_n}} = \sum_{i=1}^{I_n} \Big( \min_{c \in \mathcal{Y}_K^{(t)}} P_c(\boldsymbol{x}_{b_n} + \boldsymbol{z}_i) - \max_{c \notin \mathcal{Y}_K^{(t)}} P_c(\boldsymbol{x}_{b_n} + \boldsymbol{z}_i) \Big) \cdot \boldsymbol{z}_i. \tag{21}$$

**Approach 5:** The CW adversarial objective in optimizing the $\ell_2$-norm of perturbation is originally given in-terms of logits $\mathcal{L}$ of the classifier that is related with the prediction probabilities for a given input $\boldsymbol{x}$ as: $P(\boldsymbol{x}) = \text{softmax}(\mathcal{L}(\boldsymbol{x}))$. With the access of logits, Eq. 21 can be rewritten as:

$$\boldsymbol{g}_{\boldsymbol{x}_{b_n}} = \sum_{i=1}^{I_n} \Big( \min_{c \in \mathcal{Y}_K^{(t)}} \mathcal{L}_c(\boldsymbol{x}_{b_n} + \boldsymbol{z}_i) - \max_{c \notin \mathcal{Y}_K^{(t)}} \mathcal{L}_c(\boldsymbol{x}_{b_n} + \boldsymbol{z}_i) \Big) \cdot \boldsymbol{z}_i. \tag{22}$$

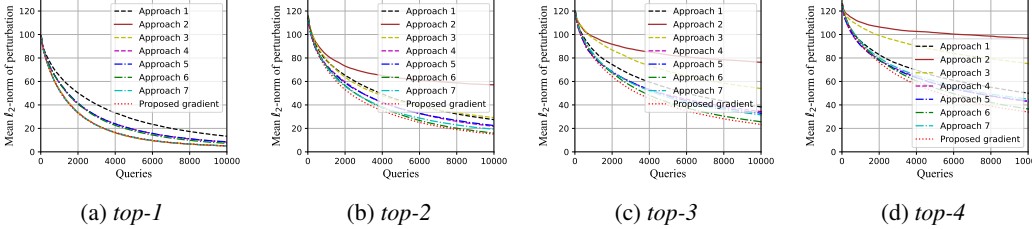

Figure 7: Comparing the variation of $\ell_2$-norm of perturbations with queries of different gradient estimation approaches on the decision boundary with our proposed gradient estimation in Eq. 12 in crafting diverse *top-K* targeted adversarial examples.

**Approach 6:** Considering only the first component of the weight to $\boldsymbol{z}_i$ in Eq. 12, the estimated gradient can be expressed as:

$$\boldsymbol{g}_{\boldsymbol{x}_{b_n}} = \sum_{i=1}^{I_n} \Big( \sum_{c \in \mathcal{Y}_K^{(t)}} \zeta_{\boldsymbol{x}_{b_n}, c, \boldsymbol{z}_i} \Big) \cdot \mathbb{1}(\boldsymbol{x}_{b_n} + \boldsymbol{z}_i) \cdot \boldsymbol{z}_i, \tag{23}$$

where $\sum_{c \in \mathcal{Y}_K^{(t)}} \zeta_{\boldsymbol{x}_{b_n}, c, \boldsymbol{z}_i}$ counts the number of target classes with increased (or decreased) confidence towards the adversarial region if the query $\boldsymbol{x}_{b_n} + \boldsymbol{z}_i$ is adversarial (or non-adversarial).

**Approach 7:** In this case, we consider the second component of the weight to $\boldsymbol{z}_i$ in Eq. 12, which aggregates the changes in confidence of the classes $c \in \mathcal{Y}_K^{(t)}$ such that $\zeta_{\boldsymbol{x}_{b_n}, c, \boldsymbol{z}_i} = 1$, to estimate the gradient at $\boldsymbol{x}_{b_n}$. Thus,

$$\boldsymbol{g}_{\boldsymbol{x}_{b_n}} = \sum_{i=1}^{I_n} \Big( \sum_{c \in \mathcal{Y}_K^{(t)}} w_{\boldsymbol{x}_{b_n}, c, \boldsymbol{z}_i} \cdot \zeta_{\boldsymbol{x}_{b_n}, c, \boldsymbol{z}_i} \Big) \cdot \boldsymbol{z}_i. \tag{24}$$

**Comparative analysis.** We present a comparative analysis of the aforementioned gradient estimation approaches with our proposed gradient estimation, as expressed in Eq. 12, on the decision boundary for targeted attacks. To facilitate the comparison, we randomly choose 100 images from the ImageNet (Deng et al., 2009) dataset and perform attacks against the popular pre-trained ResNet-50 (He et al., 2016) classifier. For a particular *top-K* and a source image, all the attacks using different estimation techniques start from a same randomly chosen initial boundary point.

In Fig. 7a, it is evident that all attacks utilizing prediction probabilities for gradient estimation outperform Approach-1, which solely relies on the classifier's decision to estimate the gradient for crafting *top-1* adversarial examples. In this *top-1* setting, the gradient estimation with Approach-6 closely resembles that of the decision-based (Approach-1). The weight associated with $\boldsymbol{z}_i$ using Approach-6 reduces to $\big(\zeta_{\boldsymbol{x}_{b_n}, y_t, \boldsymbol{z}_i} \mathbb{1}(\boldsymbol{x}_{b_n} + \boldsymbol{z}_i)\big) \in \{-1, 0, 1\}$, where $y_t$ is the target label. The key advantage of gradient estimation using Approach-6 is that it excludes anomalous queries— those are non-adversarial but increase the confidence toward the target class, and vice versa. In multi-class classification problems, there might be scenarios, for a target class $y_t$, though $P_{y_t}(\boldsymbol{x}_{b_n} + \boldsymbol{z}_i) - P_{y_t}(\boldsymbol{x}_{b_n}) > 0$, the query $\boldsymbol{x}_{b_n} + \boldsymbol{z}_i$ is still non-adversarial ($P_c(\boldsymbol{x}_{b_n} + \boldsymbol{z}_i) > P_{y_t}(\boldsymbol{x}_{b_n} + \boldsymbol{z}_i)$ for a $c \in [C] \backslash \{y_t\}$). Since the decision-based approach (Approach-1) treats all queries equally by assigning unit absolute value weights of either -1 or 1 to $\boldsymbol{z}_i$ based on whether a query is adversarial, irrespective of considering the discussed phenomenon, the inclusion of anomalous queries negatively impacts the performance of the attack using it.

In the *top-*1 setting, attacks with Approach-2, Approach-3, Approach-7, and the proposed gradient estimation offer quite strong performance. Approach-2 and Approach-3, essentially reduce to the same expression in *top-*1 setting, account for all queries by evaluating the increase or decrease in confidence toward the target class from the decision boundary, using this change in confidence as a weight for $\boldsymbol{z}_i$. Both Approach-7 and the proposed method (essentially both reduce to the same expression under *top-*1 setting) also assign weights based on the change in confidence, similar to Approach-2 and Approach-3, but they exclude anomalous queries. However, Approach-2 and Approach-3 still perform comparably to the proposed gradient estimation method, as the weights associated with the anomalous queries are relatively small. Now, turning to Approach-4 and Approach-5, which are based on CW adversarial objective, gradient esti-

mation using these approaches is not optimal. With these approaches, on the decision boundary, $\min_{c \in \mathcal{Y}_K^{(t)}} P_c(\boldsymbol{x}_{b_n} + \boldsymbol{z}_i) - \max_{c \notin \mathcal{Y}_K^{(t)}} P_c(\boldsymbol{x}_{b_n} + \boldsymbol{z}_i) > 0 \implies \mathbb{1}(\boldsymbol{x}_{b_n} + \boldsymbol{z}_i))$, and vice versa. Thus, there are no difficulties with anomalous queries. The possible reasons behind this sub-optimal performance by these approaches because the weights that are calculated based on the difference between the minimum prediction score among the target classes and the maximum prediction score among the other classes. This weighting does not accurately reflect the increase in confidence toward the target class from the decision boundary, leading to less effective gradient estimations.

Turning to Figs. 7b, 7c and 7d, we observed that when $K > 1$, Approach-2, which sums the changes in prediction probability across all target classes in $\mathcal{Y}_K^{(t)}$, performs well for *top-1* attacks but suffers a significant performance drop, even bellow Approach-1, when $K$ increases. The degradation in performance for Approach-2 is attributed to the fact that the likelihood of $\sum_{c \in \mathcal{Y}_K^{(t)}} w_{\boldsymbol{x}_b, c, \boldsymbol{z}_i} > 0$ when the query $\boldsymbol{x}_{b_n} + \boldsymbol{z}_i$ is adversarial decreases with the increase of $K$, while the expectation is with an adversarial query, the weight assigned to $\boldsymbol{z}_i$ should be positive. Consequently, as $K$ grows, gradient estimation with Approach-2 becomes less accurate and struggles to converge. Likewise, Approach-3 also suffers from a similar performance drop with $K$. This is because, on the decision boundary, the difference between $\min_{c \in \mathcal{Y}_K^{(t)}} P_c(\boldsymbol{x}_{b_n} + \boldsymbol{z}_i)$ and $\min_{c \in \mathcal{Y}_K^{(t)}} P_c(\boldsymbol{x}_{b_n})$ does not carry too much information of whether the query $\boldsymbol{x}_{b_n} + \boldsymbol{z}_i$ is adversarial. The probability of $\min_{c \in \mathcal{Y}_K^{(t)}} P_c(\boldsymbol{x}_{b_n} + \boldsymbol{z}_i) - \min_{c \in \mathcal{Y}_K^{(t)}} P_c(\boldsymbol{x}_{b_n}) > 0$ when a query is adversarial reduces as well with higher $K$ using this approach that enhances the likelihood of the anomalous queries. Approaches based on the CW adversarial objective, such as Approach-4 and Approach-5, continue to exhibit sub-optimal performance for higher $K$ due to the issues previously discussed.

For Approach-6 and Approach-7—where Approach-6 counts the number of target classes with increased (or decreased) confidence if the query is adversarial (or non-adversarial), and Approach-7 sums the confidence increases (or decreases) for these classes— while Approach-6 consistently outperforms all other aforementioned approaches for higher $K$, Approach-7 outperforms those when $K$ is smaller. For a query at $\boldsymbol{x}_{b_n}$ with added $\boldsymbol{z}_i$ on the decision boundary, these approaches focus on the impact of $\boldsymbol{z}_i$ on each of the target classes rather than looking at a whole. For example, they do not consider the impact of $\boldsymbol{z}_i$ on the class $\arg \min_{c \in \mathcal{Y}_K^{(t)}} P_c(\boldsymbol{x}_{b_n} + \boldsymbol{z}_i)$, but all classes in $\mathcal{Y}_K^{(t)}$. They assign more weight to $\boldsymbol{z}_i$ in estimating gradient, if it impacts more target classes. By incorporating the indicator function $\zeta_{\boldsymbol{x}_{b_n}, c, \boldsymbol{z}_i}$, they exclude queries that are adversarial but do not positively impact any $c \in \mathcal{Y}_K^{(t)}$. Notably, our proposed gradient estimation method, which integrates the strengths of both Approach-6 and Approach-7, outperforms all other approaches when $K > 1$.

### A.3 IMPACT OF NON-ADVERSARIAL QUERIES

The proposed GSBA$^K$ calculates gradients by incorporating both adversarial and non-adversarial queries. When estimating gradients at a boundary point $\boldsymbol{x}_{b_n}$, if a query $\boldsymbol{x}_q = \boldsymbol{x}_{b_n} + \boldsymbol{z}_i$ is non-adversarial due to added noise $\boldsymbol{z}_i$, GSBA$^K$ incorporates this noise by inverting it and scaling it with the absolute value of the weights assigned to $\boldsymbol{z}_i$ in Eq. 12 for targeted attacks and Eq. 13 for untargeted attacks. Eq. 13 can be rewritten as follows:

$$\boldsymbol{g}_{\boldsymbol{x}_{b_n}} = \sum_{i=1}^{I_n} |F_{\boldsymbol{x}_{b_n} + \boldsymbol{z}_i}| \mathbb{1}(\boldsymbol{x}_{b_n} + \boldsymbol{z}_i) \boldsymbol{z}_i. \tag{25}$$

Likewise, Eq. 12 can be rewritten as:

$$\boldsymbol{g}_{\boldsymbol{x}_{b_n}} = \sum_{i=1}^{I_n} \Big( \sum_{c \in \mathcal{Y}_K^{(t)}} \zeta_{\boldsymbol{x}_{b_n}, c, \boldsymbol{z}_i} \Big) | \sum_{c \in \mathcal{Y}_K^{(t)}} w_{\boldsymbol{x}_{b_n}, c, \boldsymbol{z}_i} \zeta_{\boldsymbol{x}_{b_n}, c, \boldsymbol{z}_i} | \mathbb{1}(\boldsymbol{x}_{b_n} + \boldsymbol{z}_i) \boldsymbol{z}_i. \tag{26}$$

Fig. 8 compares the performance of the proposed attack in crafting *top-1* adversarial examples when estimating gradients using both adversarial and non-adversarial queries versus considering only adversarial queries for both untargeted and targeted attacks. These experiments are conducted against the ResNet-50 (He et al., 2016) classifier using 100 correctly classified images from ImageNet (Deng et al., 2009). While Fig. 8a and Fig. 8c depict the obtained median $\ell_2$-norm of perturbation for different query budget for untargeted and targeted attacks, respectively, Fig. 8b and Fig. 8d demonstrate the corresponding attack success rate (ASR) with a perturbation threshold $r_{th} = 2$ and $r_{th} = 4$

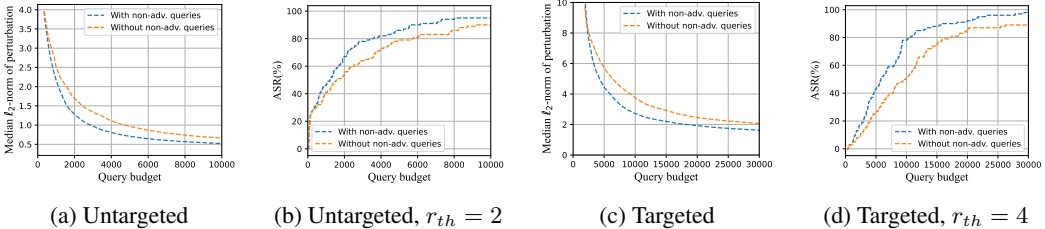

(a) Untargeted     (b) Untargeted, $r_{th} = 2$     (c) Targeted     (d) Targeted, $r_{th} = 4$

Figure 8: Impact of non-adversarial queries in estimating the gradients at decision boundary on crafting *top-1* untargeted and targeted adversarial examples against ResNet-50.

for untargeted and targeted attacks, respectively. The results clearly demonstrate that incorporating both adversarial and non-adversarial queries in gradient estimation results in improved performance compared to not considering non-adversarial queries in gradient estimation.

## A.4 IMPACT OF GRADIENT-BASED SEARCH FOR INITIAL BOUNDARY POINT

Geometric attacks start by finding an initial boundary point and then iteratively reduce the perturbation by utilizing the decision boundary's geometry. In literature, untargeted decision-based attacks find the initial boundary point in a random direction, while targeted attacks find it by using `BinarySearch` between the source and an image with the target class. Instead of employing a random direction or binary search to locate the initial boundary point, the proposed GSBA[K] employs the estimated gradient direction for both untargeted and targeted attacks. Fig. 9 demonstrates the significant improvement in the median $\ell_2$-norm

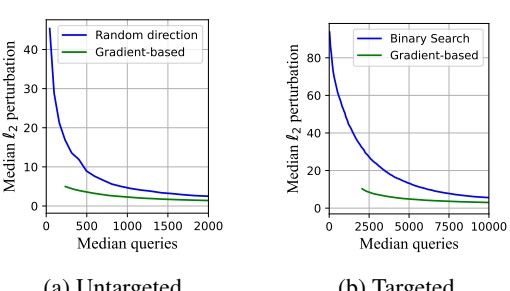

(a) Untargeted     (b) Targeted

Figure 9: Impact of gradient-based initialization against ResNet-50.

of perturbation in crafting 100 *top-1* adversarial examples against the ResNet-50 classifier using the gradient-based initial boundary finding approach for both untargeted and targeted attacks.

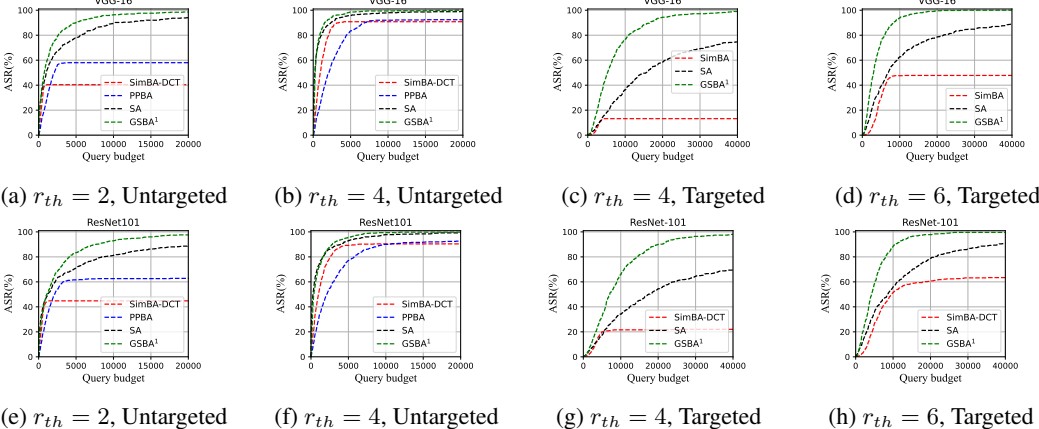

(a) $r_{th} = 2$, Untargeted    (b) $r_{th} = 4$, Untargeted    (c) $r_{th} = 4$, Targeted    (d) $r_{th} = 6$, Targeted

(e) $r_{th} = 2$, Untargeted    (f) $r_{th} = 4$, Untargeted    (g) $r_{th} = 4$, Targeted    (h) $r_{th} = 6$, Targeted

Figure 10: ASR(%) versus queries in generating *top-1* adversarial examples for different perturbation thresholds utilizing GSBA[1] and baseline attacks against single-label multi-class classifiers VGG-16 and ResNet-101 on ImageNet. GSBA[1] outperforms the baselines for both the untargeted and targeted attacks.

Table 4: ASR(%) for different perturbation thresholds ($r_{th}$) and query budgets against VGG-16.

| | | Attack Type | $r_{th}$ | \multicolumn Untargeted 1000 | 5000 | 10000 | 20000 | 30000 | $r_{th}$ | Targeted 5000 | 10000 | 20000 | 30000 | 40000 |
|---|---|---|---|---|---|---|---|---|---|---|---|---|---|---|
| VGG-16 | top-1 | SA[1] | $r_{th}$=1 | 24.8 | 49.8 | 62.4 | 72.5 | 77.8 | $r_{th}$=2 | 1.8 | 7.0 | 15.1 | 24.4 | 29.8 |
| | | GSBA[1] | | **31.1** | **64.0** | **83.0** | **90.1** | **91.0** | | **11.2** | **28.0** | **52.0** | **63.4** | **71.0** |
| | | SA[1] | $r_{th}$=2 | 49.6 | 77.7 | 89.9 | 94.0 | 96.3 | $r_{th}$=4 | 17.9 | 36.4 | 58.2 | 69.1 | 74.6 |
| | | GSBA[1] | | **50.1** | **91.0** | **98.0** | **99.1** | **99.1** | | **48.1** | **76.6** | **94.4** | **97.1** | **99.0** |
| | | SA[1] | $r_{th}$=4 | 79.2 | 95.9 | 98.1 | 99.3 | 99.8 | $r_{th}$=6 | 38.9 | 62.5 | 78.6 | 84.7 | 88.8 |
| | | GSBA[1] | | **85.9** | **97.9** | **99.0** | **99.2** | **100.0** | | **74.7** | **93.9** | **99.5** | **100.0** | **100.0** |
| | top-2 | SA[2] | $r_{th}$=1 | 9.8 | 27.9 | 40.3 | 54.0 | 59.7 | $r_{th}$=2 | 0.4 | 1.8 | 5.4 | 10.9 | 14.1 |
| | | GSBA[2] | | **14.0** | **42.1** | **61.1** | **72.1** | **79.0** | | **3.4** | **11.7** | **29.3** | **40.8** | **47.3** |
| | | SA[2] | $r_{th}$=2 | 29.1 | 59.4 | 74.1 | 83.9 | 87.9 | $r_{th}$=4 | 7.5 | 18.3 | 35.2 | 44.6 | 49.7 |
| | | GSBA[2] | | **37.0** | **79.9** | **94.0** | **97.1** | **99.0** | | **25.7** | **50.6** | **75.0** | **85.6** | **89.8** |
| | | SA[2] | $r_{th}$=4 | 62.0 | 89.7 | 95.3 | 97.7 | 98.3 | $r_{th}$=6 | 21.6 | 37.3 | 55.7 | 63.4 | 67.0 |
| | | GSBA[2] | | **63.0** | **97.0** | **97.1** | **99.0** | **100.0** | | **47.7** | **74.6** | **93.0** | **95.1** | **96.0** |
| | top-3 | SA[3] | $r_{th}$=1 | 4.9 | 15.5 | 26.5 | 40.5 | 45.6 | $r_{th}$=2 | 0.2 | 0.8 | 2.4 | 5.3 | 8.0 |
| | | GSBA[3] | | **7.9** | **33.1** | **49.9** | **69.0** | **72.1** | | **1.5** | **5.8** | **17.5** | **24.7** | **33.2** |
| | | SA[3] | $r_{th}$=2 | 17.8 | 46.7 | 62.6 | 74.8 | 81.0 | $r_{th}$=4 | 4.2 | 11.9 | 22.4 | 29.2 | 32.4 |
| | | GSBA[3] | | **24.0** | **72.1** | **87.1** | **95.1** | **97.1** | | **11.1** | **32.7** | **59.2** | **70.2** | **76.0** |
| | | SA[3] | $r_{th}$=4 | 50.4 | 82.8 | 91.9 | 96.0 | 97.3 | $r_{th}$=6 | 12.9 | 24.8 | 36.0 | 42.1 | 47.0 |
| | | GSBA[3] | | **52.1** | **93.0** | **97.0** | **98.0** | **99.0** | | **26.8** | **57.1** | **74.9** | **83.2** | **86.3** |
| | top-4 | SA[4] | $r_{th}$=1 | 2.3 | 12.7 | 20.4 | 33.3 | 38.5 | $r_{th}$=2 | 0.2 | 0.3 | 1.6 | 3.9 | 5.1 |
| | | GSBA[4] | | **4.0** | **23.0** | **41.9** | **57.1** | **62.0** | | **0.8** | **3.6** | **11.4** | **16.2** | **20.6** |
| | | SA[4] | $r_{th}$=2 | 13.0 | 41.8 | 53.9 | 69.4 | 75.9 | $r_{th}$=4 | 2.3 | 5.9 | 13.4 | 19.1 | 21.3 |
| | | GSBA[4] | | **15.1** | **58.0** | **79.0** | **92.9** | **93.9** | | **6.7** | **18.1** | **40.2** | **53.2** | **60.8** |
| | | SA[4] | $r_{th}$=4 | **45.4** | 78.0 | 88.7 | 95.2 | 96.3 | $r_{th}$=6 | 6.4 | 15.7 | 23.6 | 31.0 | 32.8 |
| | | GSBA[4] | | 42.0 | **89.1** | **95.9** | **97.9** | **98.1** | | **15.0** | **37.8** | **60.9** | **69.7** | **75.4** |

Table 5: ASR(%) for different perturbation thresholds ($r_{th}$) and query budgets against ResNet-101.

| | | Attack Type | $r_{th}$ | Untargeted 1000 | 5000 | 10000 | 20000 | 30000 | $r_{th}$ | Targeted 5000 | 10000 | 20000 | 30000 | 40000 |
|---|---|---|---|---|---|---|---|---|---|---|---|---|---|---|
| ResNet-101 | top-1 | SA[1] | $r_{th}$=1 | 27.3 | 44.7 | 52.6 | 61.8 | 65.5 | $r_{th}$=2 | 3.8 | 6.8 | 14.1 | 21.4 | 25.7 |
| | | GSBA[1] | | **28.2** | **59.6** | **72.8** | **83.4** | **85.4** | | **8.5** | **23.0** | **44.9** | **59.7** | **67.9** |
| | | SA[1] | $r_{th}$=2 | 47.3 | 71.2 | 81.1 | 88.9 | 91.2 | $r_{th}$=4 | 17.4 | 34.3 | 54.4 | 64.7 | 69.4 |
| | | GSBA[1] | | **49.1** | **83.5** | **93.0** | **97.5** | **98.8** | | **32.7** | **65.7** | **90.0** | **96.4** | **98.0** |
| | | SA[1] | $r_{th}$=4 | **73.0** | 93.0 | 97.8 | 99.2 | 99.9 | $r_{th}$=6 | 35.2 | 55.9 | 78.8 | 86.1 | 90.8 |
| | | GSBA[1] | | 70.9 | **95.7** | **99.4** | **99.8** | **100.0** | | **58.5** | **88.6** | **97.7** | **99.6** | **99.7** |
| | top-2 | SA[2] | $r_{th}$=1 | **11.4** | 24.5 | 32.6 | 43.0 | 44.0 | $r_{th}$=2 | 1.2 | 2.6 | 5.8 | 8.7 | 11.1 |
| | | GSBA[2] | | 11.3 | **38.6** | **52.9** | **67.0** | **73.3** | | **1.9** | **8.9** | **24.1** | **34.8** | **42.0** |
| | | SA[2] | $r_{th}$=2 | 27.7 | 53.1 | 63.4 | 73.4 | 77.7 | $r_{th}$=4 | 7.2 | 14.3 | 28.2 | 39.2 | 46.5 |
| | | GSBA[2] | | **27.8** | **66.2** | **82.5** | **94.0** | **96.0** | | **14.8** | **38.5** | **67.9** | **79.4** | **87.7** |
| | | SA[2] | $r_{th}$=4 | **54.9** | 82.6 | 91.2 | 96.7 | 97.7 | $r_{th}$=6 | 14.5 | 30.9 | 52.7 | 62.1 | 68.1 |
| | | GSBA[2] | | 48.3 | **87.7** | **95.8** | **99.4** | **99.6** | | **29.8** | **61.7** | **88.6** | **95.0** | **96.9** |
| | top-3 | SA[3] | $r_{th}$=1 | 6.4 | 17.3 | 26.3 | 35.6 | 40.5 | $r_{th}$=2 | 0.9 | 1.4 | 2.7 | 4.1 | 4.8 |
| | | GSBA[3] | | **6.6** | **28.7** | **42.4** | **58.8** | **66.6** | | **1.3** | **4.6** | **12.5** | **21.1** | **28.7** |
| | | SA[3] | $r_{th}$=2 | **19.8** | 41.5 | 52.8 | 63.3 | 68.1 | $r_{th}$=4 | 3.3 | 7.2 | 15.2 | 20.3 | 22.9 |
| | | GSBA[3] | | 16.6 | **55.3** | **74.9** | **88.2** | **92.1** | | **6.3** | **22.7** | **47.6** | **61.2** | **72.2** |
| | | SA[3] | $r_{th}$=4 | **43.0** | 74.0 | 83.9 | 92.2 | 94.4 | $r_{th}$=6 | 8.7 | 16.7 | 29.7 | 38.9 | 43.7 |
| | | GSBA[3] | | 38.2 | **81.8** | **94.1** | **98.3** | **99.5** | | **16.7** | **41.4** | **72.7** | **83.6** | **89.6** |
| | top-4 | SA[4] | $r_{th}$=1 | **3.8** | 12.0 | 17.0 | 23.1 | 26.2 | $r_{th}$=2 | 0.5 | 0.8 | 1.3 | 2.1 | 2.9 |
| | | GSBA[4] | | 3.4 | **20.9** | **35.6** | **50.2** | **58.4** | | **0.5** | **1.4** | **6.7** | **12.2** | **16.9** |
| | | SA[4] | $r_{th}$=2 | **13.6** | 35.6 | 46.3 | 58.4 | 62.6 | $r_{th}$=4 | 1.5 | 5.0 | 8.1 | 11.8 | 14.8 |
| | | GSBA[4] | | 11.9 | **47.4** | **69.3** | **83.4** | **88.5** | | **2.4** | **12.5** | **30.5** | **44.8** | **53.5** |
| | | SA[4] | $r_{th}$=4 | **38.3** | 68.3 | 79.5 | 87.9 | 92.5 | $r_{th}$=6 | 3.7 | 9.3 | 17.3 | 22.7 | 28.3 |
| | | GSBA[4] | | 31.8 | **74.6** | **90.4** | **97.2** | **98.5** | | **8.0** | **25.3** | **53.2** | **70.1** | **78.7** |

# B    COMPARISON WITH *top-1* BASELINES

Fig. 10 depicts the variation of ASR with differing query budgets across various $r_{th}$ values, comparing the performance of GSBA[1] with the baselines SimBA-DCT (Guo et al., 2019a), PPBA (Li et al., 2020b) and SA (Andriushchenko et al., 2020) in generating *top-1* untargeted and targeted adversarial examples against the VGG-16 (Simonyan & Zisserman, 2014) and ResNet-101 (He et al., 2016) single-label multi-class classifiers on the ImageNet dataset (Deng et al., 2009). The depicted curves are derived from 1000 randomly selected samples correctly classified by the respective classifier. From this figure, GSBA[1] shows significantly better ASR in crafting adversarial examples with small perturbation thresholds for untargeted attacks. For targeted attacks, on the other hand, the proposed GSBA[1] outperforms the baselines by a considerable margin in ASR performance. The obtained ASR against VGG-16 and ResNet-101 is quite similar to the one obtained for ResNet-50, as demonstrated in the main paper.

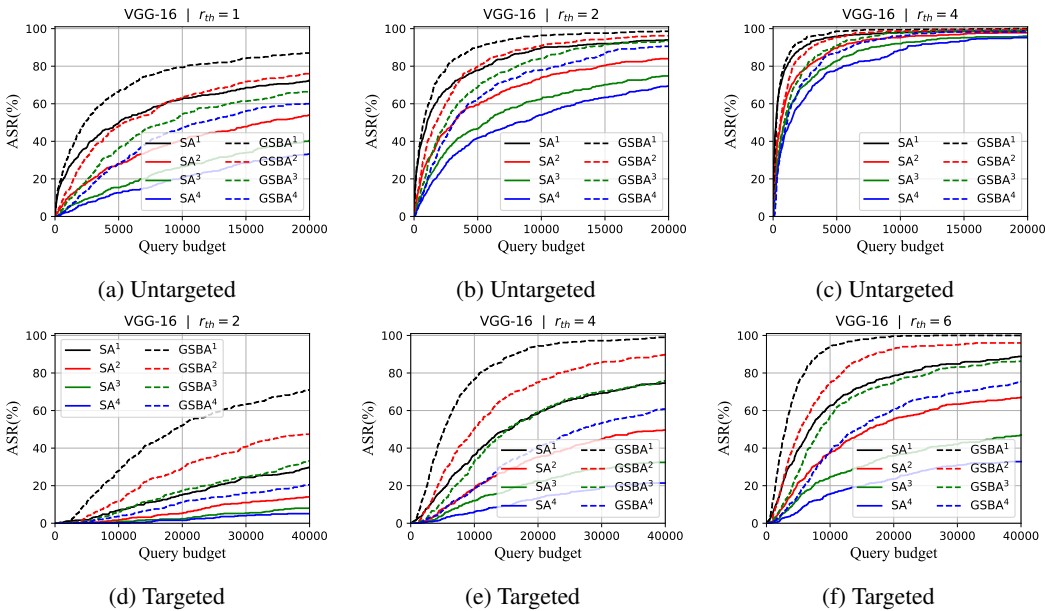

Figure 11: ASR(%) versus queries in crafting up to *top-4* adversarial examples for different perturbation thresholds utilizing GSBA[K] and SA[K] against single-label multi-class classifier VGG-16 on ImageNet. GSBA[K] consistently outperforms the state-of-the-art baseline SA[K] in crafting untargeted adversarial examples, with larger gains when the perturbation threshold is smaller and/or the query budget is larger. Moreover, the relative gain in ASR of GSBA[K] over SA[K] increases with $K$. For the targeted attack, GSBA[K] offers significantly better ASR than SA[K] across the board.

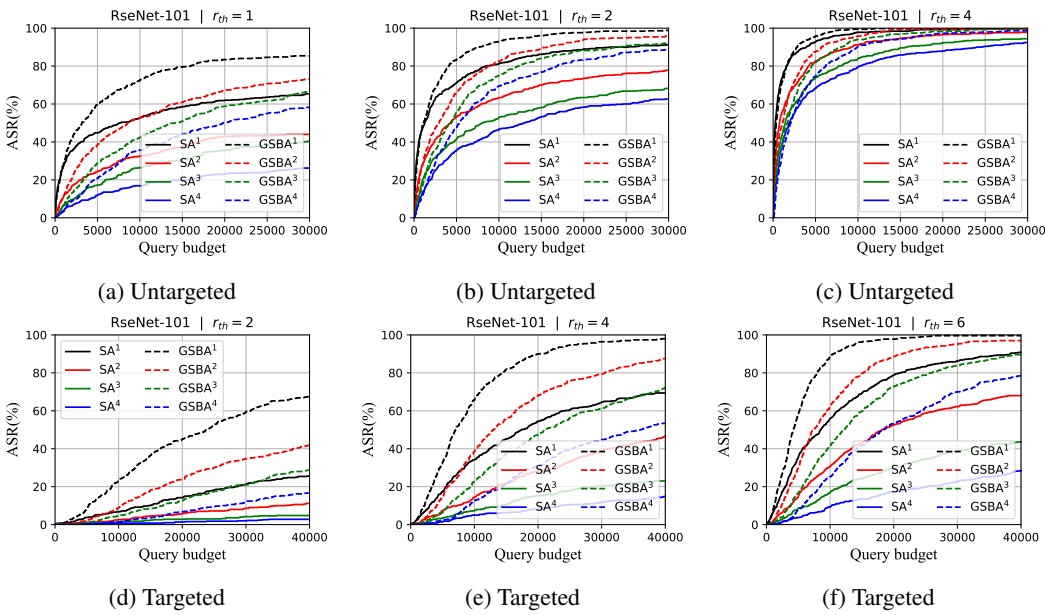

Figure 12: ASR(%) versus queries in crafting up to *top-4* adversarial examples for different perturbation thresholds utilizing GSBA[K] and SA[K] against single-label multi-class classifier ResNet-101 on ImageNet. GSBA[K] consistently outperforms the state-of-the-art baseline SA[K] in crafting untargeted adversarial examples, with larger gains when the perturbation threshold is smaller and/or the query budget is larger. Moreover, the relative gain in ASR of GSBA[K] over SA[K] increases with $K$. For the targeted attack, GSBA[K] offers significantly better ASR than SA[K] across the board.

Table 6: ASR(%) by $SA^K$ and $GSBA^K$ against different robust classifiers.

| | | Attack | Inc-v3$_{adv}$ | IR-v2$_{ens}$ | RN50$_{IN}$ | RN50$_{fine}$ | RN50$_{Aux}$ |
|---|---|---|---|---|---|---|---|
| Untargeted | $Q{=}5000, r_{th}{=}2$ | $SA^1$ | 80.0 | 63.0 | 71.0 | 76.0 | 63.0 |
| | | $GSBA^1$ | 82.0 | 69.0 | 81.0 | 86.0 | 75.0 |
| | | $SA^2$ | 63.0 | 38.0 | 50.0 | 52.0 | 37.0 |
| | | $GSBA^2$ | 72.0 | 54.0 | 66.0 | 65.0 | 54.0 |
| | | $SA^3$ | 51.0 | 30.0 | 41.0 | 43.0 | 25.0 |
| | | $GSBA^3$ | 64.0 | 44.0 | 55.0 | 58.0 | 44.0 |
| | | $SA^4$ | 41.0 | 26.0 | 35.0 | 36.0 | 20.0 |
| | | $GSBA^4$ | 55.0 | 35.0 | 46.0 | 47.0 | 33.0 |
| | $Q{=}30000, r_{th}{=}2$ | $SA^1$ | 88.0 | 81.0 | 90.0 | 93.0 | 89.0 |
| | | $GSBA^1$ | 94.0 | 93.0 | 97.0 | 100.0 | 97.0 |
| | | $SA^2$ | 79.0 | 63.0 | 78.0 | 79.0 | 63.0 |
| | | $GSBA^2$ | 91.0 | 85.0 | 96.0 | 97.0 | 97.0 |
| | | $SA^3$ | 73.0 | 58.0 | 69.0 | 75.0 | 54.0 |
| | | $GSBA^3$ | 87.0 | 83.0 | 95.0 | 95.0 | 86.0 |
| | | $SA^4$ | 67.0 | 49.0 | 58.0 | 62.0 | 44.0 |
| | | $GSBA^4$ | 86.0 | 80.0 | 92.0 | 95.0 | 85.0 |
| Targeted | $Q{=}10000, r_{th}{=}8$ | $SA^1$ | 27.0 | 15.0 | 69.0 | 75.0 | 55.0 |
| | | $GSBA^1$ | 62.0 | 39.0 | 95.0 | 93.0 | 86.0 |
| | | $SA^2$ | 8.0 | 8.0 | 39.0 | 48.0 | 24.0 |
| | | $GSBA^2$ | 35.0 | 20.0 | 79.0 | 78.0 | 63.0 |
| | | $SA^3$ | 2.0 | 4.0 | 26.0 | 30.0 | 15.0 |
| | | $GSBA^3$ | 20.0 | 13.0 | 60.0 | 65.0 | 42.0 |
| | | $SA^4$ | 0.0 | 5.0 | 16.0 | 19.0 | 10.0 |
| | | $GSBA^4$ | 8.0 | 10.0 | 42.0 | 46.0 | 27.0 |
| | $Q{=}40000, r_{th}{=}8$ | $SA^1$ | 44.0 | 22.0 | 94.0 | 97.0 | 87.0 |
| | | $GSBA^1$ | 96.0 | 74.0 | 100.0 | 100.0 | 99.0 |
| | | $SA^2$ | 15.0 | 12.0 | 79.0 | 78.0 | 59.0 |
| | | $GSBA^2$ | 74.0 | 51.0 | 99.0 | 98.0 | 98.0 |
| | | $SA^3$ | 7.0 | 7.0 | 61.0 | 59.0 | 34.0 |
| | | $GSBA^3$ | 46.0 | 39.0 | 94.0 | 95.0 | 89.0 |
| | | $SA^4$ | 4.0 | 6.0 | 46.0 | 38.0 | 26.0 |
| | | $GSBA^4$ | 32.0 | 25.0 | 89.0 | 93.0 | 80.0 |

## C  ADDITIONAL RESULTS FOR *top-K* ON IMAGENET

Table 4 and Table 5 compares the obtained ASR of SOTA baseline $SA^K$ and proposed $GSBA^K$ against single-label multi-class classifiers VGG-16 (Simonyan & Zisserman, 2014) and ResNet-101 (He et al., 2016), respectively, for both the untargeted and targeted attacks for different query budgets and perturbation thresholds $r_{th}$ in crafting up to *top-4* adversarial examples on ImageNet (Deng et al., 2009). The corresponding curves demonstrating ASR versus query budgets against VGG-16 and ResNet-101 are depicted in Fig. 11 and Fig. 12, respectively.

As can be seen from the results for untargeted attacks, the ASR against VGG-16 and ResNet-101 closely mirrors our findings against ResNet-50. The performance of the proposed $GSBA^K$ consistently outperforms the state-of-the-art score-based attack $SA^K$ in crafting adversarial examples, with larger gains when the perturbation threshold is smaller and/or the query budget is larger. When the query budget is small, all black-box attack methods are limited, so no significant gain can be expected; on the other hand, when the allowed perturbation threshold is large, the state-of-the-art method already achieves satisfactory performance, so no significant gain can be expected either. Moreover, the relative gain in ASR of $GSBA^K$ over $SA^K$ increases with $K$. For the targeted attack, $GSBA^K$ offers significantly better ASR than $SA^K$ across the board against VGG-16 and ResNet-101 classifiers, as we have seen for ResNet-50.

## D  PERFORMANCE AGAINST ROBUST CLASSIFIERS

We compare the performance of the proposed $GSBA^K$ and $SA^K$ against several robust classifiers. These include adversarially-trained Inception-V3 (Kurakin et al., 2016), ensemble-adversarially-trained-ResNet-Inception-V2 (Tramèr et al., 2017), and three robustly trained ResNet-50 (RN50): RN50$_{IN}$ (Geirhos et al., 2018), RN50$_{fine}$ (Geirhos et al., 2018) and RN50$_{Aux}$ (Hendrycks et al., 2019). RN50$_{IN}$ is trained on a combination of stylized and natural ImageNet, RN50$_{fine}$ is fine-tuned with an auxiliary dataset and RN50$_{Aux}$ is trained with advanced data augmentation techniques. We assess the performance of these robust models using 100 randomly selected samples from ImageNet for both untargeted and targeted attacks.

Table 6 demonstrates the obtained ASR(%) for query budgets of 5000 and 30000 at a perturbation threshold $r_{th} = 2$ for the untargeted attack, and for query budget of 10000 and 40000 considering the perturbation threshold $r_{th} = 8$ for the targeted attack. These results show that $GSBA^K$ in the *top-*

Table 7: ASR against ResNet50 on 100 ImageNet images for different *top-K* targeted attacks with a perturbation threshold $r_{th} = 20$.

| K | 1 | 3 | 5 | 10 | 15 | 20 |
|---|---|---|---|---|---|---|
| ASR(%) | 100.00 | 100.00 | 98.00 | 69.00 | 36.00 | 11.00 |

Table 8: ASR against ResNet50 on 100 ImageNet images for different *top-K* untargeted attacks with a perturbation threshold $r_{th} = 20$.

| K | 1 | 5 | 10 | 20 | 30 | 40 | 50 |
|---|---|---|---|---|---|---|---|
| ASR(%) | 100.00 | 99.00 | 84.00 | 71.00 | 54.00 | 43.00 | 30.00 |

Table 9: ASR against Inception-V3 on 100 PASCAL VOC images for different *top-K* targeted attacks considering best target classes with a perturbation threshold $r_{th} = 20$.

| top-K | 1 | 3 | 5 | 7 | 9 |
|---|---|---|---|---|---|
| ASR(%) | 100.00 | 82.00 | 61.00 | 21.00 | 2.00 |

$K$ setting follows a similar pattern to its performance against standard classifiers. Additionally, from this table, adversarially trained classifiers demonstrate greater robustness than the robustly trained classifiers like $RN50_{IN}$, $RN50_{fine}$, and $RN50_{Aux}$.

## E  IMPACT OF LARGER $K$ ON ASR

We conduct attacks with higher values of $K$ to evaluate its impact on ASR. As shown in Table 7, we report the ASR against ResNet50 for targeted attacks, where the *top-K* predicted labels are replaced by the target classes. Table 8 presents the ASR for untargeted attacks on 100 ImageNet images, where the *top-K* predicted labels of source images are replaced by any $K$ labels other than the *top-K* source labels. Furthermore, Table 9 illustrates the ASR variation for *top-K* targeted attacks on Inception-V3 using 100 images from the multi-label PASCAL VOC2012 dataset. From these results, we observe that ASR decreases as $K$ increases, with a sharper decline for targeted attacks. This trend suggests that the proposed black-box attack, which lacks knowledge of the target model, becomes ineffective as $K$ approaches the extreme of $K = C/2$ under the perturbation constraint, where $C$ is the total number of classification labels.

Table 10: Median $\ell_2$-norm of perturbations (lower is better) for different query budgets and different dimension reduction factors for *top-2* targeted attacks on 100 ImageNet images against ResNet50.

| f | 1 | 2 | 3 | 4 | 6 | 8 |
|---|---|---|---|---|---|---|
| $Q$=4000 | 15.82 | 12.55 | 11.74 | 9.29 | 11.72 | 10.22 |
| $Q$=8000 | 10.30 | 8.47 | 7.05 | 5.72 | 6.40 | 6.92 |

Table 11: ASR(%) with $r_{th} = 20$ for different query budgets and different dimension reduction factors for *top-2* targeted attacks on 100 ImageNet images against ResNet50.

| f | 1 | 2 | 3 | 4 | 6 | 8 |
|---|---|---|---|---|---|---|
| $Q$=4000 | 47.00 | 59.00 | 64.00 | 72.00 | 66.00 | 68.00 |
| $Q$=8000 | 89.00 | 94.00 | 95.00 | 99.00 | 97.00 | 96.00 |

## F  SAMPLING NOISE FROM LOW-FREQUENCY SUBSPACE

The low-frequency subspace obtained through Discrete Cosine Transformation (DCT) is crucial because it captures essential image information, including the gradient information (Guo et al., 2018; Li et al., 2020a), which are vital for our approach. To sample a noise $z_i \in [0,1]^{C_h \times H \times W}$ from low-frequency subspace, the coefficients of the low-frequency components with dimension $C_h \times \frac{W}{f} \times \frac{H}{f}$ are drawn from a normal distribution with zero mean and $\sigma = 0.0002$ standard deviation while setting the remaining frequency components to zero, and then revert this representation back to the original image space using the Inverse DCT, as discussed in (Guo et al., 2018). This process effectively allows us to generate perturbations that influence the more critical, low-frequency aspects of the image, minimizing disruption to the less important high-frequency details. The parameter $f$ serves as a dimension reduction factor, controlling how much of the frequency space is retained in

the noise sample. For instance, if the dimension of an image is $3 \times 224 \times 224$, the dimension of the low-frequency space with dimension reduction factor $f = 4$ would be 9408. This corresponds to selecting 6.25% of the frequency components. We choose the dimension reduction factor as $f = 4$ for all our experiments following (Reza et al., 2023; Li et al., 2020a). The median $\ell_2$-norm of perturbations and ASR for different query budgets and different dimension reduction factors for targeted *top-2* attack on 100 images are reported in Table 10 and Table 11, respectively, which conforms with the finding in (Reza et al., 2023) even for *top-2* attacks.

## G  LIMITATIONS AND POTENTIAL NEGATIVE IMPACTS

**Limitations:** The proposed GSBA$^K$ is based on querying the target classifier to craft adversarial examples, and it offers query-efficient performance as compared to the baseline attacks. However, crafting *top-K* adversarial examples, particularly for targeted attacks with larger values of $K$, still demands a significant number of queries, leaving ample room for further improvement. Moreover, the designs of our untargeted and targeted attacks are based on the assumption of low curvature and high curvature decision boundaries, respectively. However, while the curvature is high when the boundary point is far from the source image, it tends to become flatter as the boundary point comes closer to the source image from the viewpoint of the source image. Additionally, as the proposed attack proceeds with optimizing boundary points on the decision boundary, poor transferability is incurred due to the variation of decision boundaries across classifiers. Thus, a more accurate gradient estimation technique along with adaptive boundary point search can be a future endeavor to improve the performance further. Additionally, efforts can be made to improve the transferability of the geometric adversarial attacks.

**Potential societal negative impacts:** Adversarial attacks pose significant security risks in real-world machine learning systems due to the potential misuse of adversarial perturbations for malicious purposes. Our proposed black-box GSBA$^K$ method aims to generate *top-K* adversarial examples, which can be leveraged for various real-world scenarios, as outlined in the introduction section. While our experiments primarily focus on deceiving image classifiers, the versatility of our approach extends to other domains, such as image annotation, object detection, and recommendation systems, broadening the scope of potential misuse by malicious users. However, it's important to emphasize that our objective is to highlight the vulnerabilities of machine learning systems in aggressive *top-K* settings, ultimately advocating for the implementation of defense mechanisms to enhance their security against adversarial attacks. One potential defense strategy against our proposed attack involves restricting queries around the decision boundaries.

## H  CRAFTED ADVERSARIAL EXAMPLES

In this section, we demonstrate a number of generated *top-K* adversarial examples and their corresponding perturbation, along with detailed information, for different benign inputs against different classifiers. While Fig. 13, Fig. 14 and Fig. 15 demonstrate crafted untargeted adversarial examples against ResNet-50, VGG-16 and ResNet-101 single-label multi-class classifiers, Fig. 16, Fig. 17 and Fig. 18 display the crafted targeted adversarial examples against these classifiers. Fig. 19 depicts the generated *top-2* adversarial examples against a classifier with multi-label learning. The value of $\ell_2$ at the title of each adversarial example demonstrates the $\ell_2$-norm of the crafted perturbation obtained after spending a given query budget.

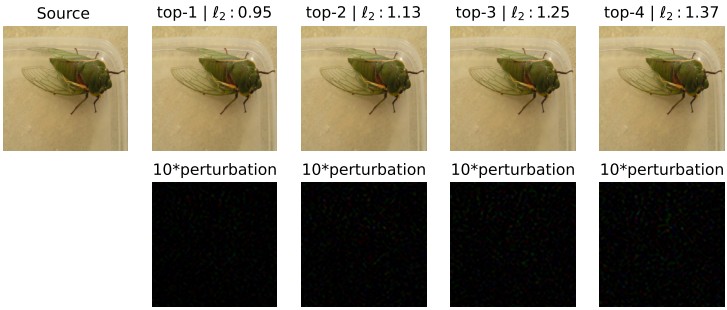

(a) Top-5 predictions of a source image on ResNet-50 are: [*cicada, leafhopper, lacewing, fly, grasshopper*]. Top-5 predictions of the *top-K* adversarial examples are as follows: *top-1*: [*dragonfly, cicada, fly, damselfly, lacewing*], *top-2*: [*fly, dragonfly, cicada, damselfly, lacewing*], *top-3*: [*dragonfly, lacewing, damselfly, cicada, fly*], and *top-4*: [*dragonfly, damselfly, lacewing, fly, cicada*].

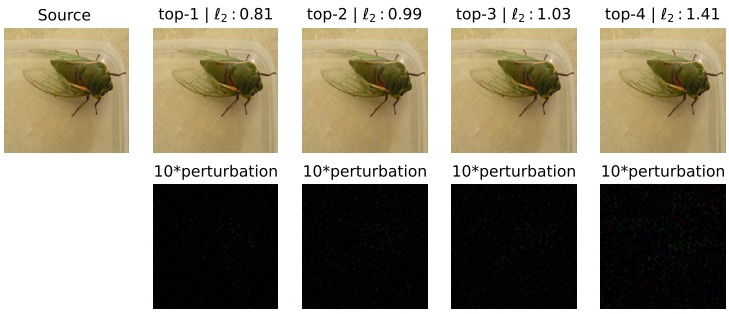

(b) Top-5 predictions of a source image on VGG-16 are: [*cicada, leafhopper, grasshopper, lacewing, mantis*]. Top-5 predictions of the *top-K* adversarial examples are as follows: *top-1*: [*leafhopper, cicada, mantis, lacewing, grasshopper*], *top-2*: [*leafhopper, mantis, cicada, grasshopper, cricket*], *top-3*: [*mantis, leafhopper, grasshopper, cicada, cricket*], and *top-4*: [*grasshopper, mantis, leafhopper, lacewing, cicada*].

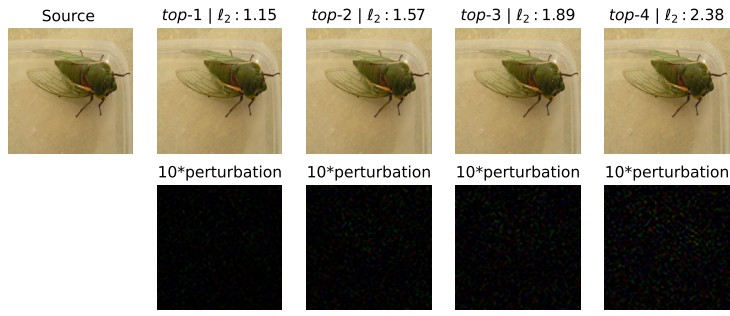

(c) Top-5 predictions of a source image on ResNet-101 are: [*cicada, leafhopper, lacewing, cricket, grasshopper*]. Top-5 predictions of the *top-K* adversarial examples are as follows: *top-1*: [*leafhopper, cicada, grasshopper, cricket, lacewing*], *top-2*: [*leafhopper, grasshopper, cicada, cricket, mantis*], *top-3*: [*grasshopper, leafhopper, cricket, cicada, cockroach*], and *top-4*: [*cricket, grasshopper, leafhopper, mantis, cicada*].

Figure 13: Crafted *top-K* **untargeted** adversarial examples of a source image with ground truth label cicada, utilizing a query budget of 5000, against different single-label multi-class classifiers.

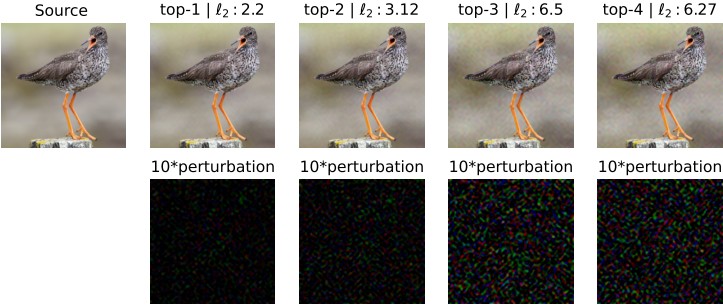

(a) Top-5 predictions of a source image on ResNet-50 are: [*redshank, ruddy turnstone, dowitcher, oystercatcher, limpkin*]. Top-5 predictions of the *top-K* adversarial examples are as follows: *top-1*: [*dowitcher, redshank, limpkin, water ouzel, ruddy turnstone*], *top-2*: [*dowitcher, limpkin, redshank, black stork, bittern*], *top-3*: [*dowitcher, limpkin, red-backed sandpiper, redshank, cicada*], and *top-4*: [*dowitcher, limpkin, spoonbill, red-backed sandpiper, redshank*].

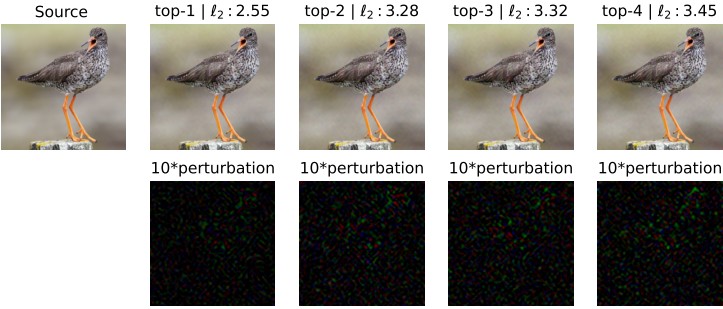

(b) Top-5 predictions of a source image on VGG-16 are: [*redshank, oystercatcher, dowitcher, ruddy turnstone, red-breasted merganser*]. Top-5 predictions of the *top-K* adversarial examples are as follows: *top-1*: [*ruddy turnstone, redshank, dowitcher, water ouzel, red-backed sandpiper*], *top-2*: [*ruddy turnstone, dowitcher, redshank, red-backed sandpiper, water ouzel*], *top-3*: [*cicada, water ouzel, ruddy turnstone, redshank, leafhopper*], and *top-4*: [*cicada, leafhopper, weevil, water ouzel, redshank*].

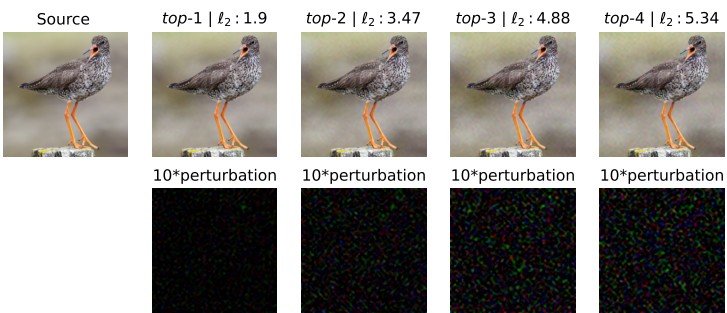

(c) Top-5 predictions of a source image on ResNet-101 are: [*redshank, oystercatcher, ruddy turnstone, dowitcher, red-breasted merganser*]. Top-5 predictions of the *top-K* adversarial examples are as follows: *top-1*: [*dowitcher, redshank, ruddy turnstone, limpkin, robin*], *top-2*: [*dowitcher, limpkin, redshank, ruddy turnstone, black stor*], *top-3*: [*limpkin, dowitcher, bittern, redshank, black stork*], and *top-4*: [*dowitcher, partridge, ruffed grouse, brambling, redshank*].

Figure 14: Crafted *top-K* **untargeted** adversarial examples of a source image with ground truth label redshank, utilizing a query budget of 5000, against different single-label multi-class classifiers.

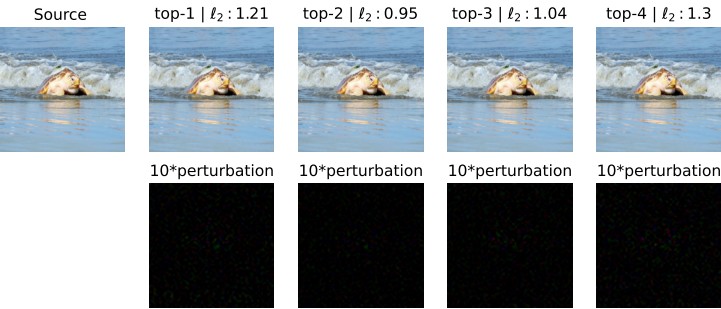

(a) Top-5 predictions of a source image on ResNet-50 are: [*loggerhead, leatherback turtle, hippopotamus, terrapin, mud turtle*]. Top-5 predictions of the *top-K* adversarial examples are as follows: *top-1*: [*leatherback turtle, loggerhead, amphibian, speedboat, conch*], *top-2*: [*amphibian, hippopotamus, loggerhead, leatherback turtle, speedboat*], *top-3*: [*amphibian, hippopotamus, speedboat, loggerhead, leatherback turtle*], and *top-4*: [*amphibian, conch, hippopotamus, speedboat, loggerhead*].

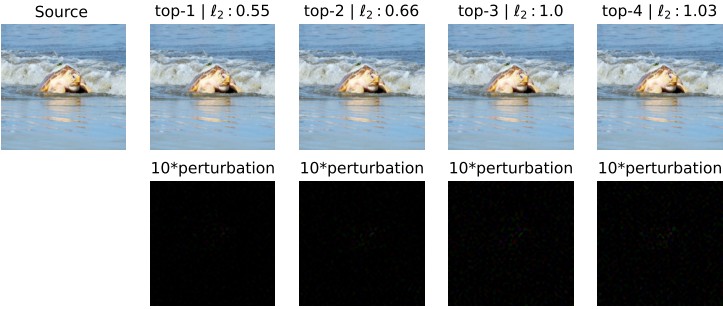

(b) Top-5 predictions of a source image on VGG-16 are: [*loggerhead, leatherback turtle, conch, hermit crab, terrapin*]. Top-5 predictions of the *top-K* adversarial examples are as follows: *top-1*: [*conch, loggerhead, hermit crab, leatherback turtle, hippopotamus*], *top-2*: [*conch, hermit crab, loggerhead, leatherback turtle, fiddler crab*], *top-3*: [*conch, hippopotamus, hermit crab, loggerhead, bathing cap*], and *top-4*: [*conch, hermit crab, bathing cap, hippopotamus, loggerhead*].

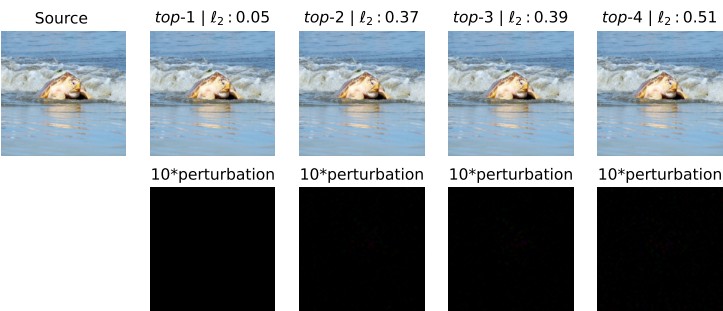

(c) Top-5 predictions of a source image on ResNet-101 are: [*loggerhead, hippopotamus, leatherback turtle, conch, hermit crab*]. Top-5 predictions of the *top-K* adversarial examples are as follows: *top-1*: [*hippopotamus, loggerhead, leatherback turtle, conch, hermit crab*], *top-2*: [*hippopotamus, bikini, loggerhead, bathing cap, conch*], *top-3*: [*hippopotamus, bikini, bathing cap, loggerhead, swimming trunks*], and *top-4*: [*hippopotamus, conch, hermit crab, bikini, loggerhead*].

Figure 15: Crafted *top-K* **untargeted** adversarial examples of a source image with ground truth label loggerhead, utilizing a query budget of 5000, against different single-label multi-class classifiers.

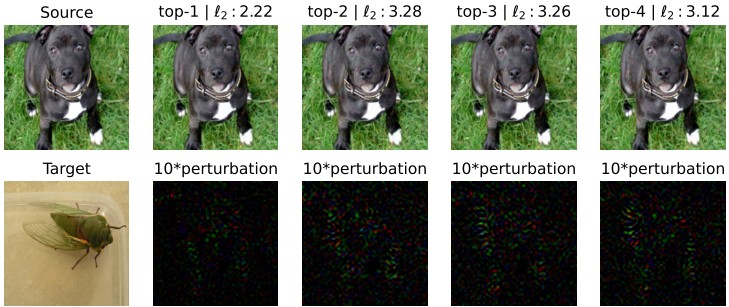

(a) Top-5 predictions of a source image and a target image on ResNet-50 are [*Staffordshire bullterrier, American Staffordshire terrier, Boston bull, French bulldog, basenji*] and [*cicada, leafhopper, lacewing, fly, grasshopper*], respectively. Top-5 predictions of the *top-K* adversarial examples are as follows: *top-1*: [*cicada, Staffordshire bullterrier, American Staffordshire terrier, miniature pinscher, muzzle*], *top-2*: [*cicada, leafhopper, American Staffordshire terrier, Staffordshire bullterrier, cricket*], *top-3*: [*cicada, lacewing, leafhopper, American Staffordshire terrier, Staffordshire bullterrier*], and *top-4*: [*cicada, fly, lacewing, leafhopper, dragonfly*].

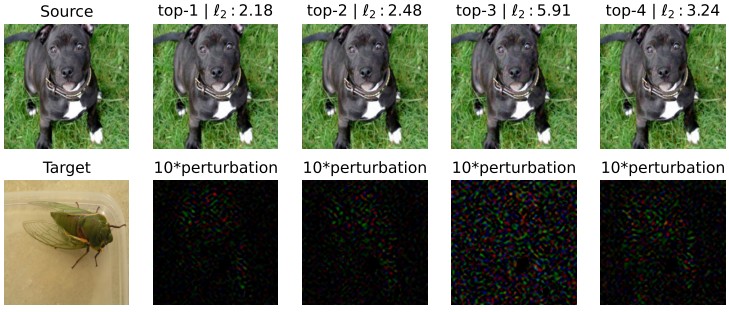

(b) Top-5 predictions of a source image and a target image on VGG-16 are [*Staffordshire bullterrier, American Staffordshire terrier, soccer ball, whippet, tennis ball*] and [*cicada, leafhopper, grasshopper, lacewing, mantis*], respectively. Top-5 predictions of the *top-K* adversarial examples are as follows: *top-1*: [*cicada, muzzle, Staffordshire bullterrier, miniature pinscher, rhinoceros beetle*], *top-2*: [*cicada, leafhopper, rhinoceros beetle, leaf beetle, muzzle*], *top-3*: [*grasshopper, leafhopper, cicada, muzzle, cricket*], and *top-4*: [*grasshopper, leafhopper, cicada, lacewing, muzzle*].

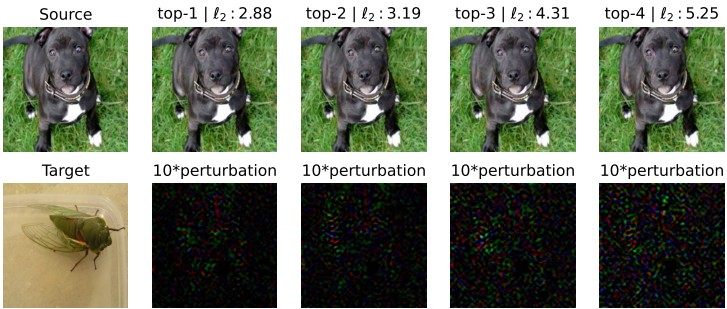

(c) Top-5 predictions of a source image target image on ResNet-101 are [*Staffordshire bullterrier, American Staffordshire terrier, French bulldog, bull mastiff, Boston bull*] and [*cicada, leafhopper, lacewing, cricket, grasshopper*], respectively. Top-5 predictions of the *top-K* adversarial examples are as follows: *top-1*: [*cicada, Staffordshire bullterrier, American Staffordshire terrier, miniature pinscher, Chihuahua*], *top-2*: [*cicada, leafhopper, Staffordshire bullterrier, cricket, American Staffordshire terrier*], *top-3*: [*cicada, leafhopper, lacewing, cricket, Staffordshire bullterrier*], and *top-4*: [*cricket, cicada, leafhopper, lacewing, grasshopper*].

Figure 16: Crafted *top-K* **targeted** adversarial examples of a source image with ground truth label Staffordshire bullterrier and a target image with *top-1* classification label cicada, utilizing a query budget of 30000, against different single-label multi-class classifiers.

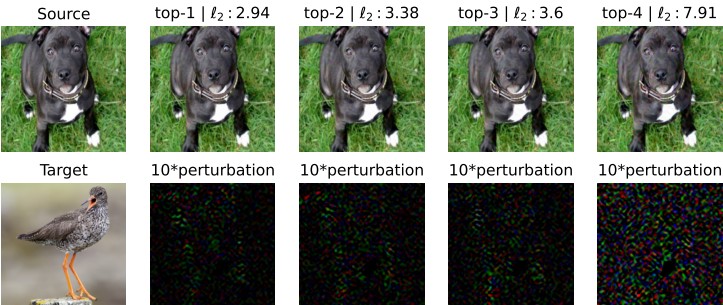

(a) Top-5 predictions of a source image and a target image on ResNet-50 are [*Staffordshire bullterrier, American Staffordshire terrier, Boston bull, French bulldog, basenji*] and [*redshank, ruddy turnstone, dowitcher, oystercatcher, limpkin*], respectively. Top-5 predictions of the *top-K* adversarial examples are as follows: *top-1*: [*redshank, Italian greyhound, American Staffordshire terrier, Weimaraner, Staffordshire bullterrier*], *top-2*: [*ruddy turnstone, redshank, Italian greyhound, American Staffordshire terrier, Staffordshire bullterrier*], *top-3*: [*ruddy turnstone, dowitcher, redshank, American Staffordshire terrier, red-backed sandpiper*], and *top-4*: [*ruddy turnstone, redshank, oystercatcher, dowitcher, magpie*].

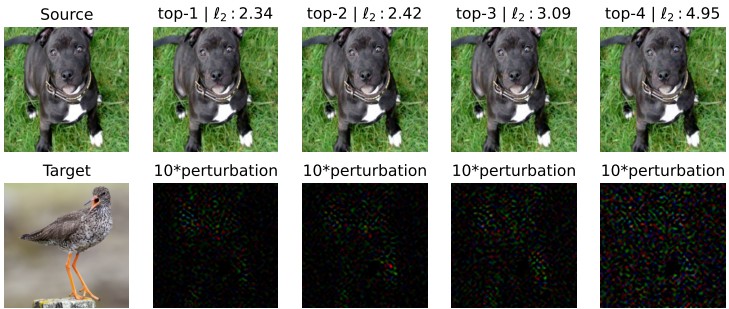

(b) Top-5 predictions of a source image and a target image on VGG-16 are [*Staffordshire bullterrier, American Staffordshire terrier, soccer ball, whippet, tennis ball*] and [*redshank, oystercatcher, dowitcher, ruddy turnstone, red-breasted merganser*], respectively. Top-5 predictions of the *top-K* adversarial examples are as follows: *top-1*: [*redshank, whippet, Scottish deerhound, American Staffordshire terrier, magpie*], *top-2*: [*oystercatcher, redshank, whippet, magpie, European gallinule*], *top-3*: [*redshank, oystercatcher, dowitcher, whippet, goose*], and *top-4*: [*ruddy turnstone, redshank, oystercatcher, dowitcher, whippet*].

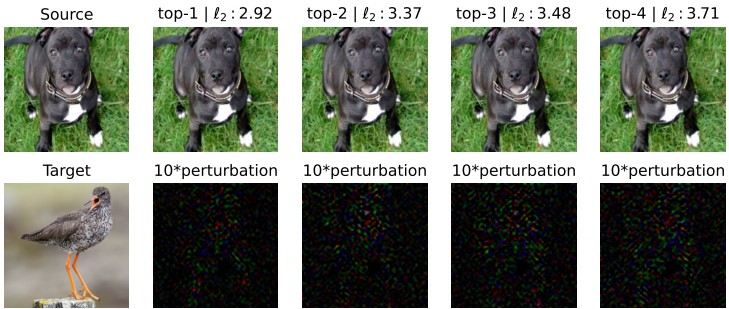

(c) Top-5 predictions of a source image and a target image on ResNet-101 are [*Staffordshire bullterrier, American Staffordshire terrier, French bulldog, bull mastiff, Boston bull*] and [*redshank, oystercatcher, ruddy turnstone, dowitcher, red-breasted merganser*], respectively. Top-5 predictions of the *top-K* adversarial examples are as follows: *top-1*: [*redshank, American Staffordshire terrier, Staffordshire bullterrier, reel, kelpie*], *top-2*: [*oystercatcher, redshank, Staffordshire bullterrier, American Staffordshire terrier, kelpie*], *top-3*: [*ruddy turnstone, redshank, oystercatcher, Staffordshire bullterrier, American Staffordshire terrier*], and *top-4*: [*redshank, ruddy turnstone, oystercatcher, dowitcher, American Staffordshire terrier*].

Figure 17: Crafted *top-K* **targeted** adversarial examples of a source image with ground truth label Staffordshire bullterrier and a target image with *top-1* classification label redshank, utilizing a query budget of 30000, against different single-label multi-class classifiers.

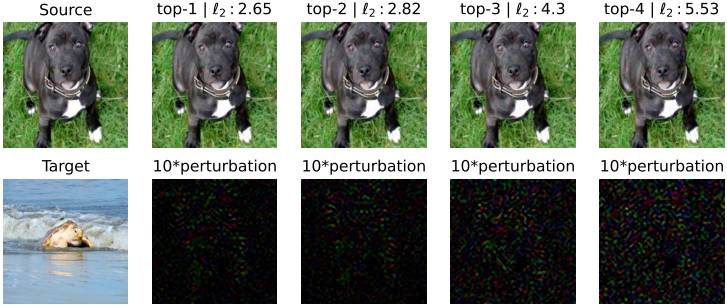

(a) Top-5 predictions of a source image and a target image on ResNet-50 are [*Staffordshire bullterrier, American Staffordshire terrier, Boston bull, French bulldog, basenji*] and [*loggerhead, leatherback turtle, hippopotamus, terrapin, mud turtle*], respectively. Top-5 predictions of the *top-K* adversarial examples are as follows: *top-1*: [*loggerhead, Staffordshire bullterrier, American Staffordshire terrier, terrapin, leatherback turtle*], *top-2*: [*leatherback turtle, loggerhead, Staffordshire bullterrier, American Staffordshire terrier, terrapin*], *top-3*: [*leatherback turtle, hippopotamus, loggerhead, Staffordshire bullterrier, terrapin*], and *top-4*: [*terrapin, leatherback turtle, hippopotamus, loggerhead, mud turtle*].

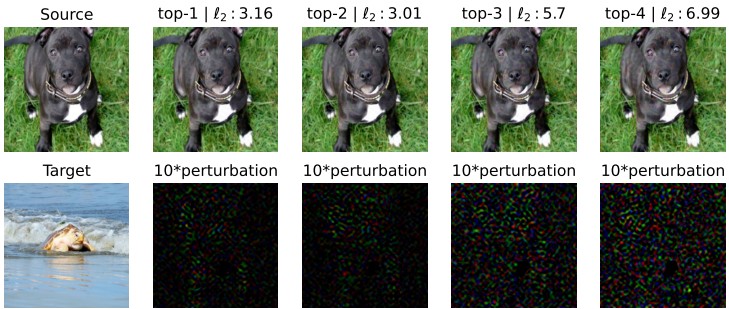

(b) Top-5 predictions of a source image and a target image on VGG-16 are [*Staffordshire bullterrier, American Staffordshire terrier, soccer ball, whippet, tennis ball*] and [*loggerhead, leatherback turtle, conch, hermit crab, terrapin*], respectively. Top-5 predictions of the *top-K* adversarial examples are as follows: *top-1*: [*loggerhead, black-and-tan coonhound, terrapin, Staffordshire bullterrier, mud turtle*], *top-2*: [*leatherback turtle, loggerhead, terrapin, mud turtle, Staffordshire bullterrier*], *top-3*: [*loggerhead, leatherback turtle, conch, terrapin, mud turtle*], and *top-4*: [*hermit crab, leatherback turtle, conch, loggerhead, terrapin*].

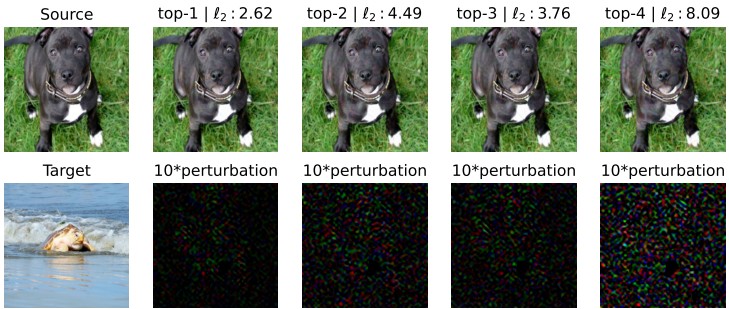

(c) Top-5 predictions of a source image and a target image on ResNet-101 are [*Staffordshire bullterrier, American Staffordshire terrier, French bulldog, bull mastiff, Boston bull*] and [*loggerhead, hippopotamus, leatherback turtle, conch, hermit crab*], respectively. Top-5 predictions of the *top-K* adversarial examples are as follows: *top-1*: [*loggerhead, Staffordshire bullterrier, American Staffordshire terrier, mud turtle, terrapin*], *top-2*: [*hippopotamus, loggerhead, Staffordshire bullterrier, American Staffordshire terrier, Mexican hairless*], *top-3*: [*loggerhead, leatherback turtle, hippopotamus, Staffordshire bullterrier, American Staffordshire terrier*], and *top-4*: [*loggerhead, hippopotamus, leatherback turtle, conch, triceratops*].

Figure 18: Crafted *top-K* **targeted** adversarial examples of a source image with ground truth label Staffordshire bullterrier and a target image with *top-1* classification label loggerhead, utilizing a query budget of 30000, against different single-label multi-class classifiers.

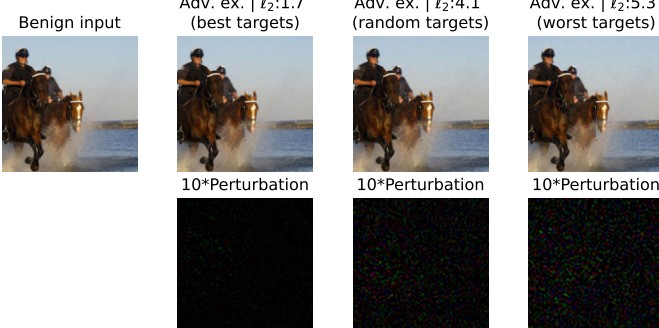

(a) Prediction order of a source image with true label set {*person, horse*} is: [*person, horse, boat, car, dog, bicycle, cow, motorbike, bus, chair, potted plant, bird, bottle, **tv/monitor**, sheep, cat, train, **sofa**, dining table, aeroplane*]. The best, **random** and worst target sets are {*boat, car*}, {*tv/monitor, sofa*} and {*dining table, aeroplane*}.

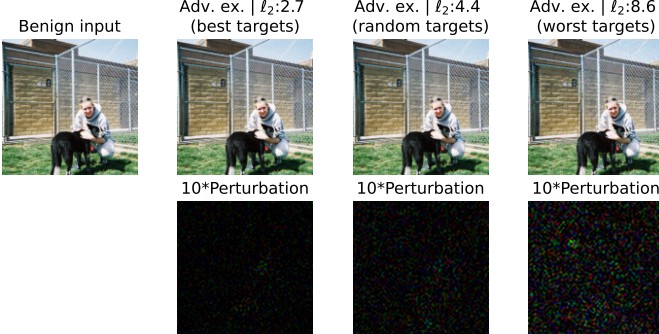

(b) Prediction order of a source image with true label set {*person, dog*} is: [*person, dog, sheep, car, horse, bottle, cow, cat, bicycle, **potted plant**, boat, sofa, chair, tv/monitor, **motorbike**, bird, train, dining table, bus, aeroplane*]. The best, **random** and worst target sets are {*sheep, car*}, {*potted plant, motorbike*} and {*bus, aeroplane*}.

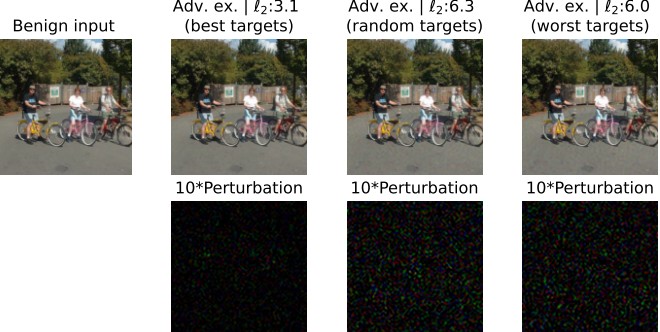

(c) Prediction order of a source image with true label set {*person, bicycle*} is: [*person, bicycle, bottle, car, chair, potted plant, bus, motorbike, cow, dog, horse, dining table, **sofa**, cat, train, **tv/monitor**, boat, sheep, bird, aeroplane*]. The best, **random** and worst target sets are {*bottle, car*}, {*tv/monitor, sofa*} and {*bird, aeroplane*}.

Figure 19: Crafted *top-2* adversarial examples for benign inputs with two true labels against Inception-V3 (Szegedy et al., 2016) on PASCAL VOC 2012 dataset (Everingham et al., 2015) with best, random and worst target label sets utilizing a query budget of 30000.

