# OpenReview forum: "GSBA$^K$: $top$-$K$ Geometric Score-based Black-box Attack"
_ICLR.cc/2025/Conference — ICLR 2025 Poster_

### Official Review · Reviewer_uZC7 · 2024-10-30

**Soundness:** 3
**Presentation:** 3
**Contribution:** 3
**Rating:** 6
**Confidence:** 3

**Summary:**

The paper introduces GSBA$^K$, a comprehensive and query-efficient geometric score-based attack specifically designed for the challenging top-K setting. GSBA$^K$ incorporates innovative gradient estimation techniques to identify a better initial boundary point and exploits the geometric properties of decision boundaries to improve both query efficiency and attack adaptability. The top-K setting presents significant difficulties in adversarial attacks, particularly for targeted attacks. The proposed method combines a novel gradient technique with the existing CGBA approach to address the complexities of score-based black-box attacks in the top-K scenario. While the proposed approach is promising, the clarity of the paper’s writing limits the accessibility of the method. For instance, it is unclear what the notation $\zeta _ {\mathbf{x} _ {b_n}, c, z _ i}$ in Eq. (6) represents or why $\mathbf{x} _ {b_n}, c, z _ i$ appear in the subscript. The distinction between the two versions of gradient estimation—one on the decision boundary and the other in the non-adversarial region—warrants clarification. An overview would help provide a broader perspective on why both approaches are necessary and how they contribute differently to the attack method. Most importantly, the paper lacks a theoretical analysis of the accuracy of the gradient estimation (e.g., examining the expected cosine similarity between the estimated gradient and the true gradient).

**Strengths:**

The top-K setting poses substantial challenges for adversarial attacks, especially in targeted scenarios. The proposed method integrates a novel gradient technique with the existing CGBA approach to tackle the complexities of score-based black-box attacks within the top-K framework. Experimental results demonstrate promising performance.

**Weaknesses:**

1. Although the ablation study and experiments in the Appendix provide valuable insights, the paper *lacks a theoretical analysis of the gradient estimation accuracy* (e.g., evaluating the expected cosine similarity between the estimated and true gradients). Without this analysis, the quality of the estimated gradient remains uncertain from a theoretical perspective.

3. Some baseline comparisons appear to be missing. For instance, the QuadAttacK method [1] is not included in the experiments. How does the ASR of GSBA$^K$ compare to QuadAttacK?

3. The writing in Sections 4.1 and 4.2 makes it difficult to understand the proposed approach. For example, the notation $\zeta _ {\mathbf{x} _ {b_n}, c, z _ i}$ in Eq. (6) is unclear, and the reason for including $\mathbf{x} _ {b_n}, c, z _ i$ in the subscript is not well-explained. The symbols used in the formulas are overly complex and lack brevity.

[1] Paniagua, T., Grainger, R., & Wu, T. (2023). QuadAttacK: A Quadratic Programming Approach to Learning Ordered Top-K Adversarial Attacks. In *Advances in Neural Information Processing Systems 36 (NeurIPS 2023)*.

**Questions:**

1. Targeted attacks in the top-K setting for black-box scenarios are highly challenging. A key question is how to ensure that the estimated gradient consistently points toward the narrow adversarial region corresponding to the $K$ target classes. Additionally, how is a high attack success rate (ASR) maintained in such targeted attacks?

2. Could you provide a simplified overview explaining why there are two versions of gradient estimation (one on the decision boundary and the other in the non-adversarial region)? This would help make the rationale for this choice more accessible and easier to understand.

---

> ### Author Response · Authors · 2024-11-21
> **Response to Reviewer uZC7 (pt 1/2)**
>
> We sincerely appreciate for your thoughtful comments. Below, we provide responses to each of the points raised.
>
> ---
>
> **C1. Ablation study and experiments in the Appendix provide valuable insights, the paper lacks of theoretical analysis.**
>
> Thank you for recognizing the value of our ablation study and experiments presented in the Appendix. We acknowledge that our paper lacks a theoretical analysis of gradient estimation for $top$-$K$ attacks. However, we would like to emphasize that our work is the first to propose $top$-$K$ untargeted and targeted attacks in the challenging black-box setting. Through comprehensive experiments, we demonstrate the effectiveness of our proposed gradient estimation approach in guiding $top$-$K$ attacks. Furthermore, as you noted, the ablation study and the experiments presented in Appendix offer significant insights into the rationale behind our gradient estimation choices. While we agree that a theoretical analysis in the $top$-$K$ setting is important, it presents substantial challenges and is left for future exploration.
>
> ---
>
> **C2.  How does the ASR of GSBA$^K$ compare to QuadAttacK?**
>
> The QuadAttack is a white-box attack that assumes full access to the target model for crafting adversarial examples. In contrast, our proposed method, GSBA$^K$, operates as a black-box attack, relying solely on querying the target model for class-wise prediction probabilities. Since these two approaches fundamentally differ in their assumptions and access levels, a direct comparison is not meaningful. Nevertheless, having the access to the target model, QuadAttacK is expected to outperform our black-box approach.
>
> ---
>
> **C3. The notation $\zeta_{\mathbf x_{b_n}, c, \mathbf z_i}$ is not clear. Why $\mathbf x_{b_n}, c, \mathbf z_i$ appear in the subscript?**
>
> We appreciate your question for clarification regarding the notation $\zeta_{\mathbf x_{b_n}, c, \mathbf z_i}$. This notation is used to describe the impact of querying a boundary point $\mathbf x_{b_n}$ with added noise $\mathbf z_i$ on a specific target class $c \in \mathcal{Y}_K^{(t)}$.
>
> Specifically:
>
>    - $\zeta_{\mathbf x_{b_n}, c, \mathbf z_i} = 1$ indicates that the query $\mathbf x_{b_n} + \mathbf z_i$ either (1) results in an adversarial query that increases the confidence for the target class $c$, or (2) is non-adversarial and decreases the confidence for $c$.
>
>   -  $\zeta_{\mathbf x_{b_n}, c, \mathbf z_i} = 0$ indicates the opposite: either the query is adversarial but decreases the confidence for $c$, or it is non-adversarial while increasing the confidence for $c$.
>
> The subscripts $\mathbf x_{b_n}, c, \mathbf z_i$ are included in the notation to concisely capture the relationship between the boundary point $\mathbf x_{b_n}$, the added noise $\mathbf z_i$, and the target class $c$. This shorthand allows us to efficiently represent the impact of the perturbed query $\mathbf x_{b_n} + \mathbf z_i$ on the confidence for the target class $c$. We hope that our explanation helps dispel the confusion; however, if you have a better suggestion on the notations and writing, we will appreciate if you let us know.
>
> ---

---

> ### Author Response · Authors · 2024-11-21
> **Response to Reviewer uZC7 (pt 2/2)**
>
> **Q1. How the proposed gradient estimation consistently points towards the narrow adversarial region corresponding to the $K$ target classes?**
>
> The proposed gradient estimation within the non-adversarial region helps to find the narrow targeted adversarial region by iteratively estimating gradient and moving the benign input image towards the estimated gradient direction. The goal of the adversary is to estimate  gradient direction that maximizes the minimum confidence among the $K$ target classes. This is achieved using Eq. (12), which estimates the gradient leveraging the decision function $\phi_{\mathbf x'\_{b_0}}(\mathbf z_i)$ from Eq. (10). This indicator function indicates whether the query $\mathbf x'_{b_0} + \mathbf z_i$ increases the minimum confidence among the target classes.
>
> For improved gradient estimation, Eq. (12) assigns weights to $\mathbf z_i$ based on their impact on each of the target classes. For instance, consider the prediction probabilities with the target classes from the target classifier for $\mathbf x'_{b_0}$ as $\lbrace..., 0.12,0.13,0.1,...\rbrace$.
> Assume, $\mathbf z_1$ changes the prediction probabilities of the target classes as $\lbrace..., 0.13,0.14,0.11,...\rbrace$, while $\mathbf z_2$ modifies these to $\lbrace..., 0.11,0.12,0.11,...\rbrace$.
>
> While both $\mathbf z_1$ and $\mathbf z_2$ enhance the minimum confidence among the target classes (i.e., $\phi_{\mathbf x'_{b_0}}(\mathbf z_i)=1$ for both), $\mathbf z_1$ has a greater impact as it improves confidence across all target classes.
>
> According to Eq. (12), the weight associated with a $\mathbf z_i$, for a query $\mathbf x'\_{b_0} + \mathbf z_i$ in the non-adversarial region,
> in estimating gradient is $\big( \sum_{c\in \mathcal{Y}_K^{(t)}} \chi\_{x'\_{b_0}, c, \mathbf z_i } \big)$
> $\big( \sum\_{c\in \mathcal{Y}_K^{(t)}} w\_{x'\_{b_0}, c, \mathbf z_i }  \chi\_{x'\_{b_0}, c, \mathbf z_i } \big)$.
>
> For these examples, form Eq. (11), $\lbrace\chi\_{\mathbf x'\_{b_0}, c, \mathbf z_1}\rbrace_{c=1}^C = \lbrace..., 1,1,1,...\rbrace$ and $\lbrace\chi\_{\mathbf x'\_{b_0}, c, \mathbf z_2}\rbrace_{c=1}^C = \lbrace..., 0,0,1,...\rbrace$,
> while $ \sum\_{c\in \mathcal{Y}_K^{(t)}} w\_{x'\_{b_0}, c, \mathbf z_i }  \chi\_{x'\_{b_0}, c, \mathbf z_1 } = 0.3$ and $ \sum\_{c\in \mathcal{Y}_K^{(t)}} w\_{x'\_{b_0}, c, \mathbf z_i }  \chi\_{x'\_{b_0}, c, \mathbf z_2 }  = 0.1$.
> Consequently, while the weight associated with $\mathbf z_1$ is $3\*0.3$, and it is with $\mathbf z_2$ is $1\*0.1$. Hence, our proposed method estimate better gradient direction towards the targeted adversarial region by weighing $\mathbf z_i$ according to its impact on the number of target classes. By iteratively finding the direction that enhances the minimum confidence among the target classes along with increasing the confidence of all the target classes, our proposed method effectively finds the narrow adversarial region.
>
> ---
>
> **Q2. Why there are two versions of gradient estimation?**
>
> We appreciate you to seeking a simplified overview explaining why there are two versions of gradient estimation. As the type of queries around a decision boundary point is different than the queries inside the non-adversarial region, we choose two different versions of gradient estimation. In case of the queries around the decision boundary point, the queries can be either in adversarial region or non-adversarial region. In gradient estimation on the decision boundary, this information plays an important role. However, when we estimate the gradient inside the non-adversarial region, targeting the adversarial region, all the queries fall within the non-adversarial region. Thus, for the targeted attack, for example, we use a different decision function, as in Eq. (10), that split the image space into two regions: $\phi_{\mathbf x'\_{b_0}}(\mathbf z_i)=1$ indicates the region of queries that ensure increased in the minimum confidence among the target class and $\phi_{\mathbf x'\_{b_0}}(\mathbf z_i)=-1$ indicates the rest of the image space. As the decision function in estimating the gradient on the boundary point and the decision function in estimating the gradient within the non-adversarial region are different, we use two different versions of gradient estimations.
>
> ---

---

> > ### Comment · Reviewer_uZC7 · 2024-11-24
> > **Further questions for A1**
> >
> > **Q3**: In gradient estimation, as it iteratively identifies directions to minimize the confidence of the original class while increasing the confidence of the target classes, does the top-$K$ setting only ensure that the final adversarial examples' top-$K$ predicted labels match the predefined target labels, regardless of their order? For instance, if I specify a predefined order for the top-5 target labels, can the $GSBA^K$ algorithm guarantee this order? In other words, does the top-$K$ setting ($K>1$) fail to ensure that the top-1 label corresponds to the first predefined target label in the set?
> >
> > By the way, is there a typo in your A3? Specifically, $\\{ \chi _ {\mathbf{x} _ {b' _ 0}, c, \mathbf{z}_1} \\} _ {c=1}^C = \\{...,0,0,1,...\\}$ and $\\{ \chi _ {\mathbf{x} _ {b' _ 0}, c, \mathbf{z}_2} \\} _ {c=1}^C = \\{...,1,1,1,...\\}$.  Based on the prediction probabilities of the target classes, could the subscripts $\mathbf{z}_1$ and $\mathbf{z}_2$ in the above equations be reversed or incorrectly written?

---

> ### Author Response · Authors · 2024-11-24
>
> Thank you for your question. Yes, our $top$-$K$ targeted adversarial attack crafts adversarial examples by changing the $top$-$K$ predictions of benign images with a predefined set of target labels regardless of the order of the target labels. The attack is considered successful if the $top$-$K$ predictions of perturbed images contain all the target labels. For more details on benign inputs and corresponding various $top$-$K$ adversarial examples, please refer to Appendix G of our original submission. That said, preserving the order of target labels within the top-K predictions in a black-box setting is a more challenging task and represents an interesting avenue for future research.
>
> We sincerely apologize for the typo in our previous explanation. You are correct in pointing out the issue with the provided examples. Specifically: $\lbrace\chi\_{\mathbf x'\_{b_0}, c, \mathbf z_1}\rbrace_{c=1}^C = \lbrace..., 1,1,1,...\rbrace$ and $\lbrace\chi\_{\mathbf x'\_{b_0}, c, \mathbf z_2}\rbrace_{c=1}^C = \lbrace..., 0,0,1,...\rbrace$. We have revised and corrected our response to address this error. Please let us know if you have any additional concerns.

---

> > ### Comment · Reviewer_uZC7 · 2024-11-25
> >
> > Thank you for your response.
> >
> > While the paper currently lacks a theoretical analysis of the gradient estimation accuracy and the proposed algorithm does not ensure the order of the top-K predicted labels, I encourage the authors to explore these aspects in future research, aiming for more challenging attacks with theoretical guarantees and methods that can preserve the order of the top-K labels. I have adjusted my score to 6.

---

> > > ### Author Response · Authors · 2024-11-25
> > >
> > > Thank you for recognizing the value of our $top$-$K$ adversarial attack and for raising your score. Your suggestion to explore methods that preserve the order of $top$-$K$ predicted labels in the black-box setting and to consider theoretical aspects represents a valuable direction for future research.

---

> ### Comment · Reviewer_uZC7 · 2024-12-03
> **The difference between the proposed GSBA^K and CGBA?**
>
> As the discussion period has been extended, I revisited this paper and found that there is no substantial difference between the proposed $GSBA^K$ and CGBA, except for the top-K setting. Therefore, the novelty appears to be limited. Specifically, Algorithm 1 in this paper closely resembles that of CGBA: both utilize the same query number for gradient estimation, adopt a similar binary search approach, and employ the same method for identifying the boundary point on the semicircular path. Could you clarify the key differences between the two methods?

---

> > ### Author Response · Authors · 2024-12-04
> > **Response to Reviewer uZC7**
> >
> > Thank you for revisiting the paper and raising the question about the differences between CGBA and the proposed GSBA$^K$. We appreciate the opportunity to clarify the key distinctions and novel contributions of GSBA$^K$. As outlined in the paper (lines 232–234), GSBA$^K$ introduces new gradient estimation methods to estimate gradients more accurately both within the non-adversarial region and at the decision boundary, especially in the aggressive $top$-$K$ setting. Below, we detail the differences between the two methods:
> >
> > ---
> >
> > **1. Finding the Initial Boundary Point**
> >
> > **CGBA:** For an untargeted attack, CGBA uses a random direction to identify a point in the adversarial region. Then it applies a binary search between this adversarial point and the source sample to locate the initial boundary point. For a targeted attack, due to the inefficiency of random exploration, a binary search is directly conducted between the source sample and a target image point to find the initial boundary point. These approaches are commonly used to locate the initial boundary point in decision-based untargeted and targeted adversarial attacks, respectively.
> >
> > **GSBA$^K$:** GSBA$^K$, in contrast, employs a gradient-based approach in the non-adversarial region to iteratively and efficiently move toward the adversarial region. Once a point in the adversarial region is identified, the binary search is performed between this point and the source image for both untargeted and targeted attacks, resulting in a much better initial boundary point (lines 256–260). This gradient-based initialization significantly reduces the initial perturbation compared to CGBA for both untargeted and targeted attacks.
> >
> > **Empirical Comparison:** As shown in Appendix A.4, in a targeted top-1 attack, the median $\ell_2$ norm of perturbation achieved by GSBA$^K$ using our proposed gradient-based initialization is approximately 10 for a query budget of 2500. In contrast, GSBA$^K$ employing CGBA's initialization under the same query budget results in a median $\ell_2$ norm larger than 25. From an alternative perspective, to achieve a comparable median $\ell_2$ norm of 10, CGBA's initialization demands a substantially higher query budget of over 7500. Similar improvements, leveraging the gradient-based approach, are observed for untargeted attacks as well.
> >
> > ---
> >
> > **2. Gradient Estimation at the Decision Boundary**
> >
> > **CGBA:** CGBA assigns uniform weights (magnitude one) to all queries when estimating the gradient at the decision boundary.
> >
> > **GSBA$^K$:** GSBA$^K$ introduces non-uniform weights based on each query's contribution to achieving the adversarial goal. As shown in Eq. (12), the proposed gradient estimator assigns weights to queries based on their impact across the target classes, enabling more precise gradient estimation at the decision boundary. This improved estimation guides the process of locating the subsequent boundary point, along a semicircular path similarly as in CGBA, but with reduced perturbation compared to CGBA (the semicircular path is different as the estimated gradient at the decision boundary is different).
> >
> > **Empirical Comparison:** Fig. 7 demonstrates that GSBA$^K$’s gradient estimation produces a more refined boundary point compared to CGBA (shown as Approach 1). Notably, this comparison uses the same initial boundary point for both methods, as determined by CGBA. From this figure, for targeted attacks with a query budget of 4000 in the top-1 setting, GSBA$^K$ achieves a median $\ell_2$ norm of perturbation of about 15, whereas that of CGBA exceeds 30. Viewed from an alternative perspective, under the same setting, to obtain a boundary point with a median $\ell_2$-norm of 20, GSBA$^K$ requires approximately 3500 queries, while CGBA requires nearly twice as many. The performance improvement in the top-K setting when K>1 is equally prominent.
> >
> > Incorporating GSBA$^K$’s gradient-based initialization would further enhance performance beyond what is observed for GSBA$^K$ in Fig. 7.
> >
> > ---
> >
> > **Summary**
> >
> > In summary, CGBA is largely built on the exploration strategy to identify the boundary point in the high-dimensional space,  and it is decision-based;  GSBA$^K$ is score-based, and built on exploitation strategy by leveraging the score information to provide more accurate gradient estimation. The key novelty of GSBA$^K$ lies in its novel gradient estimators, which improve gradient estimation in both the non-adversarial region and at the decision boundary. While employing the same query number for gradient estimation in each step, this advancement ensures dramatically improved performance in the challenging $top$-$K$ setting.
> >
> > Notably, GSBA$^K$, to the best of our knowledge, is the first work to extend black-box adversarial attacks to the $top$-$K$ setting, not only for untargeted attacks but also for the more challenging targeted attacks.

---

### Official Review · Reviewer_hY9G · 2024-11-03

**Soundness:** 4
**Presentation:** 4
**Contribution:** 3
**Rating:** 6
**Confidence:** 3

**Summary:**

The paper addresses the problem of score-based adversarial attacks on top-K predictions. They propose a novel GSBA^K attack that uses gradient estimation in the non-adversarial region and on the decision boundary to craft adversarial perturbations. The authors discuss both targeted and untargeted settings. The approach is extensively evaluated on multiple datasets and compared to existing methods in the top-K setting.

**Strengths:**

Originality: the paper proposes a novel and efficient black-box attack that relies on existing approaches to some extent but contains original elements such as novel gradient estimation methods.

Quality: proposed attack is extensively evaluated and compared to existing work. The asttack is tested on both vanilla and robust models (Table 6). Limitations and Social Impact are discussed in the Appendix F.

Reproducibility: the code is provided in the supplementary materials, not carefully checked.

Clarity: the paper is well structured, the geometrical intuition is illustrated in numerous Figures.

Significance: the paper seems to introduce valuable improvement into the black-box robustness evaluation field and addresses a challenging and insufficiently investigated field of top-K predictions.

**Weaknesses:**

Some minor points:

1) Avoid duplication of very long expressions in Equations (7), (9), (12), (13) to improve readability. Consider denoting the nominator as h and use g = h / ||h||_2

3) typo in line 024: experiental

**Questions:**

1) line 271: “experimentally it is observed that …” - do you have an explanation for this observation? It seems a bit counterintuitive to me.

2) In the line 396, could you please also state the number of classes in each dataset, especially Pascal VOC? It could be useful to compare the number of target classes K that you use in your attacks with the total number of classes N. Have you investigated extreme cases e. g. when K ~ N/2 i. e. we need to completely shift the prediction scores?

---

> ### Author Response · Authors · 2024-11-21
> **Response to Reviewer hY9G (pt 1/2)**
>
> Thank you for highlighting the strengths of our paper and providing constructive feedbacks. As suggested, we will revise the manuscript to avoid lengthy expressions, improving clarity and readability in the revised version. Below, we address your questions in detail.
>
> ---
>
> **Q1. Do you have an explanation of the observation $\zeta_{\mathbf x_{b_n}, c, \mathbf z_i} = \zeta_{\mathbf x_{b_n}, c, -\mathbf z_i}$?**
>
> Thank you for asking an explanation of the observation $\zeta_{\mathbf x_{b_n}, c, \mathbf z_i} = \zeta_{\mathbf x_{b_n}, c, -\mathbf z_i}$.
> Recall that for a target class     $c \in \mathcal{Y}_K^{(t)}$, the indicator function,
>
>  $\zeta\_{\mathbf x_{b_n}, c, \mathbf z_i} = 1$ indicates that the query $\mathcal x_{b_n} + \mathcal z_i$, near the boundary point $\mathcal x_{b_n}$, is adversarial with an increased confidence in the target class $c$ or the query $\mathcal x_{b_n} + \mathcal z_i$ is non-adversarial with a decreased confidence in $c$. Moreover, the claim $\zeta_{\mathcal x_{b_n}, c, \mathcal z_i} = \zeta_{\mathcal x_{b_n}, c, -\mathcal z_i}$ demonstrates that if the query $\mathcal x_{b_n} + \mathcal z_i$ is adversarial and increases confidence in $c$, the query with inverting $\mathbf z_i$, i.e., $\mathbf x_{b_n} - \mathbf z_i$, will be non-adversarial with decreased confidence in $c$ with a very high probability, and vice versa.
> This intuition arises from the observation that if an added vector/noise $\mathbf z_i$ with a point $\mathbf x$ in image space increases the confidence on a particular class $c$, i.e., $P_c(\mathbf x+\mathbf z_i) - P_c(\mathbf x)>0$, subtracting $\mathbf z_i$ from $\mathbf x$ is likely to decrease the confidence on class $c$, i.e., $P_c(\mathbf x-\mathbf z_i) - P_c(\mathbf x)<0$.
> Additionally, it is highly probable that queries around a boundary point $\mathbf x_{b_n}$ admit $\mathbf{1}(\mathbf x_{b_n}+\mathbf z_i) = -\mathbf{1}(\mathbf x_{b_n}-\mathbf z_i)$, which indicates that for a query $\mathbf x_{b_n}+\mathbf z_i$,  $\mathbf{1}(\mathbf x_{b_n}+\mathbf z_i)=1 \implies \mathbf{1}(\mathbf x_{b_n}-\mathbf z_i)=-1$, and vice versa.
> Intuitively, for a boundary point, if we know that moving in a given direction will result in entering the adversarial region, taking the opposite direction will likely have the opposite effect for most boundary types.
> To justify this intuition, we sample 1,000 noises from the Gaussian distribution and estimate the probability of $\mathbf{1}(\mathbf x_{b_n}+\mathbf z_i) = -\mathbf{1}(\mathbf x_{b_n}-\mathbf z_i)$ using the following equation:
>
>
>
> $\Pr (\mathbf{1}(\mathbf x_{b_n}+\mathbf z_i) = - \mathbf{1}(\mathbf x_{b_n}-\mathbf z_i)) = {1 \over 1000} {\sum\_{i=1}^{1000}}  \mathbf{\bar 1} (\mathbf{1}( \mathbf x\_{b_n} +\mathbf z_i) = - \mathbf{1}(  \mathbf x\_{b_n} -\mathbf z_i))$
>
> where $\mathbf{\bar 1}()$ return 1 if $\mathbf{1}(\mathbf x_{b_n}+\mathbf z_i) = -\mathbf{1}(\mathbf x_{b_n}-\mathbf z_i)$ satisfies, 0 otherwise, and $\mathbf z_i \sim \mathcal{N}(0, \sigma^2)$. The experimental results for different $top$-$K$ targeted attack settings are given below that demonstrates that more than 98\% times $\mathbf{1}(\mathbf x_{b_n}+\mathbf z_i) = -\mathbf{1}(\mathbf x_{b_n}-\mathbf z_i)$ satisfies.
>
> Likewise, to justify this claim $\zeta_{\mathbf x_{b_n}, c, \mathbf z_i} = \zeta_{\mathbf x_{b_n}, c, -\mathbf z_i}$, we sample 1,000 noises from the Gaussian distribution and estimate the probability of $\zeta_{\mathbf x_{b_0}, c, \mathbf z_i} = \zeta_{\mathbf x_{b_0}, c, -\mathbf z_i}$ using the following equation:
>
> $\Pr (\zeta_{\mathbf x_{b_0}, c, \mathbf z_i} = \zeta_{\mathbf x_{b_0}, c, -\mathbf z_i}) = {1 \over 1000 \cdot |\mathcal{Y}_K^{(t)}|} {\sum\_{i=1}^{1000}} {\sum\_{c \in \mathcal{Y}_K^{(t)}}} \mathbf{\underline 1} (\zeta\_{\mathbf x\_{b_0}, c, \mathbf z_i} = \zeta\_{\mathbf x\_{b_0}, c, -\mathbf z_i})$
>
> where $\mathbf{\underline 1}()$ return 1 if $\zeta_{\mathbf x_{b_0}, c, \mathbf z_i} = \zeta_{\mathbf x_{b_0}, c, -\mathbf z_i}$ satisfies, 0 otherwise. The experimental results for different $top$-$K$ targeted attack settings are given below that demonstrates that about 97\% time $\zeta_{\mathbf x_{b_0}, c, \mathbf z_i} = \zeta_{\mathbf x_{b_0}, c, -\mathbf z_i}$ satisfies.
>
>
> | $K$  | 1    | 2    | 4    | 6    |
> |-------------|:------:|:------:|:------:|:------:|
> | $\Pr (\mathbf{1}(\mathbf x_{b_n}+\mathbf z_i) = - \mathbf{1}(\mathbf x_{b_n}-\mathbf z_i))$ | 0.986 | 0.986 | 0.986 | 0.987 |
> | $\Pr (\zeta_{\mathbf x_{b_0}, c, \mathbf z_i} = \zeta_{\mathbf x_{b_0}, c, -\mathbf z_i})$ | 0.969 | 0.971 | 0.973 | 0.974 |
>
>
> **Table 1**: Experimental results to demonstrate the probability of $\mathbf{1}(\mathbf x_{b_n} + \mathbf z_i) = -\mathbf{1}(\mathbf x_{b_n} - \mathbf z_i)$ and $\zeta_{\mathbf x_{b_0}, c, \mathbf z_i} = \zeta_{\mathbf x_{b_0}, c, -\mathbf z_i}$.
>
> ---

---

> ### Author Response · Authors · 2024-11-21
> **Response to Reviewer hY9G (pt 2/2)**
>
> **Q2.1. What would be the impact of larger $K$ on ASR?**
>
> In response to your query, we conduct attacks with higher values of $K$ to evaluate its impact on ASR. As shown in Table 2, we report the ASR against ResNet50 for targeted attacks, where the $top$-$K$ predicted labels are replaced by the target classes. Table 3 presents the ASR for untargeted attacks on 100 ImageNet images, where the $top$-$K$ predicted labels of benign inputs are replaced by arbitrary labels. Furthermore, Table 4 illustrates the ASR variation for attacks on Inception-V3 using 100 images from the PASCAL VOC2012 dataset. From these results, we observe that ASR decreases as $K$ increases, with a sharper decline for targeted attacks. This trend suggests that the proposed black-box attack, which lacks knowledge of the target model, becomes ineffective as $K$ approaches the extreme of $K = N/2$ under the perturbation constraint.
>
>
> | $K$  | 1    | 3    | 5    | 10    | 15    | 20    |
> |-------------|:------:|:------:|:------:|:------:|:------:|:------:|
> |ASR(%) | 100.00 | 100.00 | 98.00 | 69.00 | 36.00 | 11.00|
>
> **Table 2**: ASR against ResNet50 on 100 ImageNet images for different $top$-$K$ targeted attack with a perturbation threshold $r_{th}=20$.
>
>
> | $K$  | 1    | 5    | 10    | 20    | 30    | 40    | 50   |
> |-------------|:------:|:------:|:------:|:------:|:------:|:------:|:------:|
> |ASR(%) | 100.00 | 99.00 | 84.00 | 71.00 | 54.00 | 43.00| 30.00 |
>
> **Table 3**: ASR(%) against ResNet50 on 100 ImageNet images for different $top$-$K$ untargeted attack with a perturbation threshold $r_{th}=20$, where the $top$-$K$ predicted label of the source images are replaced by the any $K$ labels among the rest.
>
>
> | $K$  | 1    | 3    | 5    | 7    | 9   |
> |-------------|:------:|:------:|:------:|:------:|:------:|
> |ASR(%) |100.00 | 82.00 | 61.00 | 21.00 | 2.00 |
>
> **Table 4**: ASR(%) against Inception-V3 on 100 PASCAL VOC images for different $top$-$K$ targeted attack  considering best target classes with a perturbation threshold $r_{th}=20$.
>
> ---
>
> **Q2.2. What is the number of classes in each dataset?**
>
> For our experiments, we utilize two datasets: ImageNet, which contains 1,000 classes, and PASCAL VOC 2012, which includes 20 classes. We will specify this more clearly in our revised version.

---

> > ### Comment · Reviewer_hY9G · 2024-11-24
> > **Thank you**
> >
> > Thank your for your rebuttal. I have decided to keep my score.

---

> > > ### Author Response · Authors · 2024-11-25
> > >
> > > Thank you.

---

### Official Review · Reviewer_LM2j · 2024-11-03

**Soundness:** 3
**Presentation:** 3
**Contribution:** 3
**Rating:** 8
**Confidence:** 4

**Summary:**

This paper addresses a novel setting in black-box adversarial attacks, specifically focusing on generating top-k adversarial examples with access to the target classifier's class prediction probabilities. While traditional decision-based black-box attacks rely solely on class labels, this work leverages additional information in the form of prediction probabilities to enhance attack effectiveness.
The authors propose a geometry-based framework that generates both targeted and untargeted attacks without requiring a surrogate model. The method operates iteratively: given an input data point and query access to a black-box model, it systematically updates the data point to approach the decision boundary through repeated classifier queries. At the initial boundary point, by exploiting the geometry of the decision boundary, the framework identifies boundary points with reduced perturbation norms.
Comprehensive experiments presented in both the main paper and appendix demonstrate that the additional information provided by prediction probabilities significantly enhances attack effectiveness. While existing methods fail to adequately leverage this information and perform poorly in this novel setting, the proposed approach consistently achieves superior results compared to current alternatives.

**Strengths:**

**Originality**: This work presents the first systematic investigation of black-box adversarial attacks in the top-k setting with access to prediction probabilities, addressing a previously unexplored practical scenario.

**Quality**: The methodology demonstrates technical soundness, particularly in its approach to gradient approximation. The rationale of approximating gradients in adversarial and non-adversarial regions in different ways for top-k setting makes sense.

**Clarity**: The paper is mostly clear and well-written. However, there are some missing details which I have asked in the question section. In terms of logical flow, I personally think that the section about estimating the gradients in non-adversarial region should be discussed before boundary region since that is the first step in the attack generation.

**Significance**: The paper makes substantial contributions to adversarial machine learning by introducing a novel attack setting that leverages prediction probability information in top-k predictions. The paper demonstrates how this additional information can be effectively used.

**Weaknesses:**

There are a few key weaknesses:
1. Lines 270 - 271: experimentally, it is observed that $\zeta\_{\boldsymbol{x}\_{b_n}, c, \boldsymbol{z}\_i}=\zeta\_{\boldsymbol{x}\_{b_n}, -c, \boldsymbol{z}\_i}$. This might be true for some directions, but this cannot be guaranteed. The trained models can have flat surface across a direction. The formula (7) is based on that. I understand that this works in practice, but this does not have a strong/concrete evidence.
2. One drawback of (5) is that the formulation does not consider relative change i.e., the change in a class prediction from 0.5 → 0.52 is considered the same as 0.02 → 0.04 where one might argue that the latter is better than the former.
3. Similarly, Eq(12) can weigh two directions/vectors equally even if one would overall hurt the performance of majority of classes as compared to the other. For example, consider two vectors z1 and z2. Assume we are considering top-3 attack. Consider the model predicts initially with the target class probabilities shown {….., 0.03,0.04,0.02} (assume the other class probabilities are fixed). Assume

    z1 changes the predictions to {…., 0.023,0.021,0.32}: reduces the confidence of two classes and increases one.

    z2 changes the predictions to {….., 0.13, 0.14, 0.12}: equally increases the confidence of all classes.

4. Another issue is that the method might be too dependent on the initial boundary point because once it finds a point on the boundary, to find the next boundary point, it requires that the change in the probability should be larger as in Eq(5). It is actually very likely that the confidence would decrease for the top-k predictions and correspondingly might increase for the the rest but still the top k predictions remain top k. For example, {0.3,0.3,0.3,0.1} → {0.28,0.28,0.28,0.16}. This might explain why the method doesn’t perform that well for small norms.
5. The experiments do not provide confidence intervals. For example, in table 2, it is important to know how much of deviation there is across different target classes (through standard deviation). I would highly encourage the authors to add that in the paper, if possible.
6. Although there are extensive experiments, the missing experiments are for larger K and for multi-label datasets with larger number of labels e.g., MS-COCO, OpenImages. It would be great to see the performance for larger data sizes in multi-label settings.
7. Lines 177 - 178: Should be $P(\boldsymbol{x}):[0,1]^{C\_h \times W \times H} \rightarrow [0,1]^C$ since the paper considers probabilities.
8. Since the work is relevant to multi-label learning in general, it is good to cite and mention recent multi-label attacks e.g., [1,2].

--
1. Mahmood, Hassan, and Ehsan Elhamifar. "Semantic-Aware Multi-Label Adversarial Attacks." Proceedings of the IEEE/CVF Conference on Computer Vision and Pattern Recognition. 2024.
2. Jia, Jinyuan, Wenjie Qu, and Neil Gong. "Multiguard: Provably robust multi-label classification against adversarial examples." Advances in Neural Information Processing Systems 35 (2022): 10150-10163.

**Questions:**

1. From Table 1, I do see that the performance on targeted attacks drops quickly as the K increases. For example, the best top-2 accuracy of 98.2% to top-4 accuracy of 81.4%. Which makes me question that how well the method would scale for larger multi-label datasets and for larger K e.g., K = 20 for OpenImages dataset? Please provide comments or any analysis.
2. It is not clear how the target labels are selected for targeted attack setting for ImageNet experiments. For PASCAL-VOC, how many experiments are performed for random target class setting in Table 2 and how much do the results vary? Did you observe any influence of choosing specific target classes on the performance?
3. For DCT, what percentage of low frequency components were selected? Have you performed any experiments to show the impact of choosing different number of components?
4. Since you are considering the setting where the sum of probabilities is 1, maximizing the probability of a set of target classes would reduce the probabilities of other classes. In multi-label learning, the labels are correlated and it is difficult to maximize a set of target class probabilities while decreasing the other highly correlated classes. This was observed in one of the recent works [1]. Did you observe something similar? Any comments?

--

1. Mahmood, Hassan, and Ehsan Elhamifar. "Semantic-Aware Multi-Label Adversarial Attacks." Proceedings of the IEEE/CVF Conference on Computer Vision and Pattern Recognition. 2024.

---

> ### Author Response · Authors · 2024-11-21
> **Response to Reviewer LM2j (pt 1/4)**
>
> We sincerely appreciate your thorough review and the many insightful comments you provided. We will incorporate your suggestion to present Section 4.2 before Section 4.1 and discuss relevant papers in our revised version. Below, we address your comments in detail.
>
> ---
>
> **C1. Are there any concrete evidence for the claim $\zeta_{\mathbf x_{b_n}, c, \mathbf z_i} = \zeta_{\mathbf x_{b_n}, c, -\mathbf z_i}$?**
>
>
> We appreciate you for seeking evidence for the claim $\zeta_{\mathbf x_{b_n}, c, \mathbf z_i} = \zeta_{\mathbf x_{b_n}, c, -\mathbf z_i}$. As we mentioned that experimentally, with a high probability, it is observed that $\zeta_{\mathbf x_{b_n}, c, \mathbf z_i} = \zeta_{\mathbf x_{b_n}, c, -\mathbf z_i}$, we provide some experimental evidences for the claim.
> Recall that for a target class     $c \in \mathcal{Y}_K^{(t)}$, the indicator function,
>
> $\zeta\_{\mathbf x_{b_n}, c, \mathbf z_i} = 1$ indicates that the query $\mathcal x_{b_n} + \mathcal z_i$, near the boundary point $\mathcal x_{b_n}$, is adversarial with an increased confidence in the target class $c$ or the query $\mathcal x_{b_n} + \mathcal z_i$ is non-adversarial with a decreased confidence in $c$. Moreover, the claim $\zeta_{\mathcal x_{b_n}, c, \mathcal z_i} = \zeta_{\mathcal x_{b_n}, c, -\mathcal z_i}$ demonstrates that if the query $\mathcal x_{b_n} + \mathcal z_i$ is adversarial and increases confidence in $c$, the query with inverting $\mathbf z_i$, i.e., $\mathbf x_{b_n} - \mathbf z_i$, will be non-adversarial with decreased confidence in $c$ with a very high probability, and vice versa.
> To justify the claim $\zeta_{\mathbf x_{b_n}, c, \mathbf z_i} = \zeta_{\mathbf x_{b_n}, c, -\mathbf z_i}$, we sample 1,000 noises from the Gaussian distribution and estimate the probability of $\zeta_{\mathbf x_{b_0}, c, \mathbf z_i} = \zeta_{\mathbf x_{b_0}, c, -\mathbf z_i}$ using the following equation:
>
> $\Pr (\zeta_{\mathbf x_{b_0}, c, \mathbf z_i} = \zeta_{\mathbf x_{b_0}, c, -\mathbf z_i}) = {1 \over 1000 \cdot |\mathcal{Y}_K^{(t)}|} {\sum\_{i=1}^{1000}} {\sum\_{c \in \mathcal{Y}_K^{(t)}}} \mathbf{\underline 1} (\zeta\_{\mathbf x\_{b_0}, c, \mathbf z_i} = \zeta\_{\mathbf x\_{b_0}, c, -\mathbf z_i})$
>
> where $\mathbf{\underline 1}()$ return 1 if $\zeta_{\mathbf x_{b_0}, c, \mathbf z_i} = \zeta_{\mathbf x_{b_0}, c, -\mathbf z_i}$ satisfies, 0 otherwise. The experimental results for different $top$-$K$ targeted attack settings are given below that demonstrates that about 97\% time $\zeta_{\mathbf x_{b_0}, c, \mathbf z_i} = \zeta_{\mathbf x_{b_0}, c, -\mathbf z_i}$ satisfies.
>
> | $K$  | 1    | 2    | 4    | 6    |
> |-------------|:------:|:------:|:------:|:------:|
> | $\Pr (\zeta_{\mathbf x_{b_0}, c, \mathbf z_i} = \zeta_{\mathbf x_{b_0}, c, -\mathbf z_i})$ | 0.969 | 0.971 | 0.973 | 0.974 |
>
> **Table 1**: Experimental results to demonstrate the probability of  $\zeta_{\mathbf x_{b_0}, c, \mathbf z_i} = \zeta_{\mathbf x_{b_0}, c, -\mathbf z_i}$.
>
> ---
>
> **C2. Would it be better if we replace Eq. (5) with relative change, i.e., $w_{\mathbf x_{b_n}, c, \mathbf z_i} = \big(P_c(\mathbf x_{b_n}+\mathbf z_i) - P_c(\mathbf x_{b_n})\big)/P_c(\mathbf x_{b_n})?$**
>
> Thanks for suggesting that the relative change might work better. To observe whether the suggesting change works better or not, we compare it with our proposed method under the same setting as used in depicting Fig. 7 in Appendix. The obtained $\ell_2$-norm of perturbation of 100 images against ResNet50 for targeted attack with different query budgets is given in the following table. From this table, our proposed method still performs better than the suggested change in most of the cases.
>
> | Query budget  | 1000    | 2000    | 4000    | 8000    |
> |-------------|:------:|:------:|:------:|:------:|
> | Proposed (K=1) |  50.17 | 32.3 | 16.05 | 6.59 |
> | Proposed with relative change (K=1) | 50.25 | 32.96 | 16.44 | 6.91 |
> | Proposed (k=2) | 68.74 | 52.06 | 34.21 | 18.51 |
> | Proposed with relative change (K=2) | 70.96 | 54.16 | 35.35 | 20.52 |
> | Proposed (k=3) | 78.47 | 62.89 | 45.47 | 28.05 |
> | Proposed with relative change (K=3) | 83.66 | 66.42 | 46.55 | 26.41 |
> | Proposed (k=4) | 90.04 | 74.76 | 57.14 | 38.77 |
> | Proposed with relative change (K=4) | 92.07 | 77.81 | 58.91 | 37.58 |
>
> **Table 2**: Mean $\ell_2$-norm of perturbation of the proposed method and proposed method by considering relative change.
>
> ---

---

> ### Author Response · Authors · 2024-11-21
> **Response to Reviewer LM2j (pt 2/4)**
>
> **C3. Does Eq. (12) weigh two directions/vectors equally even if one would overall hurt the performance of majority of classes as compared to the other?**
>
> Thanks for deeply thinking about the proposed method by considering some numerical values of probabilities to justify the effectiveness of the proposed method.
> Recall that in Eq. (12), the weight associated with a $\mathbf z_i$, for a query $\mathbf x'_{b_0} + \mathbf z_i$ in the non-adversarial region,
>
> in estimating gradient is $\big( \sum_{c\in \mathcal{Y}_K^{(t)}} \chi\_{x'\_{b_0}, c, \mathbf z_i } \big)$
> $\big( \sum\_{c\in \mathcal{Y}_K^{(t)}} w\_{x'\_{b_0}, c, \mathbf z_i }  \chi\_{x'\_{b_0}, c, \mathbf z_i } \big)$.
>
> For the given examples, form Eq. (11), $\lbrace\chi\_{\mathbf x'\_{b_0}, c, \mathbf z_1}\rbrace_{c=1}^C = \lbrace..., 0,0,1,...\rbrace$ and $\lbrace\chi\_{\mathbf x'\_{b_0}, c, \mathbf z_2}\rbrace_{c=1}^C = \lbrace..., 1,1,1,...\rbrace$,
> while $ \sum\_{c\in \mathcal{Y}_K^{(t)}} w\_{x'\_{b_0}, c, \mathbf z_i }  \chi\_{x'\_{b_0}, c, \mathbf z_1 } = 0.3$ and $ \sum\_{c\in \mathcal{Y}_K^{(t)}} w\_{x'\_{b_0}, c, \mathbf z_i }  \chi\_{x'\_{b_0}, c, \mathbf z_2 }  = 0.3$.
> Consequently, while the weight associated with $\mathbf z_1$ is $1\*0.3$, and it is $3\*0.3$ for $\mathbf z_2$. Hence, we can conclude that our method weigh more that enhances predictions on more target classes and vice versa.
>
> ---
>
>
> **C4. Does finding the next boundary point require a larger change in the prediction probability?**
>
>
> Finding the next boundary point mainly depends on how accurately the gradient on the boundary point is estimated. To achieve this, the weight of a noise $\mathbf z_i$ is determined by evaluating its impact on the number of target classes when applied to the query $\mathbf x_{b_n} + \mathbf z_i$. As described in Eq. (7), gradient estimation at the boundary point $\mathbf x_{b_n}$ involves assigning weights based on the query's impact on the number of target classes (via Eq. (6)) and aggregating the changes in prediction probabilities for these classes using Eq. (5).
> In essence, both the change in prediction probabilities and the number of target classes positively affected by $\mathbf z_i$ play a significant role. Therefore, it is not accurate to assert that a larger change in prediction probability is always required to locate the next boundary point.
> Nevertheless, we are not sure if we fully understand the points you raised. Please further clarify if applicable, and we will be happy to further address the remaining concerns.
>
> ---
>
> **C5, Q2. How many random experiments are performed for the random target class setting in the experiments with PASCAL VOC2012 images? Is there any influence of choosing specific target classes on the performance?**
>
>
> In demonstrating Table 2 in the main paper, we take 100 samples and randomly choose the target labels. Then for a given perturbation threshold $r_{th}$ and query budget, we determine the ASR as in Table 2 in the main paper. As you suggested, we perform additional three experiments by changing the random seed and obtain the mean and standard deviation of ASR for $K=2$. The results are demonstrated in the following table. From this table, there are some deviation as the attack success largely depends on the target classes that are chosen for attack. We will add this result and relevant discussion in the revision.
>
>
> | Query  | 10000    | 20000    | 30000    | 40000    |
> |-------------|:------:|:------:|:------:|:------:|
> | $r_{th}$=4 | 9.6$\pm$3.57	| 23.8$\pm$3.48	| 38.2$\pm$6.15	| 45.0$\pm$5.40 |
> | $r_{th}$=6	| 21.8$\pm$3.83	| 38.6$\pm$4.49	| 46.4$\pm$5.92	| 53.0$\pm$5.76 |
> | $r_{th}$=8	| 29.6$\pm$4.41	| 52.2$\pm$4.46	| 61.0$\pm$4.35	| 67.6$\pm$5.07 |
>
> **Table 3**: Mean $\pm$ Std Dev of ASR for 100 VOC images with two true label for $top$-$2$ targeted attack setting for four experiments. The target classes are chosen randomly.
>
> According to the Table 2 in the main paper, the attack success rate largely depends on the target classes we choose. If the target classes are the best class, the classes with highest confidence among the classes other than the true labels, it is easier to attack than the randomly chosen target or worst target classes.
>
> ---

---

> ### Author Response · Authors · 2024-11-21
> **Response to Reviewer LM2j (pt 3/4)**
>
> **C6, Q1. What would be the impact of larger $K$ on ASR?**
>
> In response to your query, we conduct attacks with higher values of $K$ to evaluate its impact on ASR. As shown in Table 4, we report the ASR against ResNet50 for targeted attacks, where the $top$-$K$ predicted labels are replaced by the target classes. Table 5 presents the ASR for untargeted attacks on 100 ImageNet images, where the $top$-$K$ predicted labels of benign inputs are replaced by arbitrary labels. Furthermore, Table 6 illustrates the ASR variation for attacks on Inception-V3 using 100 images from the PASCAL VOC2012 dataset. From these results, we observe that ASR decreases as $K$ increases, with a sharper decline for targeted attacks. This trend suggests that the proposed black-box attack, which lacks knowledge of the target model, becomes ineffective as $K$ approaches the extreme of $K = N/2$ under the perturbation constraint, where $N$ is the number of labels in the dataset. Given that the ImageNet dataset includes 1,000 classification labels, we anticipate observing a similar trend in attacking models trained on the OpenImages dataset, given the comparable scale and diversity of their label distributions.
>
>
> | $K$  | 1    | 3    | 5    | 10    | 15    | 20    |
> |-------------|:------:|:------:|:------:|:------:|:------:|:------:|
> |ASR(%) | 100.00 | 100.00 | 98.00 | 69.00 | 36.00 | 11.00|
>
> **Table 4**: ASR against ResNet50 on 100 ImageNet images for different $top$-$K$ targeted attack with a perturbation threshold $r_{th}=20$.
>
>
> | $K$  | 1    | 5    | 10    | 20    | 30    | 40    | 50   |
> |-------------|:------:|:------:|:------:|:------:|:------:|:------:|:------:|
> |ASR(%) | 100.00 | 99.00 | 84.00 | 71.00 | 54.00 | 43.00| 30.00 |
>
> **Table 5**: ASR(%) against ResNet50 on 100 ImageNet images for different $top$-$K$ untargeted attack with a perturbation threshold $r_{th}=20$, where the $top$-$K$ predicted label of the source images are replaced by the any $K$ labels among the rest.
>
>
> | $K$  | 1    | 3    | 5    | 7    | 9   |
> |-------------|:------:|:------:|:------:|:------:|:------:|
> |ASR(%) |100.00 | 82.00 | 61.00 | 21.00 | 2.00 |
>
> **Table 6**: ASR(%) against Inception-V3 on 100 PASCAL VOC images for different $top$-$K$ targeted attack  considering best target classes with a perturbation threshold $r_{th}=20$.
>
> ---
>
> **Q2. How the target labels are selected for targeted attack setting for ImageNet experiments?**
>
> In the ImageNet experiments, the $K$ target labels for a benign input image are determined as follows: we randomly select a target image whose true label differs from that of the benign input image. The $top$-$K$ target labels are then set as the $top$-$K$ predicted labels of the selected target image. In the main paper, we briefly discuss it in lines 455-457.

---

> ### Author Response · Authors · 2024-11-21
> **Response to Reviewer LM2j (pt 4/4)**
>
> **Q3. For DCT, what percentage of low frequency components were selected? How does the performance vary with a different number of components?**
>
> For an image with dimension $C_h\times W \times H$ and with a dimension reduction factor $f$ to sample low-frequency noise, the dimension of the low frequency noise, we consider, is $C_h\times \frac{W}{f} \times \frac{H}{f}$. Hence, if the dimension of an image is $3 \times 224 \times 224$, the dimension of the low frequency space with dimension reduction factor $f=4$ would be 9408. This corresponds to selecting 6.25\% of the frequency components for our experiments. We choose dimension reduction factor as $f=4$ for all our experiments following [1,2]. The median $\ell_2$-norm of perturbations for different query budgets and different dimension reduction factor for targeted $top$-2 attack on 100 images are given bellow that conforms with the finding in [1] even for $top$-2 attacks.
>
>
> | $f$ | 1 | 2 | 3 | 4 | 6 | 8 |
> |-------------|:------:|:------:|:------:|:------:|:------:|:------:|
> | Q=4000 | 15.82 | 12.55 | 11.74 | 9.29 | 11.72 | 10.22 |
> | Q=8000 | 10.30 | 8.47 | 7.05 | 5.72 | 6.40 | 6.92 |
>
> **Table 7**: Median $\ell_2$-norm of perturbations (lower is better) for different query budgets and different dimension reduction factors for targeted attack on 100 ImageNet images against ResNet50.
>
> [1] Reza, Md Farhamdur, et al. ”Cgba: curvature-aware geometric black-box attack.” Proceedings
> of the IEEE/CVF International Conference on Computer Vision. 2023.
>
> [2] Li, Huichen, et al. "Qeba: Query-efficient boundary-based blackbox attack." Proceedings of the IEEE/CVF conference on computer vision and pattern recognition. 2020.
>
>
> ---
>
> **Q4. Is there any any correlation of the prediction among the classes?**
>
> You have rightly pointed out that labels are often correlated, and the attack success rate (ASR) can vary depending on the target class. As shown in Table 2 of the paper, we observe a higher ASR for $top$-$2$ targeted attacks when using the best target samples. However, when comparing ASR between randomly chosen targets and the worst targets (i.e., the two classes with the lowest prediction probabilities), the ASR is actually higher for the worst targets in the top-2 targeted attack scenario.
> One possible explanation is that the worst target classes, having similar prediction probabilities, may be closer to each other in the feature space. As a result, increasing the confidence for one target class may inadvertently boost the confidence for its neighbor, leading to a higher ASR. In contrast, randomly chosen target classes are less likely to be correlated and might lie farther apart in the feature space, making them harder to attack.

---

> ### Author Response · Authors · 2024-11-26
> **Looking Forward to Your Feedback**
>
> Dear reviewer LM2j,
>
> Thank you for your time and valuable feedback on our paper. In our response, we have carefully addressed all of your comments and questions. Please let us know if you require further clarification. We appreciate your insights and look forward to your continued feedback.

---

> ### Comment · Reviewer_LM2j · 2024-11-27
>
> Thank you to the authors for their detailed responses. While most of my concerns have been addressed, some questions remain, as noted in my responses. One key limitation is that targeting larger target sizes (K) leads to significantly lower success rates (as shown in response to C6). Clearly highlighting this limitation would help motivate future work in this area.
>
> ## Response to C4:
> Assume that we have an initial point $x_s$ and using the given algorithm, we can compute a point $x_{b_n}$ on the decision boundary. Assume that we are considering top-3 targeted attack and we have the following predicted probabilities by the model:
>
> $P(x_{b_n}) = [0.3, 0.3, 0.3, 0.01,….]$
>
> To find the subsequent boundary point $x_{b_{n+1}}$, Eq (11) (in the revised version) requires that the next boundary point that we try to find, should have increased confidence in the target classes. This is because of the weight:
> $w_{x_{b_n}, c, z_i} = P_c(x_{b_n}+z_i) - P_c(x_{b_n}) >0$
>
> At the boundary point $x_{b_n}$, assume that now we find a vector $z_i$ such that $\\|x_{b_n} +z_i - x_s\\|\_2 < \\|x_{b_n} - x_s\\|\_2 $ i.e., the new boundary point based on $z_i$ has smaller L2 distortion as compared to the initial point found on the boundary. Assume that the model predicts the class probabilities as:
>
> $P(x_{b_n} + z_i) = [0.28, 0.28, 0.28, 0.016,….]$
>
> Note that $(x_{b_n} + z_i)$ still leads to a successful attack as the top-3 classes are our desired classes but with lower confidence as compared to $x_{b_n}$. However, using Eq (11), $\mathbb{1}(x_{b_n} + z_i) = 1$ and the weights $w_{x_{b_n}, c, z_i}$ would all be -1 for the target classes. Hence, $\zeta_{x_{b_n},c,z_i} = 0$ for all target classes and therefore $z_i$ would not contribute to compute the gradient in Eq(12).
>
> I strongly think that this is the key reason that the proposed method doesn't perform well for small L2 norms and it would be great if it is actually addressed or talked about in the paper.
>
> ## Response to C6 - Q1 and Q3:
> Please mention this in the main paper and include these results in the appendix to make the limitations clear. For Q3, please include the attack success rates as well.

---

> > ### Author Response · Authors · 2024-11-28
> >
> > **Response to C4**
> >
> > We like to thank the reviewer for addressing one possible limitation that, the reviewer thinks, might happen in our proposed attack. Further explanation is provided below.
> > To illustrate the limitation, it is assumed the prediction probabilities on a boundary point $P(x_{b_n}) = [0.3,0.3,0.3,0.01,…]$. Also, a vector $z_i$ is considered such that $P(x_{b_n}+z_i )=[0.28,0.28,0.28,0.016,…]$.
> > In this given example, $\min_{c \in \mathcal{Y}\_K^{(t)}}P_c(x_{b_n})=0.3$ and $\max_{c \in [C] \setminus \mathcal{Y}\_K^{(t)}}P_c(x_{b_n})=0.01$.
> > In this hypothetical setting, any query with small added noise $z_i$ (without changing the top 3 classes) would be adversarial, i.e., $\mathbf 1(x_{b_n}+z_i )=1$. But actually, this is quite unlikely to happen on a decision boundary point as, on a decision boundary point we usually have $\min_{c \in \mathcal{Y}\_K^{(t)}}P_c(x_{b_n}) \approx \max_{c \in [C] \setminus \mathcal{Y}\_K^{(t)}}P_c(x_{b_n})$.
> > To see this, consider attacks against $top$-$2$ multilabel learning with PASCAL VOC 2012 dataset in our study. The prediction probabilities up to top-5 predicted labels on the decision boundaries at different iterations for attacks on 2 benign images are given below.
> >
> > | Boundary point | 1st        | 2nd        |  | 3rd        | 4th        | 5th        |
> > |----------------|------------|------------|-|------------|------------|------------|
> > | $x_{b_0}$     | 0.38064867 | 0.22667292 | | 0.2266728  | 0.05331969 | 0.03761479 |
> > | $x_{b_{50}}$     | 0.46588376 | 0.19737633 | | 0.19736484 | 0.04510405 | 0.01704748 |
> > |$x_{b_{100}}$    | 0.42662048 | 0.21430703 | | 0.21430257 | 0.04498401 | 0.01717781 |
> > | | | | | | | |
> >
> > | Boundary point | 1st        | 2nd        |  | 3rd        | 4th        | 5th        |
> > |----------------|------------|------------|-|------------|------------|------------|
> > | $x_{b_0}$     | 0.45030260 | 0.25157195 |  | 0.25157170 | 0.04441001 | 0.00097624 |
> > | $x_{b_{50}}$     | 0.31688127 | 0.28130546 |  | 0.28126410 | 0.11849701 | 0.00123801 |
> > |$x_{b_{100}}$    | 0.29548132 | 0.28693030 |  | 0.28691715 | 0.12917086 | 0.00085085 |
> > | | | | | | | |
> >
> > In the tables above, the first two prediction probabilities correspond to the target classes, and the $3^{rd}$ one represents the maximum prediction probabilities of the class other than the target classes at different boundary points.
> > As can be seen, $\min_{c \in \mathcal{Y}\_K^{(t)}}P_c(x_{b_n}) \approx \max_{c \in [C] \setminus \mathcal{Y}\_K^{(t)}}P_c(x_{b_n})$ holds, and similar observations have been made for attacks on other benign samples and larger $K$ values.
> > As $\min_{c \in \mathcal{Y}\_K^{(t)}}P_c(x_{b_n}) \approx \max_{c \in [C] \setminus \mathcal{Y}\_K^{(t)}}P_c(x_{b_n})$, if $w_{x_{b_n},c',z_i }=P_{c'} (x_{b_n}+z_i )-P_{c'} (x_{b_n})<0$, where $c'=  \arg\min_{c \in \mathcal{Y}\_K^{(t)}} P_c(x_{b_n})$, it is highly likely that $\mathbf 1(x_{b_n}+z_i)=-1$, and vice versa (for additional information, please refer to the response to Reviewer hY9G(Q1)).
> > For instance, from the 2nd table, $P(x_{b_{50}})=[0.31688127,0.28130546,0.28126410,…]$. Assume $z_i$ changes the prediction probability to $P(x_{b_{50}}+z_i)=[0.31588127,0.28030546,0.28226410,…]$. For this example, $1(x_{b_{50}}+z_i)=-1$ and $\zeta_{x_{b_n},c ,z_i}=1$ for all target classes.
> >
> > Having a boundary point $x_{b_n}$, CGBA$^K$ locates $x_{b_{n+1}}$  such that $||x_{b_{n+1}}-x_s||\_2 <||x\_{b\_{n}}-x_s||\_2$.
> > However, $x_{b_{n+1}} \neq x_{b_{n}}+z_i$ or $x_{b_{n+1}} \neq x_{b_{n}}+\epsilon \hat{g}\_{x_{b_n}}$;
> > instead, it uses $\lbrace z_i \rbrace_{i=0}^{I_n}$ queries to estimate $\hat{g}\_{x_{b_n}}$, and guided by $\hat{g}\_{x_{b_n}}$ and $(x_{b_n}-x_s)/||x_{b_n}-x_s||\_2$,
> > it exploits a semicircular trajectory to locate the $x_{b_{n+1}}$ with reduced perturbation, as depicted in Fig. 3(c).
> > The new boundary point does not necessarily enhance the prediction probabilities of the target classes, as in the given table. However, it ensures finding a boundary point with reduced perturbation satisfying the adversarial objective.
> >
> > ---
> >
> > **Response to C6-Q1 and Q3**
> >
> > Thank you. Regarding Q3, we included the results of the attack success rate in Appendix F of the revised version. This is mentioned in the revised version of the main paper (lines 529-531).

---

### Official Review · Reviewer_8mHS · 2024-11-03

**Soundness:** 3
**Presentation:** 2
**Contribution:** 3
**Rating:** 6
**Confidence:** 4

**Summary:**

This paper presents a score-based and query-efficient blackbox adversarial attack method. The proposed method is designed based on GSBA, which searches for the adversarial samples close to the classification boundary along a semi-circular path. The merit of the proposed method is more accurate gradient estimation to update incrementally the adversarial sample in the adversarial region and along the classification boundary. This study applies the method to a challenging top-k adversarial attack scenario and demonstrate its superior performance compared to the baselines in an empirical comparison.

**Strengths:**

This study offers a query-efficient solution to a challenging top-K adversarial attack problem and organise a comprehensive study in both targeted and untargeted attack scenarios for single-label and multi-label learning tasks. The core idea is to boost the blackbox gradient estimator by reweighing different sample points around the current estimate, instead of treating them equally. The weight assigned to each sampled point is defined by measuring the decision confidence variation. The sampled points are considered, only if they cause the decision confidence scores all decrease with respect to the targeted top-K classes towards the adversarial region. This weighing technique better tunes the gradient estimator to be aligned to the goal of adversarial attacks. This design improves the efficiency of attacks, rather than initialising the attack with a random or less attack-optimised gradient estimates.

**Weaknesses:**

I have the following suggestion and comments over the limitation of this study.

(1) This paper needs significant improvement to get a clear logic structure. The attack according to Figure.3 is initialised by locating an adversarial sample along the classification boundary, then followed by searching for more closer adversarial samples in the adversarial region along a semi-circular path. Therefore, Section 4.2 should be presented before 4.1.

(2) Reweighing the sampled point to form a more accurate gradient estimator also reduces the number of sampled points $\{z_{i}\}$ that are useful and have non-zero weights in gradient estimation. As a result, it may produce a better gradient estimation, yet require more sampled points especially at the beginning iterations of the attack.

(3) Why do you set the number of the queried points as $\lfloor{I_{0}\sqrt{n+1}}\rfloor$ ?

**Questions:**

1/ Regarding the second bullet point in the comments of weakness, please provide further discussion if the reweighing technique may require to increase the number of queries, since the reweighing step can exclude some sampled points from estimating the gradients.

2/ Please justify the third bullet point with further discussions.

---

> ### Author Response · Authors · 2024-11-20
>
> Thank you for your insightful comments and valuable suggestions. As recommended, we will reorder the sections, presenting Section 4.2 before Section 4.1 in the revised version. Below, we provide detailed responses to the questions you raised.
>
> ---
>
> **Q1. Does reweighing the sampled points to form a more accurate gradient estimation require more sampled points?**
>
> You correctly pointed out that our proposed gradient estimation technique, as described in Eq. (7), filters out certain sample points during gradient estimation, even when these points are assigned non-zero weights. However, our method specifically filters out anomalous sample points—those are adversarial yet exhibit a reduction in the confidence of all target classes, or vice versa.
>
> The benefits of excluding such anomalous queries are demonstrated in Fig. 7 in the Appendix. Filtering out these queries significantly enhances performance. For example, when comparing Approach 1 and Approach 6 in the top-1 setting, where the weights associated with $\mathbf z_i$ are $\lbrace -1,1\rbrace$ for Approach 1 and $\lbrace-1,0,1\rbrace$ for Approach 6, the latter shows notable improvement by assigning a weight of 0 to anomalous queries, as illustrated in Fig. 7(a). This highlights that excluding anomalous queries mitigates their negative impact on gradient estimation, thereby improving overall performance.
>
> Regarding the need for more sample points, our proposed method maintains a fixed number of queries for gradient estimation, irrespective of the presence of anomalous queries. Therefore, the approach does not consider additional samples when it filters out any. In other words, this reweighing approach effectively improves the quality of gradient estimation without the need to increase the number of queries.
>
> ---
>
>
> **Q2. Why is the number of query points set to $\lfloor I_0\sqrt{n+1} \rfloor$ ?**
>
> Consistent with existing geometric decision-based black-box attack methods [1, 2], we set the number of queries at the $n$-th iteration to $\lfloor {I_0\sqrt{n+1}} \rfloor$. These approaches incrementally increase the number of queries with each iteration to enable more accurate gradient estimation. By improving the gradient estimation, these methods effectively identify better directions to further reduce the perturbation magnitude.
>
>
> [1] Chen, Jianbo, Michael I. Jordan, and Martin J. Wainwright. "Hopskipjumpattack: A query-efficient decision-based attack." 2020 ieee symposium on security and privacy (sp). IEEE, 2020.
>
> [2] Reza, Md Farhamdur, et al. "Cgba: curvature-aware geometric black-box attack." Proceedings of the IEEE/CVF International Conference on Computer Vision. 2023.

---

> ### Author Response · Authors · 2024-11-26
> **Looking Forward to Your Feedback**
>
> Dear reviewer 8mHS,
>
> Thank you for your time and valuable feedback on our paper. In our response, we have carefully addressed all of your comments and questions. Please let us know if you require further clarification. We appreciate your insights and look forward to your continued feedback.

---

### Author Response · Authors · 2024-11-23
**Request for Feedback on Rebuttal**

Dear Reviewers,

Thank you for your time and valuable feedback on our paper. In our response, we have carefully addressed your comments and questions. If you require any further clarification, please let us know. We greatly appreciate your insights and look forward to your continued feedback.

Best regards,
Authors of submission #5766

---

### Author Response · Authors · 2024-11-24
**General Response to All the Reviewers**

We sincerely thank all reviewers for their valuable time, efforts, and constructive feedback on our paper. In addition to providing detailed responses to individual reviewers, we would like to: 1) express our gratitude for the reviewers' recognition of our work; 2) summarize the key concerns raised; and 3) highlight the major improvements made in the revised version of our paper.

---

**Key Contributions Acknowledged by Reviewers:**
  - **Novel Query-Efficient Solution:** The paper introduces a novel query-efficient approach to addressing the challenging $top$-$K$ adversarial attack problem. [8mHS, LM2j, hY9G, uZC7]
  - **Significance:** The proposed method demonstrates strong technical rigor, particularly in its gradient approximation techniques. [8mHS, LM2j, hY9G, uZC7]
  - **Clear and Well-Organized Presentation:** The paper is well-organized, with geometrical intuitions effectively illustrated through detailed figures. [hY9G]
  - **Comprehensive Experiments and Insights:** The ablation study and experiments provide valuable insights into the $top$-$K$ attack problem. [uZC7]

---

**Major Concerns Raised by Reviewers:**
  - **Logical Flow in Gradient Estimation:** The discussion of gradient estimation within the non-adversarial region should precede the discussion of gradient estimation on the decision boundary. [8mHS, LM2j]
  - **Avoid Duplication of Long Expressions:** The long gradient expression should be simplified. [hY9G]
  - **Impact of Larger K on ASR:** The effect of larger values of $K$ on the attack success rate (ASR) requires further explanation. [LM2j, hY9G]
  - **Choice of Low-Dimensional Frequency Subspace:** The impact of varying subspace dimensions in ASR experiments should be thoroughly analyzed. [LM2j]
  - **Explanation of the Observation $\zeta_{\mathbf x_{b_n}, c, \mathbf z_i} = \zeta_{\mathbf x_{b_n}, c, -\mathbf z_i}$:** The explanation of this observation should be clarified. [LM2j, hY9G]
  - **Writing Improvements:** Sections 4.1 and 4.2 need clearer and more concise writing. [uZC7]

---

**Major Updates in the Revised Version:** In this rebuttal, we provide detailed responses to reviewers’ comments and questions. Based on the reviewers’ suggestions, we made the following changes to our revised version:
  - **Revised Logical Flow:** In the revised version, we present gradient estimation within the non-adversarial region before gradient estimation on the decision boundary. Specifically, we swapped Sections 4.1 and 4.2 from the original version and improved the writing for better clarity.
  - **Simplified Gradient Expressions:** Following reviewer hY9G's suggestion, we have simplified the long gradient expressions.
  - **Impact of Larger $K$:** We have included a discussion of the impact of larger K in Appendix E of the revised version.
  -  **Impact of Low-Dimensional Frequency Subspaces:** Following reviewer LM2j's suggestion, we added a discussion on the impact of different low-dimensional frequency subspaces in Appendix F of the revised version.
  - **Highlighting Changes:** All major revisions are highlighted in blue in the revised manuscript.

---

**Additional Note on Reviewer LM2j’s Suggestion:** Reviewer LM2j suggested including the standard deviation of the results in Table 2 for different targets, potentially using different seeds. While we appreciate this suggestion, Table 2 in the original submission already includes results for three target types across $top$-1 and $top$-2 settings for various $r_{th}$ values. Recomputing all these results with different seeds would require significant time. Given this constraint, we propose including the standard deviations in the final version if allowed. For now, we provide results showing the standard deviations for different $r_{th}$ values in the $top$-2 attack setting using our proposed method, as part of our response to reviewer LM2j.

---

**Note: We would like to clarify that all equation numbers, figure numbers, and table numbers referenced in our detailed responses correspond to the original submission, which is included as supplementary material in this revised submission. Due to swapping of Section 4.1 and Section 4.2, the reference numbers inevitably change. We kindly request reviewers to refer to the original version for consistency.**

---

---

### Comment · Reviewer_uZC7 · 2025-03-16
**Suggestions for Improving the Formulas in the Paper**

Dear authors,

I appreciate the authors' contributions to the field of black-box attacks. However, there are some improvements that can be made to enhance the quality of the paper. For instance, in Eq. (9), Eq. (10), Eq. (12), Eq. (16), Eq. (17), Eq. (19), Eq. (23), and Eq. (24), the parentheses around the summation symbol are too small and should be adjusted using **\left(** and **\right)**. Similarly, in Eq. (14), Eq. (21), and Eq. (22), the parentheses around the `min` symbol are also too small and should be adjusted accordingly.

Please review each equation carefully to ensure that the parentheses are appropriately sized, especially before the Camera Ready submission deadline.

Thank you!

---

### Meta-Review · Area_Chair_UbXA · 2024-12-18

**Metareview:**

The paper proposes a score-based black-box adversarial attack targeting the top-K classification problem that is widely used in multi-label settings. It estimates the decision confidence by reweighing different sample points around the current estimate so that the confidence can later be used to calculate the gradient direction. Although there are some organization and written issues, the proposed method is novel and a good attempt at a challenging problem. The experiments are comprehensive and the results look promising. Therefore, I recommend accepting this work.

**Additional Comments On Reviewer Discussion:**

The author did a good job addressing all reviewer's concerns on the gradient estimation, paper organization, and some parameter choices.
The author also includes several important experiments that have been added to make the proposed method solid. Please remember to include the reviewer's suggestions in the revision.

---

### Decision · Program_Chairs · 2025-01-22

Accept (Poster)